# Multi-aspect Knowledge Distillation with Large Language Model

## Abstract

Recent advancements in deep learning have significantly improved performance on computer vision tasks. Previous image classification methods primarily modify model architectures or add features, and they optimize models using cross-entropy loss on class logits. Since they focus on classifying images with considering class labels, these methods may struggle to learn various *aspects* of classes (e.g., natural positions and shape changes). In contrast, humans classify images by naturally referring to multi-aspects such as context, shape, color, and other features. Inspired by this, rethinking the previous approach from a novel view, we propose a multi-aspect knowledge distillation method using Multimodal Large Language Models (MLLMs). Our approach involves: 1) querying Large Language Model with multi-aspect questions relevant to the knowledge we want to transfer to the model, 2) extracting corresponding logits from MLLM, and 3) expanding the model's output dimensions to distill these multi-aspect logits. We then apply cross-entropy loss to class logits and binary cross-entropy loss to multi-aspect logits. Through our method, the model can learn not only the knowledge about visual aspects but also the abstract and complex aspects that require a deeper understanding. We primarily apply our method to image classification, and to explore the potential for extending our model, we expand it to other tasks, such as object detection. In all experimental results, our method improves the performance of the baselines. Additionally, we analyze the effect of multi-aspect knowledge distillation. These results demonstrate that our method can transfer knowledge about various aspects to the model and the aspect knowledge can enhance model performance in computer vision tasks. This paper demonstrates the great potential of multi-aspect knowledge distillation, and we believe it offers a promising direction for future research in computer vision and beyond.

## 1 Introduction

Recent advancements in deep learning models have led to significant performance improvements in the field of computer vision, including image classification Vaswani et al. (2017); Vasu et al. (2023); Zhu et al. (2023); Novack et al. (2023), object detection Wu et al. (2023); Ma et al. (2023); Wang et al. (2023), and generative models Lee et al. (2023); Kwon et al. (2024); Lee et al. (2024). In particular, these advancements, primarily focusing on improving model architectures or incorporating additional features, have greatly enhanced performance in image classification. The methods Vaswani et al. (2017); Liu et al. (2021); Zhu et al. (2023); Tan & Le (2019); He et al. (2016) output class logits and use cross-entropy loss to optimize the models.

However, even if the images in a dataset belong to different classes, they can consist of similar features and make the task more challenging Wei et al. (2021); Parkhi et al. (2012); Krause et al. (2013); Fei-Fei et al. (2004); Wah et al. (2011); Cimpoi et al. (2014). For instance, in CUB200 dataset Wah et al. (2011), most classes share the same features that the superclass "bird" has; i.e. beak, two wings, two legs, and so on. This may require not only the class logit but also additional visual features or aspects that require deeper understanding.

How can humans effectively classify fine-grained images? When classifying fine-grained images, humans not only consider the detailed visual aspects of the given image but also take into account abstract and complex aspects that require a more profound understanding Rong et al. (2021). For

example, when given a fine-grained image of a bird, humans might think along the lines of "The beak is sharp," or "There is a river nearby," combining both detailed visual features and contextual information.

Inspired by this human ability, the question arises: Could the model's performance improve if we transfer knowledge about various aspects to it? Multi-modal Large Language Models (MLLMs) have also made significant advancements alongside Large Language Models (LLMs). By taking multi-modal inputs, MLLMs Liu et al. (2024b; 2023); Achiam et al. (2023) can understand and effectively represent visual information, enabling tasks such as visual understanding Guo et al. (2023a); Yang et al. (2022); Tsimpoukelli et al. (2021) and image captioning Li et al. (2023); Zhang et al. (2021); Wang et al. (2021). Additionally, since MLLMs can answer abstract or complex questions, unlike image classification models Vaswani et al. (2017); Liu et al. (2021); Zhu et al. (2023); Tan & Le (2019); He et al. (2016) that output class logits, we can use MLLMs to transfer various knowledge that may help classification to the model.

Rethinking previous methods from a novel view, we propose a simple yet effective multi-aspect knowledge distillation method using MLLM. Our method consists of three main stages.

First, as shown in Figure 1, we generate questions about the *aspects* the model aims to learn, based on the classes of the dataset, using the LLM. The generated questions represent the *aspects* that the model aims to learn during training. Secondly, we provide the generated questions to MLLM to obtain the logits of each aspect. Since MLLM can understand visual information and answer abstract questions, the logits of the MLLM may represent knowledge of the diverse aspects about the dataset. Finally, to distill these extracted multi-aspect logits, we simply expand the dimension of the model's output by adding the number of aspects to the number of classes, and then we optimize the model by applying cross-entropy loss to the class logits and binary cross-entropy loss to the aspect logits.

Through our method, we transfer knowledge about the aspect we want the model to learn, enabling the model to understand and learn various aspects of the data, which may be helpful for computer vision tasks.

We conduct experiments on fine-grained and coarse-grained image classification with various neural networks. Our method outperforms the baselines. Additionally, we analyze the impact of aspect knowledge and discuss the correlations between the aspects and performances of the models. Also, to explore the potential for extending our model, we expand it to other tasks, such as object detection and knowledge distillation.

In summary, our contributions are as follows:

- We propose a novel, simple yet effective multi-aspect knowledge distillation using MLLM.

- To the best of our knowledge, we are first to provide the novel view of distilling multi-aspect knowledge about abstract and complex aspects that require a deeper understanding, extending the model's output dimensions. This enables the model to learn not only about the class but also about these diverse aspects.

- We primarily apply our method to image classification, and to explore the potential for extending our model, we expand it to other tasks, such as object detection. In all experimental results, our method improves the performances of the baselines. These results demonstrate the potential of our method to be effective and easily applicable to a variety of tasks. Furthermore, we provide analysis regarding the aspects.

## 2 RELATED WORK

**Multimodal Large Language Models.** Recently, Multimodal Large Language Models (MLLMs) Achiam et al. (2023); Alayrac et al. (2022); Liu et al. (2024b); Yin et al. (2023); Zhang et al. (2024) have shown significant performance improvements in multi-modal problems such as visual question answering and image captioning by leveraging large-scale datasets to learn a joint embedding space where images and their corresponding textual descriptions are closely aligned. GPT-4o Achiam et al. (2023) has the ability to get the context and has a human-like text generation ability, showing strong performance not only in the natural language processing area but also in multi-modal tasks. InternVL Chen et al. (2024) can address both text and image data and shows

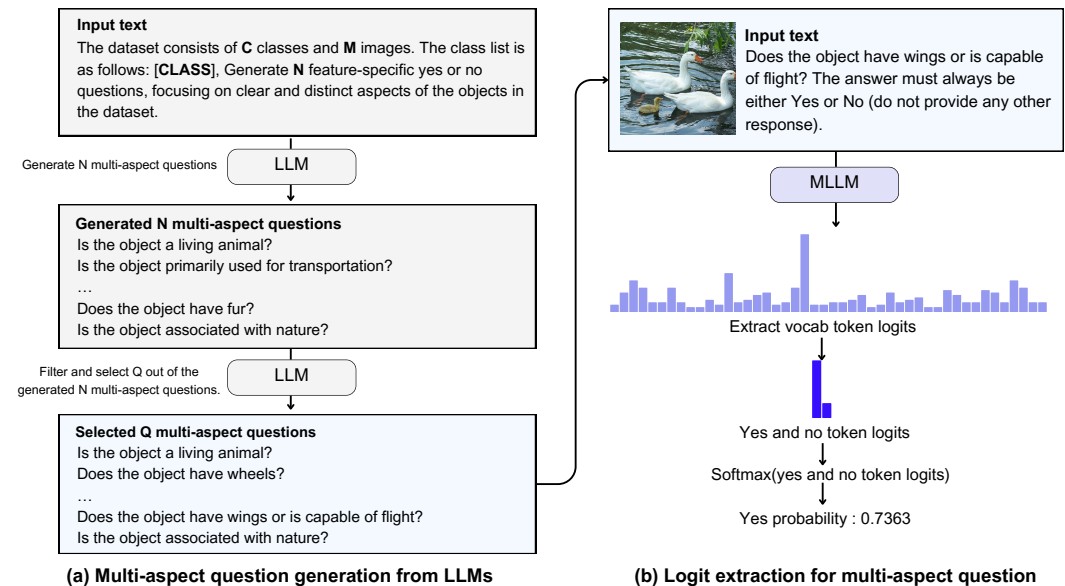

(a) Multi-aspect question generation from LLMs          (b) Logit extraction for multi-aspect question

Figure 1: **Multi-aspect question generation and logit extraction.** For multi-aspect question generation (a), we generate various aspect questions from the LLM by using the class and prompt as instructions. For logit extraction about multi-aspect questions (b), we input the generated multi-aspect questions along with the image into the MLLM to extract logits and obtain the probabilities corresponding to yes token.

better performances in various multimodal tasks (such as visual understanding, language generation, and visual QA) while using fewer computing resources compared to other MLLMs. Motivated by this, we apply the rich knowledge of MLLMs to image classification.

**Visual tasks with linguistic information.** Many studies Berrios et al. (2023); Menon & Vondrick (2022); Pratt et al. (2023); Yan et al. (2023); Salewski et al. (2024); Yang et al. (2023) try to extract linguistic information from a large language model and use it to settle the visual problems. One method Menon & Vondrick (2022) leverages the linguistic knowledge for each visual category from LLM to generate the descriptions and use the descriptions in zero-shot image classification. Another method Yan et al. (2023) creates the concise set of representative visual attributes from LLM by leveraging their learning-to-search method for interpretable visual recognition. While these methods focus on generating attributes for model training, our approach distills knowledge about various aspects, extending the model's output dimensions.

## 3 METHODOLOGY

### 3.1 MULTI-ASPECT QUESTION GENERATION FROM LLM

Our method is illustrated in Figure 1. First, as shown in Figure 1 (a), we create a total of $N$ multi-aspect questions based on the class labels of the dataset using LLM. Then, considering visual, categorical, and environmental aspects, we filter and select $Q$ multi-aspect questions using the LLM. $Q$ is the number of multi-aspect questions we want to transfer to our model. We use GPT-4o with the system prompt, "You are a good question maker.", and the instructions, "The dataset consists of $C$ classes and $M$ images. The class list is as follows: $[CLASS]$, Generate $N$ feature-specific yes or no questions, focusing on clear and distinct aspects of the objects in the images in the dataset." and "Select $Q$ of the most relevant and distinct questions from the list, focusing on various key features that distinguish different class in the dataset.". These generated aspect questions represent the knowledge we aim to transfer to the models based on datasets.

### 3.2 LOGIT EXTRACTION FOR MULTI-ASPECT QUESTIONS

We generate questions about aspects to be transferred to the model from the LLM. As shown in Figure 1 (b), using an MLLM, we input the dataset and the generated multi-aspect questions, prompting

it to answer yes or no. We then extract the logits corresponding to yes and no tokens, and apply the softmax function to both the yes and no logits. We use the softmax results of the yes logits as the targets. Let $i$ be the question index, $z_{y_i}$ be the logit for yes for the $i$-th question and $z_{n_i}$ be the logit for no for the $i$-th question respectively. The softmax probability $q_i$ is given by:

$$q_i = \frac{e^{z_{y_i}}}{e^{z_{y_i}} + e^{z_{n_i}}} \tag{1}$$

### 3.3 EXPANSION OF MODEL OUTPUT DIMENSION

To distill knowledge about multi-aspect questions into the model, we simply expand the dimension of model output. If the number of classes is $C$ and the number of multi-aspect questions is $Q$, then the dimension of the model's output $D$ is:

$$\mathbf{D} = C + Q \tag{2}$$

Also, we consider the expanded dimension $D$ such that from 1 to $C$ is the class logit dimension, and from $C + 1$ to $D$ is the aspect logit dimension. The multi-aspect logit dimension is used for the distillation of logits representing the multi-aspect questions. We provide the detail figure in the supplementary materials.

### 3.4 MUTLI-ASPECT KNOWLEDGE DISTILLATION LOSS

To distill multi-aspect logits, we extend the model outputs by the number of multi-aspect questions $Q$. The class logit dimension of model output is applied with cross-entropy loss, and the aspect logit dimension is applied with binary-cross entropy loss because we use the probability of the yes token extracted from the MLLM as the target. Let $C$ be the number of classes and $Q$ be the number of multi-aspect questions. We expand the model output to $D$. We apply cross-entropy loss to the outputs from 1 to $C$ for class classification, and binary-cross entropy loss from $C + 1$ to $D$ using multi-aspect probability $q$ as the target.

$$\hat{y} = [\hat{y}_1, \hat{y}_2, \ldots, \hat{y}_C, \hat{y}_{C+1}, \ldots, \hat{y}_D] \tag{3}$$

$$\mathcal{L}_{\text{CE}} = -\sum_{i=1}^{C} y_i \log \hat{y}_i \tag{4}$$

$$\mathcal{L}_{\text{MaKD}} = -\sum_{i=1}^{Q} [q_i \log(\hat{y}_{C+i}) + (1 - q_i) \log(1 - \hat{y}_{C+i})] \tag{5}$$

where $\hat{y}$ represents the predicted probability, $y$ are the true labels for the classes, $q$ are the targets for the aspects extracted from the MLLM and $\alpha$ is a factor for balancing the losses. The total loss is defined as follow:

$$\mathcal{L}_{\text{total}} = \mathcal{L}_{\text{CE}} + \alpha \mathcal{L}_{\text{MaKD}} \tag{6}$$

Through our approach, the model can learn both classification capabilities and the ability to understand abstract and complex concepts by distilling knowledge about the aspects from the MLLM.

## 4 EXPERIMENTS

### 4.1 IMPLEMENTATION DETAILS

**Multi-aspect question generation from LLM.** We create a total of 100 multi-aspect questions, and then tune and select the number of multi-aspect questions based on the dataset and neural network according to Section 3.1. We use GPT-4o for the generation of multi-aspect questions. Additionally,

Table 1: **Accuracy (%) on the fine-grained image test set.** We use a total of six datasets (StanfordCars Krause et al. (2013), OxfordPets Parkhi et al. (2012), DTD Cimpoi et al. (2014), 102Flowers Nilsback & Zisserman (2008), CUB200 Wah et al. (2011), and FGVC-Aircraft Maji et al. (2013)). MLLM is InternVL2-8B. Base is the baseline using cross-entropy loss with class labels. We run each experiment three times and report the average results.

**(a) StanfordCars**

|  | Zero-shot classification | | |
| --- | --- | --- | --- |
| MLLM | 14.30 | | |
|  | Base | Ours | Gap |
| ResNet18 | 77.53 | **83.38** | +5.85 |
| ResNet34 | 80.93 | **84.33** | +3.40 |
| MobileNet-V1 | 82.84 | **85.43** | +2.59 |
| EfficientNet | 86.41 | **88.07** | +1.66 |

**(b) OxfordPets**

|  | Zero-shot classification | | |
| --- | --- | --- | --- |
| MLLM | 49.38 | | |
|  | Base | Ours | Gap |
| ResNet18 | 77.07 | **82.24** | +5.17 |
| ResNet34 | 79.07 | **82.78** | +3.71 |
| MobileNet-V1 | 78.12 | **82.75** | +4.63 |
| EfficientNet | 83.42 | **85.27** | +1.85 |

**(c) DTD**

|  | Zero-shot classification | | |
| --- | --- | --- | --- |
| MLLM | 49.20 | | |
|  | Base | Ours | Gap |
| ResNet18 | 55.73 | **59.43** | +3.70 |
| ResNet34 | 53.76 | **59.89** | +6.13 |
| MobileNet-V1 | 57.22 | **61.44** | +4.22 |
| EfficientNet | 60.28 | **62.87** | +2.59 |

**(d) 102Flowers**

|  | Zero-shot classification | | |
| --- | --- | --- | --- |
| MLLM | 26.88 | | |
|  | Base | Ours | Gap |
| ResNet18 | 92.32 | **94.64** | +2.32 |
| ResNet34 | 92.75 | **94.89** | +2.14 |
| MobileNet-V1 | 94.14 | **95.56** | +1.42 |
| EfficientNet | 95.86 | **96.78** | +0.92 |

**(e) CUB200**

|  | Zero-shot classification | | |
| --- | --- | --- | --- |
| MLLM | 10.27 | | |
|  | Base | Ours | Gap |
| ResNet18 | 53.83 | **60.07** | +6.24 |
| ResNet34 | 56.48 | **61.93** | +5.45 |
| MobileNet-V1 | 58.85 | **63.41** | +4.56 |
| EfficientNet | 66.04 | **69.32** | +3.28 |

**(f) FGVC-Aircraft**

|  | Zero-shot classification | | |
| --- | --- | --- | --- |
| MLLM | 11.94 | | |
|  | Base | Ours | Gap |
| ResNet18 | 71.76 | **74.33** | +2.57 |
| ResNet34 | 75.56 | **76.93** | +1.37 |
| MobileNet-V1 | 78.22 | **80.41** | +2.19 |
| EfficientNet | 84.16 | **84.88** | +0.72 |

Table 2: **Accuracy (%) on the coarse-grained image test set.** MLLM is InternVL2-8B. Base is the baseline using cross-entropy loss with class labels. We run each experiment three times and report the average results.

**(a) Caltech101**

|  | Zero-shot classification | | |
| --- | --- | --- | --- |
| MLLM | 85.52 | | |
|  | Base | Ours | Gap |
| ResNet18 | 73.35 | **75.77** | +2.42 |
| ResNet34 | 75.36 | **77.56** | +2.20 |
| MobileNet-V1 | 76.64 | **79.14** | +2.50 |
| EfficientNet | 80.05 | **82.17** | +2.12 |

**(b) Mini-ImageNet**

|  | Zero-shot classification | | |
| --- | --- | --- | --- |
| MLLM | 76.38 | | |
|  | Base | Ours | Gap |
| ResNet18 | 76.86 | **77.72** | +0.86 |
| ResNet34 | 77.47 | **78.65** | +1.18 |
| MobileNet-V1 | 77.50 | **78.84** | +1.34 |
| EfficientNet | 73.05 | **75.07** | +2.02 |

to check the quality and hallucination of the multi-aspect questions, we manually reviewed them and confirmed there was no hallucination.

**Extract logits of answers from MLLM.** According to Section 3.1, we extract the probability values of the yes token about multi-aspect from MLLM. We choose InternVL2-8B Chen et al. (2024) as our MLLM because InternVL2-8B can perform inference on a single NVIDIA RTX 3090 and has strong benchmark performance.

**Fine-grained image classification.** We use a total of six datasets: StanfordCars Krause et al. (2013), OxfordPets Parkhi et al. (2012), DTD Cimpoi et al. (2014), 102Flowers Nilsback & Zisserman (2008), CUB200 Wah et al. (2011), and FGVC-Aircraft Maji et al. (2013). For fine-grained image classification, we train all models for 240 epochs, with batch size 16. The initial learning rate is 0.01, divided by 10 at the 150th, 180th and 210th epoch. We use SGD optimizer with the momentum of 0.9, and weight decay is set to 5e-4.

**Coarse-grained image classification.** We additionally apply our method to the Caltech101 Fei-Fei et al. (2004) and Mini-ImageNet Ravi & Larochelle (2016) datasets for coarse-grained image classification. For Caltech101 Fei-Fei et al. (2004), we train all models for 240 epochs, with batch size 16. The initial learning rate is 0.01, divided by 10 at the 150th, 180th, and 210th epoch. For Mini-ImageNet Ravi & Larochelle (2016), we use the same settings following ImageNet setting of prior work Zhao et al. (2022); Guo et al. (2023b).

More implementation details are included in supplementary materials due to the space limit.

## 4.2 EXPERIMENTAL RESULTS

**Fine-grained image classification.** We mainly focus on fine-grained image classification task. Table 1 shows the experimental results on fine-grained datasets Krause et al. (2013); Parkhi et al. (2012); Cimpoi et al. (2014); Nilsback & Zisserman (2008); Wah et al. (2011); Maji et al. (2013). As shown in Table 1, our method demonstrates significant performance improvements for all models on all datasets compared with the model using cross-entropy loss with class labels. For example, on the StanfordCars dataset with ResNet18, our method shows a 5.85% higher performance compared to the baseline. This indicates that our model effectively transfers knowledge regarding aspects and can help models become more effective when dealing with datasets that have fine-grained features (such as subtle differences in visual appearance and patterns).

**Coarse-grained image classification.** Additionally, we experiment with our approach on coarse-grained datasets. Table 2 shows the experimental results on Caltech101 Fei-Fei et al. (2004) and Mini-ImageNet Ravi & Larochelle (2016). According to Table 2, our model improves the performance of all baselines. These results indicate that our model is also effective in coarse-grained image classification and demonstrate that transferring diverse knowledge to the model can help improve performance in image classification.

## 4.3 ABLATION STUDIES

**Effect of the loss function.** In Table 3 (a), we investigate the effect of the loss function by applying KL-divergence loss to the multi-aspect logit. The result shows that using binary-cross entropy loss achieves better performance. We assume that because the multi-aspect logits represent the probability of the yes token extracted from the MLLM, using binary-cross entropy loss would bring more improvement to the classification model.

**Effect of the multi-aspect logits.** In Table 3 (b), we validate the contribution of the multi-aspect logits to image classification by comparing our method to the one that replaces the logits with a random logit following a Gaussian distribution. As shown in Table 3 (b), our method with multi-aspect logits outperforms the method with random logits. These results demonstrate that the multi-aspect logits can enhance image classification performance by representing knowledge from various aspects for each class in the dataset.

**Weight to the multi-aspect knowledge distillation loss.** Table 3 (c) presents the performance of our method with different weights to the multi-aspect logit loss on StanfordCars and Caltech101. The x-axis represents the weights $\alpha$ (0 means the baselines), while the y-axis indicates the accuracy. Our method, based on $\alpha$, demonstrates improvements in the performances of all baseline models. Additionally, we empirically find that the performance decreases when $\alpha$ value reaches 50.

**Effect of LLM on multi-aspect question generation.** To assess the impact of different LLMs on multi-aspect question generation, we compare a model that generates multi-aspect questions using GPT-3.5 with our model that generates multi-aspect questions using GPT-4o. Both models utilize InternVL2-8B as the MLLM for logit extraction, with only the LLM for multi-aspect question generation being different. In Table 3 (d), Ours(L:GPT-3.5) using GPT-3.5 for generating multi-aspect questions outperforms the baselines and shows competitive results when compared to ours(which uses GPT-4o). These results demonstrate the robustness of our method to the performance of LLMs.

**Effect of MLLM on multi-aspect logit extraction.** We further investigate the impact of using different MLLMs on our method by using LLaVA-NeXT-34B Liu et al. (2024a), which has more parameters compared to InternVL2-8B Chen et al. (2024). As shown in Table 3 (d) with Ours (M: LLaVA), our method with LLaVA-NeXT-34B outperforms the baselines and shows competitive results when compared to InternVL2-8B. However, InternVL2-8B is more parameter efficient.

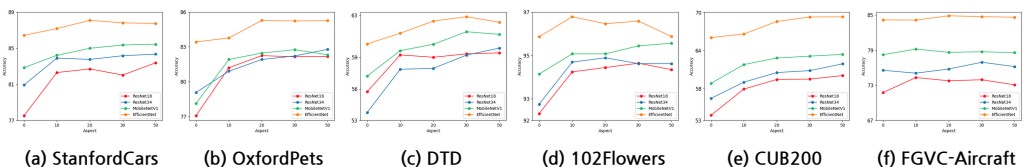

(a) StanfordCars     (b) OxfordPets     (c) DTD     (d) 102Flowers     (e) CUB200     (f) FGVC-Aircraft

Figure 2: **Ablation study on the number of multi-aspect questions.** The x-axis represents the number of aspects (0 means the baselines), while the y-axis indicates the accuracy. We run each experiment three times and report the average results.

Table 3: **Ablation study on each component.** Table (a), (b) and (d) report the accuracy (%) on StanfordCars Krause et al. (2013). Figure (c) shows different weights to the multi-aspect loss on StanfordCars and Caltech101. Res18 for ResNet18, Res34 for ResNet34, Mb-N1 for MobileNetV1 and EffiNet for EfficientNet. Rand for our method with random logits instead of multi-aspect logits. KL for our method with KL-Divergence loss on multi-aspect logit. $\alpha$ for the weighting factor of multi-aspect logit loss. We run each experiment three times and report the average results. We provide additional experimental results in the supplementary material.

**(a) Effect of the loss function**

|  | Res18 | Res34 | Mb-N1 | EffiNet |
|---|---|---|---|---|
| KL | 82.52 | 82.63 | 84.94 | 87.27 |
| Ours | **83.38** | **84.33** | **85.43** | **88.07** |

**(b) Effect of the multi-aspect logit**

|  | Res18 | Res34 | Mb-N1 | EffiNet |
|---|---|---|---|---|
| Rand | 79.36 | 81.04 | 83.39 | 86.65 |
| Ours | **83.38** | **84.33** | **85.43** | **88.07** |

**(c) Weights to the multi-aspect loss**

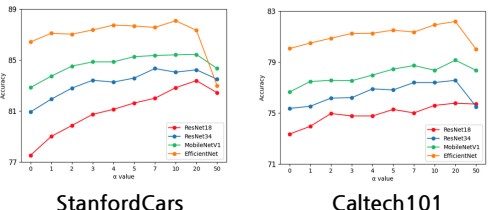

StanfordCars        Caltech101

**(d) Effect of LLM and MLLM**

|  | Res18 | Res34 | Mb-N1 | EffiNet |
|---|---|---|---|---|
| Base | 77.53 | 80.93 | 82.84 | 86.41 |
| Ours(L: GPT-3.5) | 82.46 | 83.65 | 85.25 | 87.38 |
| Ours(M: LLaVA) | **83.49** | **84.47** | 85.24 | 87.49 |
| Ours | 83.38 | 84.33 | **85.43** | **88.07** |

**Effect of the number of multi-aspect questions.** To evaluate the impact of the number of multi-aspect questions, we conduct experiments on different numbers of multi-aspect questions. First, we input the multi-aspect questions into the LLM, which ranks them based on the importance of each aspect. We then conduct experiments using the top 10, 20, 30, and 50 ranked questions in order. As shown in Figure 2, our method outperforms all baselines on all datasets and exhibit performance improvement based on the number of multi-aspect questions. This shows that multi-aspect questions can contribute to improving the performance of image classification.

### 4.4 EXTENSION OF OUR MODEL

To show the scalability of our approach, we apply our method to three tasks. First, we extend our model using traditional logit distillation. Second, we evaluate our model's performance when the dataset size is decreased. Finally, we extend our model to the object detection task.

**Extension to traditional knowledge distillation.** Since our model does not have the teacher classification model and the teacher model's class logits, it is different from traditional knowledge distillation (KD). However, since we distill the multi-aspect knowledge to be learned into logits, it simply can be integrated with existing logit distillation methods. We compare our method with KD on the StanfordCars Krause et al. (2013) and Caltech101 Fei-Fei et al. (2004). According to Table 6, the model extended with our method for KD outperforms the traditional KD approach. These results demonstrate that our approach can be effectively extended to traditional logit distillation.

**Extension to less training data.** We evaluate the performance of our model when trained with a reduced amount of training data. As shown in Table 5, our multi-aspect approach leads to greater performance improvement as the dataset size decreases. For example, on the StanfordCars dataset, ResNet18 shows a 24.01% performance improvement over the baseline when only 40% of the entire

Table 4: **Extension to class logit distillation with MLLM on Caltech101.** We run each experiment three times and report the average results.

| Teacher | MLLM (85.52) | |
|---|---|---|
| Student | Res18 | Res34 |
| Base | 73.35 | 75.36 |
| KD | 73.86 | 75.86 |
| Ours | **75.76** | **77.56** |

Table 5: **Extension to less training data.** Data represents the percentage of training data used, while the Gap indicates the gap in accuracy between the baseline and our method with ResNet18. Base is the baseline using cross-entropy loss with class labels.

| | StanfordCars | | | OxfordPets | | | Caltech101 | | |
|---|---|---|---|---|---|---|---|---|---|
| Data | Base | Ours | Gap | Base | Ours | Gap | Base | Ours | Gap |
| 40% | 25.74 | **49.75** | +24.01 | 50.71 | **58.45** | +7.74 | 57.74 | **61.30** | +3.56 |
| 60% | 54.78 | **69.49** | +14.71 | 64.21 | **71.26** | +7.05 | 64.70 | **67.77** | +3.07 |
| 80% | 69.72 | **78.04** | +8.32 | 72.33 | **78.41** | +6.08 | 68.84 | **72.35** | +3.51 |
| 100% | 77.53 | **83.38** | +5.85 | 77.07 | **82.24** | +5.17 | 73.35 | **75.77** | +2.42 |

Table 6: **Extension to traditional knowledge distillation on StanfordCars and Caltech101.** We can simply extend our method to traditional logit distillation. We run each experiment three times and report the average results.

| Dataset | Teacher | Res34(80.93) | EffiNet(86.41) |
|---|---|---|---|
| | Student | Res18(77.53) | Mb-N1(82.84) |
| Stanford | KD | 79.62 | 85.11 |
| Cars | Ours + KD | **83.44** | **86.34** |
| Dataset | Teacher | Res34(75.36) | EffiNet(80.05) |
| | Student | Res18(73.35) | Mb-N1(76.64) |
| Caltech | KD | 74.53 | 78.71 |
| 101 | Ours + KD | **76.70** | **79.70** |

Table 7: **Extension to object detection on MS-COCO based on Faster-RCNN Ren et al. (2016)-FPN Lin et al. (2017).** AP evaluated on val2017. We run each experiment three times and report the average results.

| | | AP | $AP_{50}$ | $AP_{75}$ |
|---|---|---|---|---|
| Mb-N2 | Base | 29.42 | 49.07 | 30.72 |
| | Ours | **29.65** | **49.49** | **31.02** |
| Res18 | Base | 33.18 | 53.54 | 35.31 |
| | Ours | **33.35** | **53.90** | **35.58** |
| Res50 | Base | 38.06 | 58.95 | 41.22 |
| | Ours | **38.27** | **59.30** | **41.67** |

training dataset was used. It demonstrates the potential for broader applicability in fine-grained tasks and real-world applications with limited training datasets.

**Extension to object detection.** To evaluate the scalability of our method, we evaluate the performance on object detection tasks with MS-COCO datasets. Following Zhao et al. (2022), we add features to the backbone network of Faster R-CNN Ren et al. (2016)-FPN Lin et al. (2017) and apply a multi-aspect logit loss with the number of multi-aspect questions set to 50. As shown in Table 7, our method further improves the performances of the baselines. These results show that we can effectively identifying objects in the image by learning deep visual feature from multi-aspect knowledge and may have a potential to contribute to various visual understanding tasks.

## 5 ANALYSES

### 5.1 DISTILLATION WITH MLLM ZERO-SHOT CLASSIFICATION LOGITS

According to Table 1, the MLLM shows poor zero-shot image classification performance on fine-grained datasets. These results show that they may struggle with classifying highly specific information, such as distinguishing between Yellow headed Blackbird and Eastern Towhee in the CUB200 Wah et al. (2011) dataset. Therefore, we cannot directly distill the class logits from MLLM. To leverage the features of MLLM that can understand and infer abstract and complex information, we distill knowledge through multi-aspect questions based on diverse insights and understanding beyond class labels. This shows the potential of our approach to be applied to other tasks, regardless of the performance of MLLM in specific domains.

In coarse-grained image datasets, we find that MLLM performs better than on fine-grained datasets. We assume that this is because MLLM was trained on a very large dataset, enabling it to perform general classification tasks. Since the zero-shot classification performance of MLLM on Caltech101 is better than the baseline, we may apply traditional knowledge distillation (KD) using MLLM's class logits as the teacher logits on Caltech101. According to Table 4, using MLLM's logits as a teacher result in a slight performance improvement over the baseline, but it underperforms compared to our method. Additionally, when applying our approach to coarse-grained image dataset, it improve the performance of all models over the baselines, as shown in Table 2. This shows that not only for fine-grained but also for coarse-grained tasks, it is important to consider multi-aspects rather than directly distilling the logits of MLLM, demonstrating that our approach is more effective.

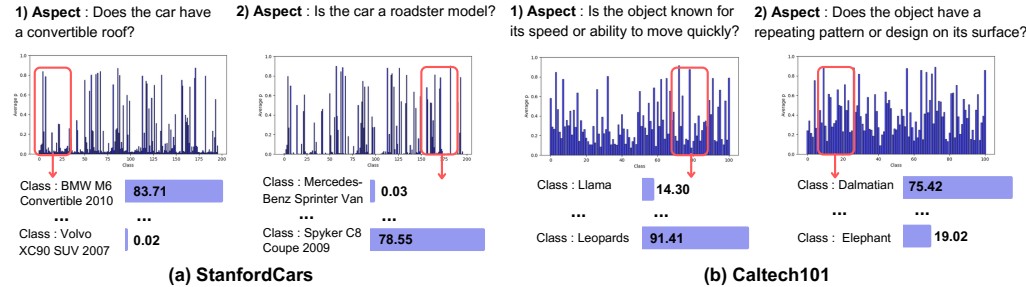

Figure 3: **Visualization of the average logit distribution for classes related to aspects.** The x-axis represents the classes, and the y-axis represents the mean of the aspect probability distribution from MLLMs in the dataset. The class names corresponding to the indices in x-axis are provided in the supplementary material due to space.

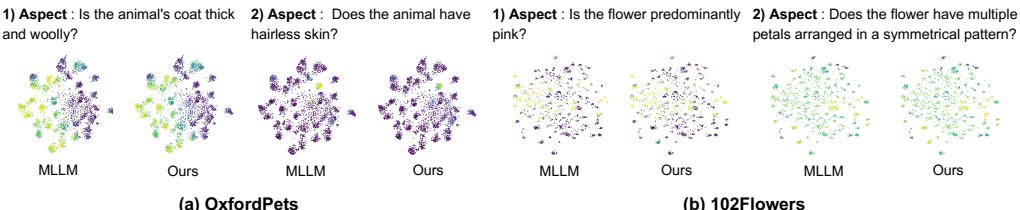

Figure 4: **Visualization of t-SNE embeddings for the datasets by aspects.** Ours is t-SNE visualizations of the aspect logits from our model (ResNet18), while MLLM is t-SNE visualizations of the aspect logits from the MLLM (InternVL2-8B). The yellow points indicate that the probability of "yes" is close to 1, and the purple points indicate that the probability of "yes" is close to 0.

### 5.2 ANALYSIS OF MULTI-ASPECT QUESTIONS GENERATED BY THE LLM

To analyze the effectiveness of the multi-aspect questions generated by the LLM in image classification, we present a histogram of the average MLLM probability values of aspects for each class, as shown in Figure 3. For example, as shown in Figure 3 (a)-1, the class "BMW M6 Convertible 2010" on StanfordCars Krause et al. (2013) has a high probability value for the aspect "Does the car have a convertible roof?". We observe that classes possessing the features of the aspect exhibit high probabilities, while those lacking the features show low probabilities.

Furthermore, the aspects of the StanfordCars, which have fine-grained features as shown in Figure 3 (a)-2, include specific questions about car features such as "Is the car a roadster model?". These results demonstrate that our multi-aspect questions effectively represent the various features of the dataset, including visual specifics and understanding, and can help classify images.

### 5.3 ANALYSIS OF THE DISCRIMINABILITY USING THE ASPECT LOGITS

To analyze the knowledge transfer across various aspects from the MLLM to the image classification model, we use t-SNE visualizations of the logits from both our model and the MLLM on these aspects, as illustrated in Figure 4. The yellow points indicate that the probability of "yes" is close to 1, and the purple points indicate that the probability of "yes" is close to 0. As shown in Figure 4, our model demonstrates that the aspect logits of our model exhibit a similar trend to the aspect logits of the MLLM in both fine-grained datasets and coarse-grained datasets. These results indicate that our method can effectively distill various knowledge about the dataset by utilizing the multi-aspect logits extracted from the MLLM.

### 5.4 ANALYSIS OF MULTI-ASPECT CLASSIFICATION OF OUR MODEL

To analyze the classification performance of our model for multi-aspect questions, we compare the probability values of our model with those of the MLLM for multi-aspect questions. As shown in Figure 5 (c), when an image of a Birman is given as input, our model outputs a probability value of 86.97 for the visual aspect "Does the animal have striking blue eyes?" and a value of 11.74 for the

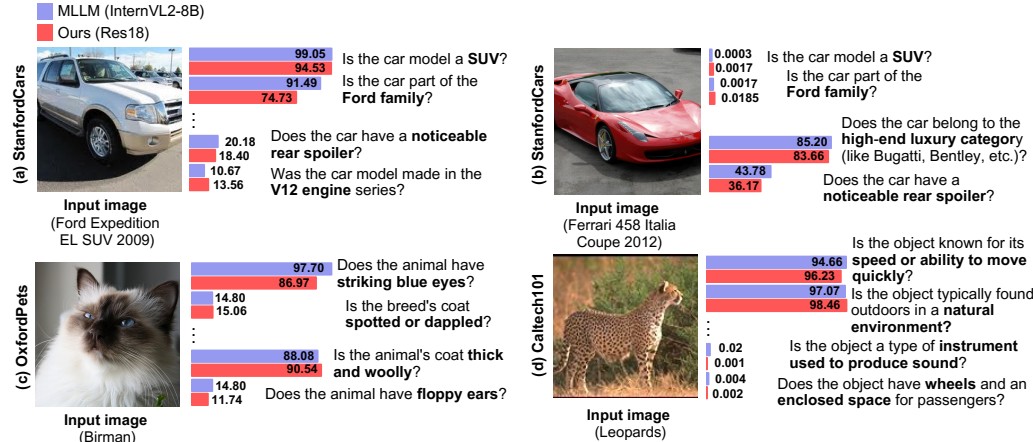

Figure 5: **Comparison of probability values for multi-aspect questions.** We compare the probability values of our model with those of the MLLM for multi-aspect questions. Our model shows similar probability values to MLLM across various multi-aspect questions.

aspect "Does the animal have floppy ears?", similar to the MLLM. These results indicate that our model effectively distill visual aspects and understands visual aspects.

Furthermore, as shown in Figure 5 (d), when an image of a Leopards is given as input, our model outputs a probability value of 96.23 for the aspect "Is the object known for its speed or ability to move quickly?" and a value of 98.46 for the aspect "Is the object typically found outdoors in a natural environment?" which are not visual aspect but abstract or require a deeper understanding of the image, similar to the MLLM.

These results suggest that the model can distill not only visual knowledge but also abstract and complex knowledge about multi-aspect knowledge.

## 5.5 TRAINING TIME AND COMPUTATIONAL COST

As we extract logits from MLLMs, this can require more computational resources compared to training only image classification models. However, since we query the MLLM about aspects in a zero-shot manner, there is no need to train the MLLM. Moreover, we utilize InternVL2-8B Chen et al. (2024) for logit extraction, which allows aspect extraction using a single NVIDIA RTX 3090. The number of parameters in our model is approximately 11.25M when using ResNet18 with 50 aspects, with the baseline also having 11.23M parameters. For StanfordCars, the training time for the baseline model is 25.42 seconds per epoch, while our model takes 27.90 seconds per epoch. In terms of inference time, our model takes 22.80 seconds, compared to the baseline's 20.59 seconds, showing slight increase. More information with different models and datasets is included in supplementary material.

## 6 CONCLUSION AND LIMITATION

In this paper, we propose a novel multi-aspect knowledge distillation method leveraging MLLM along with analyses. Unlike previous image classification methods, our method leverages MLLM to distill multi-aspect knowledge that require complex and deeper understanding beyond the class labels. Our experimental results demonstrate that the proposed method outperforms baseline models in both fine-grained and course-grained image classification tasks. Additionally, we extend our method to other tasks such as object detection, and it outperforms the baselines. Our findings provide a novel view by simply distilling multi-aspect knowledge and demonstrate the potential of our method to be applied to a variety of tasks. However, as a limitation, our approach is constrained by the necessity of pre-trained LLMs and MLLMs to generate aspects and logits used for model training. In future work, we will explore applying our method to other domains, such as image generation and image captioning.

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

APPENDIX

# A   THE TRAINING CURVE GRAPH OF LOSS

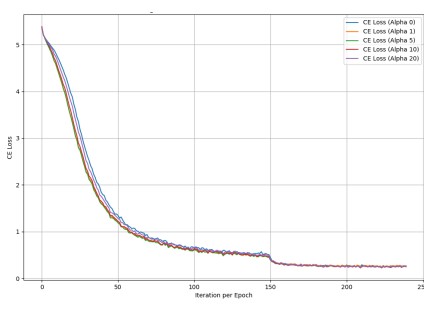 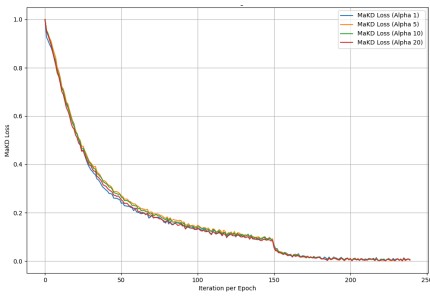

(a) Training Curve Graph of Cross Entropy Loss    (b) Normalized Training Curve Graph of MaKD Loss

Figure 6: **The training curve graph of loss with the number of iterations.** We provide two types of training curve losses. (a) cross-entropy loss and (b) our proposed MaKD loss. When applying our method, it demonstrates a lower loss trend compared to the baseline's cross-entropy loss.

