## SUPPLEMENTARY MATERIALS

**Note:** We provide detailed figures for multi-aspect logit distillation in Section A, implementation and dataset details in Section B, additional ablation study results in Section C, further details on the visualization of the logit distribution in Section D, and computational cost analyses in Section E, which were not included in the main paper due to space limitations.

## A  DETAILS OF MULTI-ASPECT LOGIT DISTILLATION

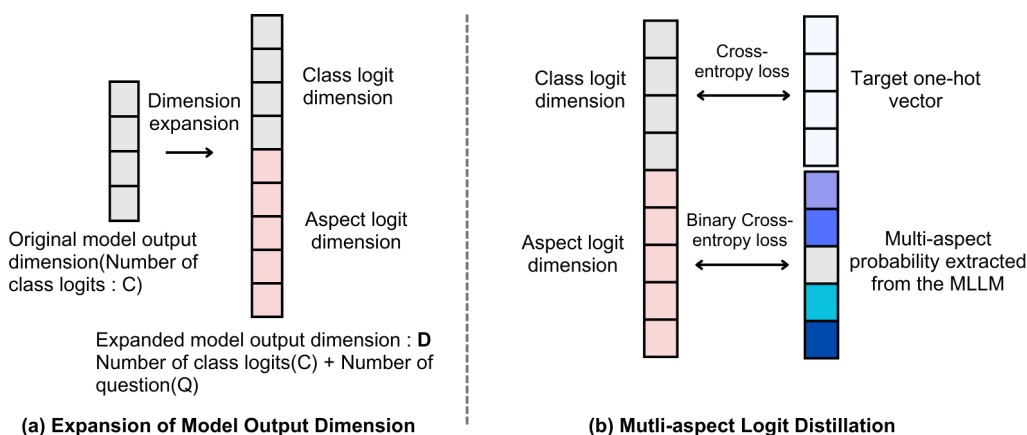

**(a) Expansion of Model Output Dimension**    **(b) Mutli-aspect Logit Distillation**

Figure 1: **Multi-aspect knowledge distillation.** To distill knowledge about multi-aspect questions into the model, we simply expand the dimension of model output. Also, we consider the expanded dimensions as the class logit dimension and the aspect logit dimension. We apply cross-entropy loss to the class logit dimension and binary cross-entropy loss to the aspect logit dimension.

## B  IMPLEMENTATION DETAILS

### B.1  DATASET DETAILS

**StanfordCars  Krause et al. (2013).** StanfordCars contains 16,185 images of 196 classes of cars. The data is split into 8,144 training images and 8,041 testing images, where each class has been split roughly in a 50-50 split. Classes are typically at the level of Make, Model, Year, ex. 2012 Tesla Model S or 2012 BMW M3 coupe.

**OxfordPets  Parkhi et al. (2012).** OxfordPets comprises 7,384 images of 37 distinct cat and dog breeds, with around 200 images per class. It is divided into 3,690 images for training and 3,694 images for testing. The dataset features significant variations in scale, pose, lighting, and others.

**Describable Textures Dataset (DTD)  Cimpoi et al. (2014).** DTD consists of 47 texture classes and a total of 5,640 images. It is divided into 3,760 images for training and 1,880 for testing, with each class containing 120 images. The image sizes range from 300x300 to 640x640 pixels, and each image contains at least 90% of the surface area representing the category's attribute.

**102Flowers  Nilsback & Zisserman (2008).** 102Flowers is designed for image classification, featuring 102 different flower classes. It is divided into 6,552 training images and 1,637 testing images. Each class includes between 40 and 258 images, with significant variations in scale, pose, and lighting conditions across the images.

**CUB200  Wah et al. (2011).** CUB200 is one of the most commonly used datasets for fine-grained visual categorization tasks. It comprises 11,788 images across 200 bird subcategories, with 5,994 images for training and 5,794 for testing. Each image has detailed annotations, including 1 subcategory label, 15 part locations, 312 binary attributes, and 1 bounding box.

**FGVC-Aircraft Maji et al. (2013).** FGVC-Aircraft consists of 9,967 aircraft images, with around 100 images corresponding to each of the 100 different model variants, the majority being airplanes. The dataset is divided into 6,667 images for training and 3,300 for testing. Each image includes annotations with a tight bounding box and a hierarchical label for the airplane model. The aircraft models are arranged in a four-level hierarchical structure.

**Caltech101 Fei-Fei et al. (2004).** Caltech101 includes images from 101 object categories, along with a background category consisting of images unrelated to those 101 categories. To focus purely on class classification, we exclude the background category. The dataset is divided into 4,310 images for training and 4,367 images for testing. Each category contains between 40 and 800 images, with most classes having approximately 50 images. The image resolution is roughly 300×200 pixels.

**Mini-ImageNet Ravi & Larochelle (2016).** Mini-ImageNet is a reduced version of the larger ImageNet Deng et al. (2009) dataset, specifically designed for few-shot learning tasks. It consists of 50,000 training images and 10,000 testing images distributed across 100 classes. Additionally, to use a higher resolution, we utilize the dataset from Ravi & Larochelle (2016).

**Microsoft Common Objects in Context (MS-COCO) Lin et al. (2014).** MS-COCO is a large-scale object detection, segmentation, key-point detection, and captioning dataset. The dataset consists of 328K images. We use the MS-COCO dataset's 2017 version, which consists of a training/validation split of 118K/5K images.

## B.2   TRAINING DETAILS

For the image classification experiments, we employed baseline models such as ResNet18, ResNet34, MobileNet-V1, and EfficientNet-b0 across various fine-grained datasets, including StanfordCars, OxfordPets, DTD, 102Flowers, CUB200, and FGVC-Aircraft, as well as coarse-grained datasets such as Caltech101 and Mini-ImageNet.

**Data preprocessing.** Input images were normalized using the channel-wise mean (0.485, 0.456, 0.406) and standard deviation (0.229, 0.224, 0.225) for RGB channels. For training, we applied a series of transformations: RandomResizedCrop with a target size of 224, followed by RandomHorizontalFlip, conversion to tensor using ToTensor, and normalization.

**Hyperparameters for fine-grained datasets.** The models were trained for 240 epochs with a batch size of 16. The initial learning rate was set to 0.01 and decreased by a factor of 10 at the 150th, 180th, and 210th epochs. We use the SGD optimizer with a momentum of 0.9 for all experiments, and weight decay is set to 5e-4.

**Hyperparameters for Caltech101 dataset.** The models were trained for 240 epochs with a batch size of 16. The initial learning rate was set to 0.01 and decreased by a factor of 10 at the 150th, 180th, and 210th epochs. We use the SGD optimizer with a momentum of 0.9 for all experiments, and weight decay is set to 5e-4.

**Hyperparameters for Mini-ImageNet dataset.** The models were trained for 100 epochs with a batch size of 64. The initial learning rate was set to 0.2 and decreased by a factor of 10 at the 30th, 60th, and 90th epochs. We use the SGD optimizer with a momentum of 0.9 for all experiments, and weight decay is set to 5e-4.

## B.3   MULTI-ASPECT QUESTION SAMPLES

Table 8, 9, 10, 11, 12, 13, 14, and 15 present the multi-aspect questions generated by GPT-4o for the StanfordCars, OxfordPets, DTD, 102Flowers, CUB200, FGVC-Aircraft, Caltech101, and Mini-ImageNet datasets, respectively. Meanwhile, Table 16, 17, and 18 show the multi-aspect questions generated by GPT-3.5-turbo for the StanfordCars, OxfordPets, and Caltech101 datasets, respectively.

## B.4   DETAILS OF LOGIT DISTILLATION WITH MLLM

We also include how MLLM distills logit to student model, as described in Section 4.4 and 5.1 of the main paper.

In traditional knowledge distillation, the teacher model typically outputs soft targets as a probability distribution over classes. The student model is then trained based on the KL divergence loss between the soft targets and the student's predicted target logits, as well as the cross-entropy loss with the actual hard targets. To enable MLLM to perform logit distillation, we make the following assumption:

'*Could the logits generated for both the predicted class index token and the remaining class index token logits during zero-shot classification be considered soft targets?*'

In this assumption, since MLLM receives information about the range of possible answers through prompts, it restricts the range of tokens generated. Given a first logit vector $\mathbf{z} \in \mathbb{R}^V$, where $V$ is the vocabulary size of tokenizer, the logits corresponding to the numerical tokens are indexed by the set $\mathcal{N} \subseteq \{1, 2, \ldots, C\}$. The softmax function applied to the logits of the numerical tokens within a specified range $[1, C]$ is given by:

$$P(t \mid 1 \leq t \leq C) = \frac{\exp(z_t)}{\sum_{n \in \mathcal{N}, 1 \leq n \leq C} \exp(z_n)} \quad \text{for} \quad t \in \mathcal{N}, 1 \leq t \leq C$$

where $z_t$ is the logit corresponding to token $t$ and the sum in the denominator is computed over all numerical tokens $n$ in the range $[1, C]$. This can be interpreted as MLLM producing a probability distribution for classifying specific classes, allowing it to generate soft targets as a teacher model. These generated soft targets can be used for training in the same way as in traditional knowledge distillation, as they remain unchanged while the student model is being trained.

## C   ADDITIONAL ABLATION STUDY RESULTS

In this section, we provide the additional ablation results for the OxfordPets fine-grained dataset and Caltech101 coarse-grained dataset. The additional ablation study results are presented in Table 1 and 2, showing the outcomes for the OxfordPets and Caltech101 datasets, respectively. Table 3 displays the results of the extension to logit distillation on the OxfordPets dataset.

Table 1: **Additional Ablation study on OxfordPets.** Each table reports the accuracy(%) on OxfordPets. Res18 for ResNet18, Res34 for ResNet34, Mb-N1 for MobileNetV1 and EffiNet for EfficientNet-b0. Rand for our method with random logits instead of multi-aspect logits. KL for our method with KL-Divergence loss on multi-aspect logit. $\alpha$ for the weighting factor of multi-aspect logit loss. We run each experiment 3 times and report the average results.

**(a) Effect of the loss function**

|  | Res18 | Res34 | Mb-N1 | EffiNet |
|---|---|---|---|---|
| KL | 75.96 | 79.52 | 79.71 | 83.92 |
| Ours | **82.24** | **82.78** | **82.75** | **85.27** |

**(b) Effect of the multi-aspect logit**

|  | Res18 | Res34 | Mb-N1 | EffiNet |
|---|---|---|---|---|
| Rand | 78.64 | 79.17 | 77.70 | 83.23 |
| Ours | **82.24** | **82.78** | **82.75** | **85.27** |

**(c) Weights to the multi-aspect loss**

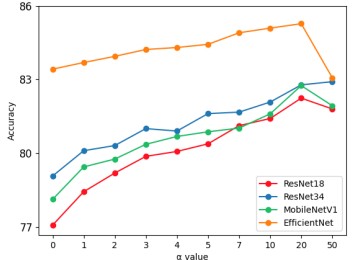

**(d) Effect of LLM and MLLM**

|  | Res18 | Res34 | Mb-N1 | EffiNet |
|---|---|---|---|---|
| Base | 77.07 | 79.07 | 78.12 | 83.42 |
| Ours(L: GPT-3.5) | 82.72 | **83.08** | 82.66 | 85.22 |
| Ours(M: LLaVA) | **82.94** | 83.04 | **82.94** | 85.19 |
| Ours | 82.24 | 82.78 | 82.75 | **85.27** |

## D   ADDITIONAL DETAILS OF VISUALIZATION (HISTOGRAM, T-SNE AND ERROR BAR)

In this section, we provide additional visualizations for our analysis, including error bars for the experimental results on each dataset, visualization of the average logit distribution, visualization of

Table 2: **Additional Ablation study on Caltech101.** Each table reports the accuracy(%) on Caltech101. Res18 for ResNet18, Res34 for ResNet34, Mb-N1 for MobileNetV1 and EffiNet for EfficientNet-b0. Rand for our method with random logits instead of multi-aspect logits. KL for our method with KL-Divergence loss on multi-aspect logit. $\alpha$ for the weighting factor of multi-aspect logit loss. We run each experiment 3 times and report the average results.

**(a) Effect of the loss function**

|      | Res18   | Res34   | Mb-N1   | EffiNet |
|------|---------|---------|---------|---------|
| KL   | 75.10   | 74.74   | 77.67   | 80.73   |
| Ours | **75.77** | **77.56** | **79.14** | **82.17** |

**(b) Effect of the multi-aspect logit**

|      | Res18   | Res34   | Mb-N1   | EffiNet |
|------|---------|---------|---------|---------|
| Rand | 73.90   | 74.94   | 76.64   | 79.80   |
| Ours | **75.77** | **77.56** | **79.14** | **82.17** |

**(c) Weights to the multi-aspect loss**

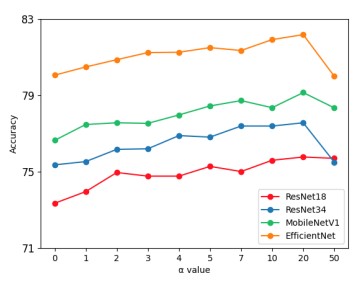

**(d) Effect of LLM and MLLM**

|                  | Res18   | Res34   | Mb-N1   | EffiNet |
|------------------|---------|---------|---------|---------|
| Base             | 73.35   | 75.36   | 76.64   | 80.05   |
| Ours(L: GPT-3.5) | **76.16** | 76.78 | 79.13   | 81.95   |
| Ours(M: LLaVA)   | 76.02   | 77.46   | 78.81   | 81.65   |
| Ours             | 75.77   | **77.56** | **79.14** | **82.17** |

Table 3: **Extension to logit distillation on OxfordPets.** We can simply extend our method to logit distillation. We run each experiment three times and report the average results.

| Dataset   |         | Teacher
Student | ResNet34(79.07)
ResNet18(77.07) | EfficientNet-b0(83.42)
MobileNetV1(78.12) |
|-----------|---------|--------------------|------------------------------------|-----------------------------------------------|
| OxfordPets | KD     |                    | 79.01                              | 80.90                                         |
|           | Ours + KD |                 | **82.68**                          | **83.13**                                      |

t-SNE embedding, and visualizations of the probability values between the MLLM and the classification model for the multi-aspect questions.

**Error bar.** To help in evaluating the quality of the experiments, we include error bars representing the standard error for the conducted experiments. The error bars for the StanfordCars, Oxford-Pets, DTD, 102Flowers, CUB200, FGVC-Aircraft, Caltech101, and Mini-ImageNet datasets are presented in Figure 2. We run each experiment 3 times and report the average results.

**Visualization of the average logit distribution.** We provide the average logit distribution for all aspects of the datasets. The visualizations of the average logit distribution graphs for the Stanford-Cars, OxfordPets, DTD, 102Flowers, CUB200, FGVC-Aircraft, Caltech101, and Mini-ImageNet datasets are shown in Figure 3, 4, 5, 6, 7, 8, 9, and 10, respectively.

**Visualization of t-SNE embeddings.** We use t-SNE to reduce the dimensionality of the predicted aspect logit probabilities from our ResNet18 and the MLLM's aspect logit probabilities to better visualize the results. For each dataset, we display the train and test results across 50 aspects. Yellow points represent a higher probability of 'Yes' (closer to 1), while purple points represent a higher probability of 'Yes' (closer to 0). The ground-truth and predicted result t-SNE embedding visualizations for the training data from the StanfordCars, OxfordPets, DTD, 102Flowers, CUB200, FGVC-Aircraft, Caltech101, and Mini-ImageNet datasets are shown in Figure 11, 12, 13, 14, 15, 16, 17, 18, 19, 20, 21, 22, 23, 24, 25, and 26.

# E  DETAILS OF COMPUTATIONAL COSTS

We calculate the computational cost on StanfordCars and OxfordPets using a single NVIDIA RTX 3090 GPU. Training refers to the average of three experiments and represents the number of seconds

taken per epoch. Inference indicates the number of seconds required to process all test sets. Table 4 shows that even with the extension of our method, there is no significant difference in time.

| StanfordCars | | | | | | |
|---|---|---|---|---|---|---|
| | **ResNet18** | | | **ResNet34** | | |
| **Aspect** | **Training** | **Inference** | **FLOPs** | **Training** | **Inference** | **FLOPs** |
| 0 | 25.418 | 20.5869 | 58.1873G | 60.416 | 51.6867 | 117.4380G |
| 10 | 25.916 | 21.0294 | 58.1875G | 60.639 | 51.8848 | 117.4382G |
| 20 | 26.411 | 21.4692 | 58.1877G | 60.861 | 52.0821 | 117.4384G |
| 30 | 26.910 | 21.9126 | 58.1878G | 61.233 | 52.4127 | 117.4386G |
| 50 | 27.903 | 22.7951 | 58.1882G | 61.529 | 52.6757 | 117.4389G |
| OxfordPets | | | | | | |
| | **ResNet18** | | | **ResNet34** | | |
| **Aspect** | **Training** | **Inference** | **FLOPs** | **Training** | **Inference** | **FLOPs** |
| 0 | 11.522 | 9.3810 | 24.0335G | 16.169 | 12.5678 | 48.5060G |
| 10 | 11.772 | 9.6062 | 24.0337G | 16.520 | 12.8841 | 48.5062G |
| 20 | 12.022 | 9.8315 | 24.0338G | 16.871 | 13.2003 | 48.5064G |
| 30 | 12.311 | 9.0919 | 24.0340G | 17.277 | 13.5661 | 48.5066G |
| 50 | 12.825 | 10.5550 | 24.0344G | 17.998 | 14.2157 | 48.5069G |

Table 4: **The computational cost.**

Table 5: **Dataset class indices.** We provide the class indices for DTD, 102Flowers, and FGVC-Aircraft, which have 47, 102, and 100 classes, respectively.

| **Index** | **DTD Class** | **102Flowers Class** | **FGVC-Aircraft Class** |
|---|---|---|---|
| 0 | banded | alpine_sea_holly | 707-320 |
| 1 | blotchy | anthurium | 727-200 |
| 2 | braided | artichoke | 737-200 |
| 3 | bubbly | azalea | 737-300 |
| 4 | bumpy | ball_moss | 737-400 |
| 5 | chequered | balloon_flower | 737-500 |
| 6 | cobwebbed | barbeton_daisy | 737-600 |
| 7 | cracked | bearded_iris | 737-700 |
| 8 | crosshatched | bee_balm | 737-800 |
| 9 | crystalline | bird_of_paradise | 737-900 |
| 10 | dotted | bishop_of_llandaff | 747-100 |
| 11 | fibrous | black-eyed_susan | 747-200 |
| 12 | flecked | blackberry_lily | 747-300 |
| 13 | freckled | blanket_flower | 747-400 |
| 14 | frilly | bolero_deep_blue | 757-200 |
| 15 | gauzy | bougainvillea | 757-300 |
| 16 | grid | bromelia | 767-200 |
| 17 | grooved | buttercup | 767-300 |
| 18 | honeycombed | californian_poppy | 767-400 |
| 19 | interlaced | camellia | 777-200 |
| 20 | knitted | canna_lily | 777-300 |
| 21 | lacelike | canterbury_bells | A300B4 |
| 22 | lined | cape_flower | A310 |
| 23 | marbled | carnation | A318 |
| 24 | matted | cautleya_spicata | A319 |
| 25 | meshed | clematis | A320 |
| 26 | paisley | coltsfoot | A321 |
| 27 | perforated | columbine | A330-200 |
| 28 | pitted | common_dandelion | A330-300 |
| 29 | pleated | corn_poppy | A340-200 |
| 30 | polka-dotted | cyclamen | A340-300 |
| 31 | porous | daffodil | A340-500 |

| Index | DTD Class | 102Flowers Class | FGVC-Aircraft Class |
|-------|-----------|------------------|---------------------|
| 32 | potholed | desert-rose | A340-600 |
| 33 | scaly | english_marigold | A380 |
| 34 | smeared | fire_lily | ATR-42 |
| 35 | spiralled | foxglove | ATR-72 |
| 36 | sprinkled | frangipani | An-12 |
| 37 | stained | fritillary | BAE-125 |
| 38 | stratified | garden_phlox | BAE_146-200 |
| 39 | striped | gaura | BAE_146-300 |
| 40 | studded | gazania | Beechcraft_1900 |
| 41 | swirly | geranium | Boeing_717 |
| 42 | veined | giant_white_arum_lily | C-130 |
| 43 | waffled | globe-flower | C-47 |
| 44 | woven | globe_thistle | CRJ-200 |
| 45 | wrinkled | grape_hyacinth | CRJ-700 |
| 46 | zigzagged | great_masterwort | CRJ-900 |
| 47 | | hard-leaved_pocket_orchid | Cessna_172 |
| 48 | | hibiscus | Cessna_208 |
| 49 | | hippeastrum | Cessna_525 |
| 50 | | japanese_anemone | Cessna_560 |
| 51 | | king_protea | Challenger_600 |
| 52 | | lenten_rose | DC-10 |
| 53 | | lotus_lotus | DC-3 |
| 54 | | love_in_the_mist | DC-6 |
| 55 | | magnolia | DC-8 |
| 56 | | mallow | DC-9-30 |
| 57 | | marigold | DH-82 |
| 58 | | mexican_aster | DHC-1 |
| 59 | | mexican_petunia | DHC-6 |
| 60 | | monkshood | DHC-8-100 |
| 61 | | moon_orchid | DHC-8-300 |
| 62 | | morning_glory | DR-400 |
| 63 | | orange_dahlia | Dornier_328 |
| 64 | | osteospermum | E-170 |
| 65 | | oxeye_daisy | E-190 |
| 66 | | passion_flower | E-195 |
| 67 | | pelargonium | EMB-120 |
| 68 | | peruvian_lily | ERJ_135 |
| 69 | | petunia | ERJ_145 |
| 70 | | pincushion_flower | Embraer_Legacy_600 |
| 71 | | pink-yellow_dahlia | Eurofighter_Typhoon |
| 72 | | pink_primrose | F-16A |
| 73 | | poinsettia | FA-18 |
| 74 | | primula | Falcon_2000 |
| 75 | | prince_of_wales_feathers | Falcon_900 |
| 76 | | purple_coneflower | Fokker_100 |
| 77 | | red_ginger | Fokker_50 |
| 78 | | rose | Fokker_70 |
| 79 | | ruby-lipped_cattleya | Global_Express |
| 80 | | siam_tulip | Gulfstream_IV |
| 81 | | silverbush | Gulfstream_V |
| 82 | | snapdragon | Hawk_T1 |
| 83 | | spear_thistle | Il-76 |
| 84 | | spring_crocus | L-1011 |
| 85 | | stemless_gentian | MD-11 |
| 86 | | sunflower | MD-80 |
| 87 | | sweet_pea | MD-87 |
| 88 | | sweet_william | MD-90 |
| 89 | | sword_lily | Metroliner |

| Index | DTD Class | 102Flowers Class | FGVC-Aircraft Class |
|-------|-----------|------------------|---------------------|
| 90 | | thorn_apple | Model_B200 |
| 91 | | tiger_lily | PA-28 |
| 92 | | toad_lily | SR-20 |
| 93 | | tree_mallow | Saab_2000 |
| 94 | | tree_poppy | Saab_340 |
| 95 | | trumpet_creeper | Spitfire |
| 96 | | wallflower | Tornado |
| 97 | | water_lily | Tu-134 |
| 98 | | watercress | Tu-154 |
| 99 | | wild_pansy | Yak-42 |
| 100 | | windflower | |
| 101 | | yellow_iris | |

Table 6: **Dataset class indices.** We provide the class indices for StanfordCars, and CUB200, which have 196, and 200 classes, respectively.

| Index | StanfordCars Class | CUB200 Class |
|-------|--------------------|--------------|
| 0 | AM General Hummer SUV 2000 | Acadian_Flycatcher |
| 1 | Acura Integra Type R 2001 | American_Crow |
| 2 | Acura RL Sedan 2012 | American_Goldfinch |
| 3 | Acura TL Sedan 2012 | American_Pipit |
| 4 | Acura TL Type-S 2008 | American_Redstart |
| 5 | Acura TSX Sedan 2012 | American_Three_toed_Woodpecker |
| 6 | Acura ZDX Hatchback 2012 | Anna_Hummingbird |
| 7 | Aston Martin V8 Vantage Convertible 2012 | Artic_Tern |
| 8 | Aston Martin V8 Vantage Coupe 2012 | Baird_Sparrow |
| 9 | Aston Martin Virage Convertible 2012 | Baltimore_Oriole |
| 10 | Aston Martin Virage Coupe 2012 | Bank_Swallow |
| 11 | Audi 100 Sedan 1994 | Barn_Swallow |
| 12 | Audi 100 Wagon 1994 | Bay_breasted_Warbler |
| 13 | Audi A5 Coupe 2012 | Belted_Kingfisher |
| 14 | Audi R8 Coupe 2012 | Bewick_Wren |
| 15 | Audi RS 4 Convertible 2008 | Black_Tern |
| 16 | Audi S4 Sedan 2007 | Black_and_white_Warbler |
| 17 | Audi S4 Sedan 2012 | Black_billed_Cuckoo |
| 18 | Audi S5 Convertible 2012 | Black_capped_Vireo |
| 19 | Audi S5 Coupe 2012 | Black_footed_Albatross |
| 20 | Audi S6 Sedan 2011 | Black_throated_Blue_Warbler |
| 21 | Audi TT Hatchback 2011 | Black_throated_Sparrow |
| 22 | Audi TT RS Coupe 2012 | Blue_Grosbeak |
| 23 | Audi TTS Coupe 2012 | Blue_Jay |
| 24 | Audi V8 Sedan 1994 | Blue_headed_Vireo |
| 25 | BMW 1 Series Convertible 2012 | Blue_winged_Warbler |
| 26 | BMW 1 Series Coupe 2012 | Boat_tailed_Grackle |
| 27 | BMW 3 Series Sedan 2012 | Bobolink |
| 28 | BMW 3 Series Wagon 2012 | Bohemian_Waxwing |
| 29 | BMW 6 Series Convertible 2007 | Brandt_Cormorant |
| 30 | BMW ActiveHybrid 5 Sedan 2012 | Brewer_Blackbird |
| 31 | BMW M3 Coupe 2012 | Brewer_Sparrow |
| 32 | BMW M5 Sedan 2010 | Bronzed_Cowbird |
| 33 | BMW M6 Convertible 2010 | Brown_Creeper |
| 34 | BMW X3 SUV 2012 | Brown_Pelican |
| 35 | BMW X5 SUV 2007 | Brown_Thrasher |
| 36 | BMW X6 SUV 2012 | Cactus_Wren |
| 37 | BMW Z4 Convertible 2012 | California_Gull |
| 38 | Bentley Arnage Sedan 2009 | Canada_Warbler |
| 39 | Bentley Continental Flying Spur Sedan 2007 | Cape_Glossy_Starling |

| Index | StanfordCars Class | CUB200 Class |
|---|---|---|
| 40 | Bentley Continental GT Coupe 2007 | Cape_May_Warbler |
| 41 | Bentley Continental GT Coupe 2012 | Cardinal |
| 42 | Bentley Continental Supersports Conv. Convertible 2012 | Carolina_Wren |
| 43 | Bentley Mulsanne Sedan 2011 | Caspian_Tern |
| 44 | Bugatti Veyron 16.4 Convertible 2009 | Cedar_Waxwing |
| 45 | Bugatti Veyron 16.4 Coupe 2009 | Cerulean_Warbler |
| 46 | Buick Enclave SUV 2012 | Chestnut_sided_Warbler |
| 47 | Buick Rainier SUV 2007 | Chipping_Sparrow |
| 48 | Buick Regal GS 2012 | Chuck_will_Widow |
| 49 | Buick Verano Sedan 2012 | Clark_Nutcracker |
| 50 | Cadillac CTS-V Sedan 2012 | Clay_colored_Sparrow |
| 51 | Cadillac Escalade EXT Crew Cab 2007 | Cliff_Swallow |
| 52 | Cadillac SRX SUV 2012 | Common_Raven |
| 53 | Chevrolet Avalanche Crew Cab 2012 | Common_Tern |
| 54 | Chevrolet Camaro Convertible 2012 | Common_Yellowthroat |
| 55 | Chevrolet Cobalt SS 2010 | Crested_Auklet |
| 56 | Chevrolet Corvette Convertible 2012 | Dark_eyed_Junco |
| 57 | Chevrolet Corvette Ron Fellows Edition Z06 2007 | Downy_Woodpecker |
| 58 | Chevrolet Corvette ZR1 2012 | Eared_Grebe |
| 59 | Chevrolet Express Cargo Van 2007 | Eastern_Towhee |
| 60 | Chevrolet Express Van 2007 | Elegant_Tern |
| 61 | Chevrolet HHR SS 2010 | European_Goldfinch |
| 62 | Chevrolet Impala Sedan 2007 | Evening_Grosbeak |
| 63 | Chevrolet Malibu Hybrid Sedan 2010 | Field_Sparrow |
| 64 | Chevrolet Malibu Sedan 2007 | Fish_Crow |
| 65 | Chevrolet Monte Carlo Coupe 2007 | Florida_Jay |
| 66 | Chevrolet Silverado 1500 Classic Extended Cab 2007 | Forsters_Tern |
| 67 | Chevrolet Silverado 1500 Extended Cab 2012 | Fox_Sparrow |
| 68 | Chevrolet Silverado 1500 Hybrid Crew Cab 2012 | Frigatebird |
| 69 | Chevrolet Silverado 1500 Regular Cab 2012 | Gadwall |
| 70 | Chevrolet Silverado 2500HD Regular Cab 2012 | Geococcyx |
| 71 | Chevrolet Sonic Sedan 2012 | Glaucous_winged_Gull |
| 72 | Chevrolet Tahoe Hybrid SUV 2012 | Golden_winged_Warbler |
| 73 | Chevrolet TrailBlazer SS 2009 | Grasshopper_Sparrow |
| 74 | Chevrolet Traverse SUV 2012 | Gray_Catbird |
| 75 | Chrysler 300 SRT-8 2010 | Gray_Kingbird |
| 76 | Chrysler Aspen SUV 2009 | Gray_crowned_Rosy_Finch |
| 77 | Chrysler Crossfire Convertible 2008 | Great_Crested_Flycatcher |
| 78 | Chrysler PT Cruiser Convertible 2008 | Great_Grey_Shrike |
| 79 | Chrysler Sebring Convertible 2010 | Green_Jay |
| 80 | Chrysler Town and Country Minivan 2012 | Green_Kingfisher |
| 81 | Daewoo Nubira Wagon 2002 | Green_Violetear |
| 82 | Dodge Caliber Wagon 2007 | Green_tailed_Towhee |
| 83 | Dodge Caliber Wagon 2012 | Groove_billed_Ani |
| 84 | Dodge Caravan Minivan 1997 | Harris_Sparrow |
| 85 | Dodge Challenger SRT8 2011 | Heermann_Gull |
| 86 | Dodge Charger SRT-8 2009 | Henslow_Sparrow |
| 87 | Dodge Charger Sedan 2012 | Herring_Gull |
| 88 | Dodge Dakota Club Cab 2007 | Hooded_Merganser |
| 89 | Dodge Dakota Crew Cab 2010 | Hooded_Oriole |
| 90 | Dodge Durango SUV 2007 | Hooded_Warbler |
| 91 | Dodge Durango SUV 2012 | Horned_Grebe |
| 92 | Dodge Journey SUV 2012 | Horned_Lark |
| 93 | Dodge Magnum Wagon 2008 | Horned_Puffin |
| 94 | Dodge Ram Pickup 3500 Crew Cab 2010 | House_Sparrow |
| 95 | Dodge Ram Pickup 3500 Quad Cab 2009 | House_Wren |
| 96 | Dodge Sprinter Cargo Van 2009 | Indigo_Bunting |
| 97 | Eagle Talon Hatchback 1998 | Ivory_Gull |

| Index | StanfordCars Class | CUB200 Class |
|---|---|---|
| 98 | FIAT 500 Abarth 2012 | Kentucky_Warbler |
| 99 | FIAT 500 Convertible 2012 | Laysan_Albatross |
| 100 | Ferrari 458 Italia Convertible 2012 | Lazuli_Bunting |
| 101 | Ferrari 458 Italia Coupe 2012 | Le_Conte_Sparrow |
| 102 | Ferrari California Convertible 2012 | Least_Auklet |
| 103 | Ferrari FF Coupe 2012 | Least_Flycatcher |
| 104 | Fisker Karma Sedan 2012 | Least_Tern |
| 105 | Ford E-Series Wagon Van 2012 | Lincoln_Sparrow |
| 106 | Ford Edge SUV 2012 | Loggerhead_Shrike |
| 107 | Ford Expedition EL SUV 2009 | Long_tailed_Jaeger |
| 108 | Ford F-150 Regular Cab 2007 | Louisiana_Waterthrush |
| 109 | Ford F-150 Regular Cab 2012 | Magnolia_Warbler |
| 110 | Ford F-450 Super Duty Crew Cab 2012 | Mallard |
| 111 | Ford Fiesta Sedan 2012 | Mangrove_Cuckoo |
| 112 | Ford Focus Sedan 2007 | Marsh_Wren |
| 113 | Ford Freestar Minivan 2007 | Mockingbird |
| 114 | Ford GT Coupe 2006 | Mourning_Warbler |
| 115 | Ford Mustang Convertible 2007 | Myrtle_Warbler |
| 116 | Ford Ranger SuperCab 2011 | Nashville_Warbler |
| 117 | GMC Acadia SUV 2012 | Nelson_Sharp_tailed_Sparrow |
| 118 | GMC Canyon Extended Cab 2012 | Nighthawk |
| 119 | GMC Savana Van 2012 | Northern_Flicker |
| 120 | GMC Terrain SUV 2012 | Northern_Fulmar |
| 121 | GMC Yukon Hybrid SUV 2012 | Northern_Waterthrush |
| 122 | Geo Metro Convertible 1993 | Olive_sided_Flycatcher |
| 123 | HUMMER H2 SUT Crew Cab 2009 | Orange_crowned_Warbler |
| 124 | HUMMER H3T Crew Cab 2010 | Orchard_Oriole |
| 125 | Honda Accord Coupe 2012 | Ovenbird |
| 126 | Honda Accord Sedan 2012 | Pacific_Loon |
| 127 | Honda Odyssey Minivan 2007 | Painted_Bunting |
| 128 | Honda Odyssey Minivan 2012 | Palm_Warbler |
| 129 | Hyundai Accent Sedan 2012 | Parakeet_Auklet |
| 130 | Hyundai Azera Sedan 2012 | Pelagic_Cormorant |
| 131 | Hyundai Elantra Sedan 2007 | Philadelphia_Vireo |
| 132 | Hyundai Elantra Touring Hatchback 2012 | Pied_Kingfisher |
| 133 | Hyundai Genesis Sedan 2012 | Pied_billed_Grebe |
| 134 | Hyundai Santa Fe SUV 2012 | Pigeon_Guillemot |
| 135 | Hyundai Sonata Hybrid Sedan 2012 | Pileated_Woodpecker |
| 136 | Hyundai Sonata Sedan 2012 | Pine_Grosbeak |
| 137 | Hyundai Tucson SUV 2012 | Pine_Warbler |
| 138 | Hyundai Veloster Hatchback 2012 | Pomarine_Jaeger |
| 139 | Hyundai Veracruz SUV 2012 | Prairie_Warbler |
| 140 | Infiniti G Coupe IPL 2012 | Prothonotary_Warbler |
| 141 | Infiniti QX56 SUV 2011 | Purple_Finch |
| 142 | Isuzu Ascender SUV 2008 | Red_bellied_Woodpecker |
| 143 | Jaguar XK XKR 2012 | Red_breasted_Merganser |
| 144 | Jeep Compass SUV 2012 | Red_cockaded_Woodpecker |
| 145 | Jeep Grand Cherokee SUV 2012 | Red_eyed_Vireo |
| 146 | Jeep Liberty SUV 2012 | Red_faced_Cormorant |
| 147 | Jeep Patriot SUV 2012 | Red_headed_Woodpecker |
| 148 | Jeep Wrangler SUV 2012 | Red_legged_Kittiwake |
| 149 | Lamborghini Aventador Coupe 2012 | Red_winged_Blackbird |
| 150 | Lamborghini Diablo Coupe 2001 | Rhinoceros_Auklet |
| 151 | Lamborghini Gallardo LP 570-4 Superleggera 2012 | Ring_billed_Gull |
| 152 | Lamborghini Reventon Coupe 2008 | Ringed_Kingfisher |
| 153 | Land Rover LR2 SUV 2012 | Rock_Wren |
| 154 | Land Rover Range Rover SUV 2012 | Rose_breasted_Grosbeak |
| 155 | Lincoln Town Car Sedan 2011 | Ruby_throated_Hummingbird |

| Index | StanfordCars Class | CUB200 Class |
|-------|--------------------|--------------|
| 156 | MINI Cooper Roadster Convertible 2012 | Rufous_Hummingbird |
| 157 | Maybach Landaulet Convertible 2012 | Rusty_Blackbird |
| 158 | Mazda Tribute SUV 2011 | Sage_Thrasher |
| 159 | McLaren MP4-12C Coupe 2012 | Savannah_Sparrow |
| 160 | Mercedes-Benz 300-Class Convertible 1993 | Sayornis |
| 161 | Mercedes-Benz C-Class Sedan 2012 | Scarlet_Tanager |
| 162 | Mercedes-Benz E-Class Sedan 2012 | Scissor_tailed_Flycatcher |
| 163 | Mercedes-Benz S-Class Sedan 2012 | Scott_Oriole |
| 164 | Mercedes-Benz SL-Class Coupe 2009 | Seaside_Sparrow |
| 165 | Mercedes-Benz Sprinter Van 2012 | Shiny_Cowbird |
| 166 | Mitsubishi Lancer Sedan 2012 | Slaty_backed_Gull |
| 167 | Nissan 240SX Coupe 1998 | Song_Sparrow |
| 168 | Nissan Juke Hatchback 2012 | Sooty_Albatross |
| 169 | Nissan Leaf Hatchback 2012 | Spotted_Catbird |
| 170 | Nissan NV Passenger Van 2012 | Summer_Tanager |
| 171 | Plymouth Neon Coupe 1999 | Swainson_Warbler |
| 172 | Porsche Panamera Sedan 2012 | Tennessee_Warbler |
| 173 | Ram C-V Cargo Van Minivan 2012 | Tree_Sparrow |
| 174 | Rolls-Royce Ghost Sedan 2012 | Tree_Swallow |
| 175 | Rolls-Royce Phantom Drophead Coupe Convertible 2012 | Tropical_Kingbird |
| 176 | Rolls-Royce Phantom Sedan 2012 | Vermilion_Flycatcher |
| 177 | Scion xD Hatchback 2012 | Vesper_Sparrow |
| 178 | Spyker C8 Convertible 2009 | Warbling_Vireo |
| 179 | Spyker C8 Coupe 2009 | Western_Grebe |
| 180 | Suzuki Aerio Sedan 2007 | Western_Gull |
| 181 | Suzuki Kizashi Sedan 2012 | Western_Meadowlark |
| 182 | Suzuki SX4 Hatchback 2012 | Western_Wood_Pewee |
| 183 | Suzuki SX4 Sedan 2012 | Whip_poor_Will |
| 184 | Tesla Model S Sedan 2012 | White_Pelican |
| 185 | Toyota 4Runner SUV 2012 | White_breasted_Kingfisher |
| 186 | Toyota Camry Sedan 2012 | White_breasted_Nuthatch |
| 187 | Toyota Corolla Sedan 2012 | White_crowned_Sparrow |
| 188 | Toyota Sequoia SUV 2012 | White_eyed_Vireo |
| 189 | Volkswagen Beetle Hatchback 2012 | White_necked_Raven |
| 190 | Volkswagen Golf Hatchback 1991 | White_throated_Sparrow |
| 191 | Volkswagen Golf Hatchback 2012 | Wilson_Warbler |
| 192 | Volvo 240 Sedan 1993 | Winter_Wren |
| 193 | Volvo C30 Hatchback 2012 | Worm_eating_Warbler |
| 194 | Volvo XC90 SUV 2007 | Yellow_Warbler |
| 195 | smart fortwo Convertible 2012 | Yellow_bellied_Flycatcher |
| 196 | | Yellow_billed_Cuckoo |
| 197 | | Yellow_breasted_Chat |
| 198 | | Yellow_headed_Blackbird |
| 199 | | Yellow_throated_Vireo |

Table 7: **Dataset class indices.** We provide the class indices for OxfordPets, Caltech101, and Mini-ImageNet, which have 37, 101, and 100 classes, respectively.

| Index | OxfordPets Class | Caltech101 Class | Mini-ImageNet Class |
|-------|------------------|------------------|---------------------|
| 0 | Abyssinian | Faces | African_hunting_dog |
| 1 | Bengal | Faces_easy | Arctic_fox |
| 2 | Birman | Leopards | French_bulldog |
| 3 | Bombay | Motorbikes | Gordon_setter |
| 4 | British_Shorthair | accordion | Ibizan_hound |
| 5 | Egyptian_Mau | airplanes | Newfoundland |
| 6 | Maine_Coon | anchor | Saluki |
| 7 | Persian | ant | Tibetan_mastiff |

| Index | OxfordPets Class | Caltech101 Class | Mini-ImageNet Class |
|---|---|---|---|
| 8 | Ragdoll | barrel | Walker_hound |
| 9 | Russian_Blue | bass | aircraft_carrier |
| 10 | Siamese | beaver | ant |
| 11 | Sphynx | binocular | ashcan |
| 12 | american_bulldog | bonsai | barrel |
| 13 | american_pit_bull_terrier | brain | beer_bottle |
| 14 | basset_hound | brontosaurus | black-footed_ferret |
| 15 | beagle | buddha | bolete |
| 16 | boxer | butterfly | bookshop |
| 17 | chihuahua | camera | boxer |
| 18 | english_cocker_spaniel | cannon | cannon |
| 19 | english_setter | car_side | carousel |
| 20 | german_shorthaired | ceiling_fan | carton |
| 21 | great_pyrenees | cellphone | catamaran |
| 22 | havanese | chair | chime |
| 23 | japanese_chin | chandelier | cliff |
| 24 | keeshond | cougar_body | clog |
| 25 | leonberger | cougar_face | cocktail_shaker |
| 26 | miniature_pinscher | crab | combination_lock |
| 27 | newfoundland | crayfish | consomme |
| 28 | pomeranian | crocodile | coral_reef |
| 29 | pug | crocodile_head | crate |
| 30 | saint_bernard | cup | cuirass |
| 31 | samoyed | dalmatian | dalmatian |
| 32 | scottish_terrier | dollar_bill | dishrag |
| 33 | shiba_inu | dolphin | dome |
| 34 | staffordshire_bull_terrier | dragonfly | dugong |
| 35 | wheaten_terrier | electric_guitar | electric_guitar |
| 36 | yorkshire_terrier | elephant | file |
| 37 | | emu | fire_screen |
| 38 | | euphonium | frying_pan |
| 39 | | ewer | garbage_truck |
| 40 | | ferry | golden_retriever |
| 41 | | flamingo | goose |
| 42 | | flamingo_head | green_mamba |
| 43 | | garfield | hair_slide |
| 44 | | gerenuk | harvestman |
| 45 | | gramophone | holster |
| 46 | | grand_piano | horizontal_bar |
| 47 | | hawksbill | hotdog |
| 48 | | headphone | hourglass |
| 49 | | hedgehog | house_finch |
| 50 | | helicopter | iPod |
| 51 | | ibis | jellyfish |
| 52 | | inline_skate | king_crab |
| 53 | | joshua_tree | komondor |
| 54 | | kangaroo | ladybug |
| 55 | | ketch | lion |
| 56 | | lamp | lipstick |
| 57 | | laptop | malamute |
| 58 | | llama | meerkat |
| 59 | | lobster | miniature_poodle |
| 60 | | lotus | miniskirt |
| 61 | | mandolin | missile |
| 62 | | mayfly | mixing_bowl |
| 63 | | menorah | nematode |
| 64 | | metronome | oboe |
| 65 | | minaret | orange |

| Index | OxfordPets Class | Caltech101 Class | Mini-ImageNet Class |
|-------|------------------|------------------|---------------------|
| 66 | | nautilus | organ |
| 67 | | octopus | parallel_bars |
| 68 | | okapi | pencil_box |
| 69 | | pagoda | photocopier |
| 70 | | panda | poncho |
| 71 | | pigeon | prayer_rug |
| 72 | | pizza | reel |
| 73 | | platypus | rhinoceros_beetle |
| 74 | | pyramid | robin |
| 75 | | revolver | rock_beauty |
| 76 | | rhino | school_bus |
| 77 | | rooster | scoreboard |
| 78 | | saxophone | slot |
| 79 | | schooner | snorkel |
| 80 | | scissors | solar_dish |
| 81 | | scorpion | spider_web |
| 82 | | sea_horse | spike |
| 83 | | snoopy | stage |
| 84 | | soccer_ball | street_sign |
| 85 | | stapler | tank |
| 86 | | starfish | theater_curtain |
| 87 | | stegosaurus | three-toed_sloth |
| 88 | | stop_sign | tile_roof |
| 89 | | strawberry | tobacco_shop |
| 90 | | sunflower | toucan |
| 91 | | tick | triceratops |
| 92 | | trilobite | trifle |
| 93 | | umbrella | unicycle |
| 94 | | watch | upright_piano |
| 95 | | water_lilly | vase |
| 96 | | wheelchair | white_wolf |
| 97 | | wild_cat | wok |
| 98 | | windsor_chair | worm_fence |
| 99 | | wrench | yawl |
| 100 | | yin_yang | |

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

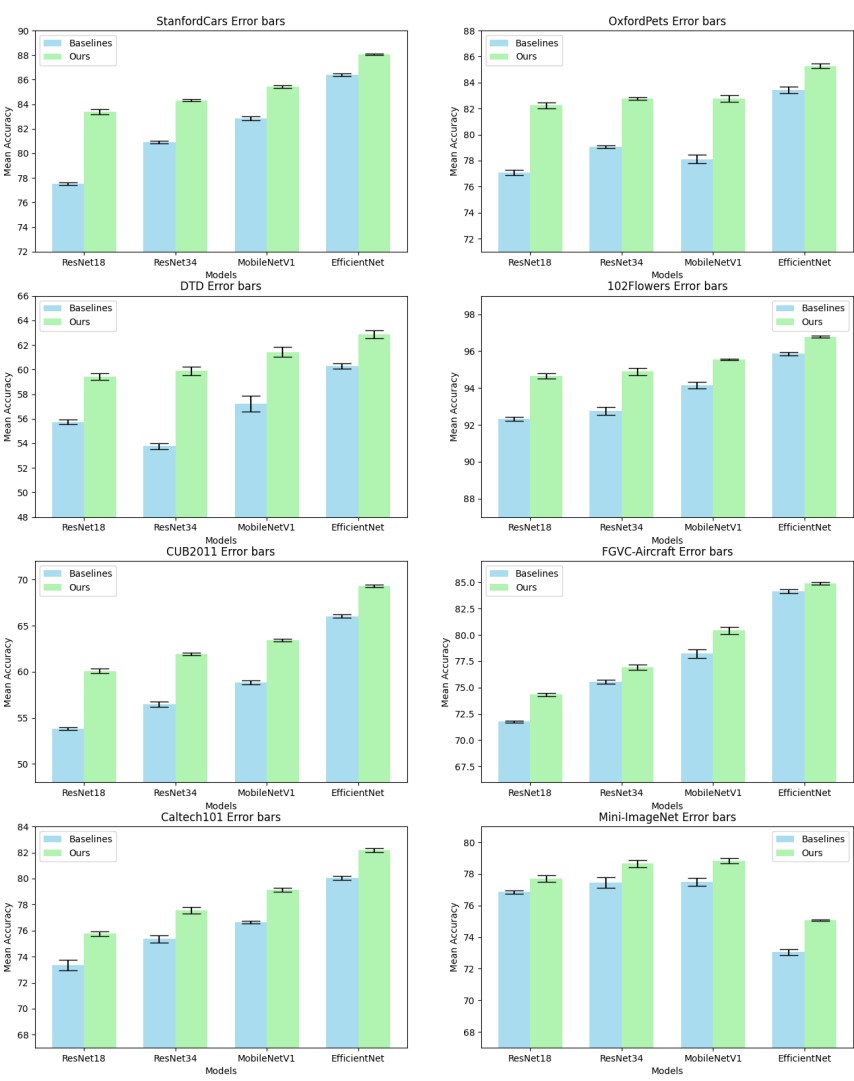

Figure 2: **Error bars of SOTA experimental results on fine-grained and coarse-grained datasets.** We run each experiment three times and report the average results. This results represent the standard deviation of the performance across multiple runs for both fine-grained and coarse-grained datasets, reflecting the variability and stability of the SOTA experiment results.

1. Does the car belong to the high-end luxury category (like Bugatti, Bentley, etc.)?
2. Is the car's make year post-2010?
3. Is the car equipped with a V8 engine?
4. Is the car's model a hatchback?
5. Is the car model a SUV?
6. Is the car a diesel-powered model?
7. Is the car a model of Chevrolet brand?
8. Does the car have a convertible roof?
9. Is the car a sports coupe model?
10. Does the car belong to the sedan category?
11. Does the picture depict a sports version of a typical car model (like Audi RS, Aston Martin V8 Vantage etc.)?
12. Is the car a hybrid vehicle?
13. Is the car from the minivan category?
14. Does the car have a noticeable rear spoiler?
15. Is the car model from the smaller Compact Class?
16. Is the make of the car BMW?
17. Is the car part of the Ford family?
18. Have the car images been taken after 2007?
19. Is the car a part of the Italian luxury car brands (like Ferrari, Lamborghini)?
20. Was the car model made in the V12 engine series?
21. Does the car have scissor doors?
22. Does the car have distinctive gull-wing doors?
23. Does the car have a rear engine layout?
24. Is the car an off-road vehicle or designed for rugged terrain usage?
25. Does the car feature a dual exhaust system?
26. Is the car a roadster model?
27. Is the car equipped with side skirts?
28. Is the car a station wagon?
29. Does the car feature a distinctive front grille with vertical slats?
30. Is the car from the Japanese automaker, Honda?
31. Is the car a part of the electric car category (like Tesla Model S, Chevrolet Bolt, etc.)?
32. Does the car have a long wheelbase version?
33. Does the car have a soft-top roof?
34. Is the car a coupe with two doors?
35. Is the car from a Korean manufacturer (like Hyundai, Kia)?
36. Does the car have a distinctive round headlight design?
37. Does the car belong to the pickup truck category?
38. Does the car have a distinctive boxy shape?
39. Is the car a 4-door model?
40. Is the car equipped with a turbocharger?
41. Is the car a plug-in hybrid?
42. Does the car feature a panoramic sunroof?
43. Is the car a muscle car (like Dodge Challenger, Chevrolet Camaro)?
44. Does the car have a noticeable hood scoop?
45. Does the car have all-wheel drive (AWD)?
46. Is the car a vintage model made before 2000?
47. Does the car have a prominent air intake on the front bumper?
48. Does the car have a distinctive rear diffuser?
49. Is the car from an American manufacturer?
50. Is the car a convertible with a hardtop?

Table 8: **Multi-aspect questions generated by GPT-4o for the StanfordCars dataset.**

1. Does the animal have a flat face?
2. Does the animal display a prominent ruff around the neck?
3. Are the ears of the breed long and floppy?
4. Is the animal's fur long?
5. Does the animal have a robust and muscular build?
6. Does the breed have a compact and muscular build?
7. Does the animal have long drooping ears?
8. Does the animal have distinctive facial markings?
9. Does the animal have striking blue eyes?
10. Does the animal have a brachycephalic (shortened head) skull?
11. Does the animal have a double coat?
12. Is the animal's body unusually slender and tall?
13. Is the breed's coat spotted or dappled?
14. Is the tail of the animal bushy or feathered?
15. Does the animal have webbed feet?
16. Does the animal have a short, stubby nose?
17. Does the animal have floppy ears?
18. Is the animal's coat curly or wavy?
19. Is the fur of the animal curly or wavy?
20. Does the animal have hairless skin?
21. Does the breed have a plumed tail?
22. Does the animal have a long, flowing coat?
23. Is the animal small-sized, typically less than 10 pounds?
24. Does the breed have a square-shaped body?
25. Is the animal typically solid-colored?
26. Is the animal predominantly white in color?
27. Is the breed characterized by a high-set tail?
28. Does the breed have a short snout?
29. Is the breed's tail bushy or fluffy?
30. Does the animal have a pronounced underbite?
31. Does the animal have an unusually squarish or boxy muzzle?
32. Is the fur patterned with spots or stripes?
33. Does the breed have a pointed muzzle?
34. Does the animal have a characteristically flat or pushed-in face with large, round eyes?
35. Does the animal have large, round eyes?
36. Is the animal's coat silky to the touch?
37. Does the breed have a long and slender tail?
38. Is the breed's coat rough or wiry?
39. Is the animal known for having a sleek and shiny coat?
40. Is the breed known for having a slender body?
41. Is the breed known for its distinctive coloration or pattern?
42. Does the breed have a broad chest?
43. Does the breed have a distinctive ruff or collar of fur around the neck?
44. Is the animal predominantly black in color?
45. Does the breed have large, bat-like ears?
46. Is the breed's coat short and dense?
47. Does the breed have a docked or naturally short tail?
48. Does the breed have small, pointed ears?
49. Is the animal known for having a lion-like appearance?
50. Is the animal's coat thick and woolly?

Table 9: **Multi-aspect questions generated by GPT-4o for the OxfordPets dataset.**

1. Are there noticeable cracks or fissures?
2. Does the texture have a scaly or reptilian appearance?
3. Does the texture feature honeycomb-like hexagonal shapes?
4. Does the texture look like a net or web?
5. Are there regular, grid-like patterns?
6. Are there noticeable swirls or spiral patterns?
7. Is the texture characterized by a dotted or spotted pattern?
8. Are there visible grooves or indentations?
9. Are there waffle-like grid patterns on the texture?
10. Does the texture have a marbled appearance with blended colors?
11. Is the texture perforated or has holes?
12. Is the pattern composed of crisscrossing lines?
13. Are there distinct, irregular blotches?
14. Does the texture have a checkered or chequered pattern?
15. Are there fibrous or thread-like elements visible?
16. Does the texture have a veined appearance?
17. Does the texture have a crystalline or gem-like appearance?
18. Are there raised, bumpy areas on the texture?
19. Does the texture appear braided with intertwining strands?
20. Is the texture wrinkled or creased?
21. Is the texture characterized by fine, lace-like details?
22. Does the texture have a smeared or smudged appearance?
23. Is the texture smeared with streaks or smears?
24. Are there visible stains or discolorations on the texture?
25. Does the texture feature pleated or folded sections?
26. Is the pattern composed of zigzag lines?
27. Does the texture have a sprinkled or speckled look?
28. Are there any interwoven or braided elements in the texture?
29. Is the texture banded with stripes of varying widths?
30. Is the texture composed of overlapping or interlaced elements?
31. Is the texture flecked with small, random spots?
32. Are there frilly or ruffled edges in the texture?
33. Does the texture have a porous or sponge-like look?
34. Is the texture marked by potholes or deep indentations?
35. Is the texture covered with polka dots?
36. Are there visible knitted or crocheted patterns?
37. Does the texture have a paisley or teardrop-shaped pattern?
38. Are there pitted or dimpled areas on the texture?
39. Does the texture have a stratified or layered appearance?
40. Does the texture feature stratified layers or bands?
41. Are there visible bubbles or circular shapes?
42. Is the texture cobwebbed with thin, thread-like lines?
43. Does the texture resemble fabric or woven material?
44. Is the texture marked by crosshatched lines?
45. Does the texture have parallel lines?
46. Are there noticeable stained or dirty areas?
47. Does the texture have a woven or interlaced look?
48. Is the texture swirly with swirling patterns?
49. Are there noticeable wrinkles or creases?
50. Does the texture have a zigzagged pattern?

Table 10: **Multi-aspect questions generated by GPT-4o for the DTD dataset.**

1. Does the flower have multiple petals arranged in a symmetrical pattern?
2. Does the flower have heart-shaped petals?
3. Does the flower have a prominent central disk surrounded by petals?
4. Are there multiple small flowers arranged in a cluster?
5. Is the flower predominantly blue or purple?
6. Does the flower exhibit a gradient of colors?
7. Does the flower have a spiky or thistle-like appearance?
8. Does the flower have a large, singular bloom?
9. Is the primary colors of the flower yellow?
10. Are the petals long and narrow, resembling a lily?
11. Does the flower have a tubular shape?
12. Is the flower predominantly red?
13. Are the petals arranged in layers or rows?
14. Are the petals overlapping?
15. Are the petals fringed or ruffled?
16. Does the flower grow in a cluster on a single stem?
17. Are the petals shaped like a star or have pointed tips?
18. Does the flower have a distinct, pronounced lip or 'tongue' petal?
19. Does the flower have a spurred petal or elongated appendage?
20. Are the petals arranged in a spiral pattern?
21. Does the flower have a strong fragrance?
22. Is the flower predominantly pink?
23. Does the flower have a bell or trumpet shape?
24. Does the flower have a daisy-like appearance?
25. Does the flower have a cup-shaped structure?
26. Does the flower have hairy or fuzzy petals?
27. Are the petals thin and delicate?
28. Are the petals bi-colored?
29. Are the petals flat and wide?
30. Does the flower have a central crown or corona?
31. Is the flower predominantly white?
32. Are the petals veined or patterned?
33. Are the petals rounded at the tips?
34. Does the flower have strap-like petals?
35. Are the petals twisted or curled?
36. Does the flower have a single petal?
37. Does the flower have a dome-shaped appearance?
38. Is the flower predominantly orange?
39. Does the flower have a flattened top?
40. Are the petals spoon-shaped?
41. Are the petals translucent or semi-transparent?
42. Does the flower have prominent stamens?
43. Is the flower predominantly green?
44. Does the flower have a geometric pattern on its petals?
45. Does the flower have a papery texture?
46. Are the petals serrated or jagged?
47. Are the petals clustered tightly together?
48. Is the flower predominantly violet?
49. Does the flower have drooping petals?
50. Are the petals reflexed or bent backward?

Table 11: **Multi-aspect questions generated by GPT-4o for the 102Flowers dataset.**

1. Does the bird have a curved beak?
2. Is the bird's beak long and pointed?
3. Is the bird predominantly blue?
4. Is the bird's primary habitat coastal areas?
5. Is the bird primarily found in water habitats?
6. Is the bird's beak hooked?
7. Is the bird's underside orange?
8. Does the bird have a long neck?
9. Is the bird's plumage mostly white?
10. Is the bird predominantly found in forests?
11. Does the bird have a thin, needle-like beak?
12. Does the bird have a crest on its head?
13. Does the bird have iridescent feathers?
14. Is the bird's beak short and thick?
15. Is the bird's beak conical?
16. Is the bird's plumage predominantly brown?
17. Does the bird have a fan-shaped tail?
18. Does the bird have a black and white striped pattern?
19. Does the bird have a red patch on its wings?
20. Is the bird's breast yellow?
21. Does the bird have a white eye stripe?
22. Does the bird have webbed feet?
23. Does the bird have a notched tail?
24. Is the bird's chest streaked?
25. Does the bird have a ring around its neck?
26. Does the bird have a black cap on its head?
27. Does the bird have a speckled breast?
28. Does the bird have long legs?
29. Is the bird's back green?
30. Does the bird have a black tail?
31. Does the bird have a mask-like pattern on its face?
32. Does the bird have a prominent eye ring?
33. Is the bird's tail short and square?
34. Does the bird have spots on its wings?
35. Is the bird's belly white?
36. Does the bird have a distinctive call that includes trills?
37. Does the bird have a yellow belly?
38. Is the bird's tail forked?
39. Is the bird's head and back grey?
40. Is the bird's wingspan larger than 12 inches?
41. Does the bird have a barred tail?
42. Is the bird's chest red?
43. Does the bird have a blue throat patch?
44. Does the bird have a bright orange beak?
45. Is the bird's head black?
46. Is the bird's beak straight?
47. Is the bird predominantly found in open grasslands?
48. Is the bird larger than a sparrow?
49. Does the bird have a yellow stripe on its wings?
50. Is the bird primarily insectivorous?

Table 12: **Multi-aspect questions generated by GPT-4o for the CUB200 dataset.**

1. Is this aircraft a turboprop model?
2. Does this aircraft have four engines?
3. Does this aircraft have a high-wing design?
4. Does this aircraft have two engines?
5. Is this a single-engine aircraft?
6. Is this aircraft used primarily for military purposes?
7. Is this aircraft a trijet (three engines)?
8. Does this aircraft feature a swept-wing design?
9. Is this aircraft a wide-body model?
10. Does the aircraft have propellers instead of jet engines?
11. Does the aircraft feature a T-tail design?
12. Does this aircraft have retractable landing gear?
13. Is this aircraft primarily used for cargo transportation?
14. Does this aircraft have an open cockpit?
15. Is the aircraft primarily used for commercial passenger flights?
16. Does this aircraft have a twin-jet engine configuration?
17. Is this aircraft primarily used for private or corporate purposes?
18. Does the aircraft have a delta wing configuration?
19. Does this aircraft have a single vertical stabilizer?
20. Is this aircraft a supersonic jet?
21. Is this aircraft used primarily for short regional flights?
22. Does this aircraft have an all-metal body?
23. Does this aircraft have winglets?
24. Does the aircraft have a long-range flight capacity?
25. Does this aircraft have a radial engine?
26. Does this aircraft have a tailwheel landing gear configuration?
27. Is this aircraft a high-performance jet?
28. Does this aircraft feature a pressurized cabin?
29. Does this aircraft have a twin-boom tail design?
30. Does this aircraft have a glass cockpit?
31. Does the aircraft feature swept-back wings?
32. Does this aircraft have a distinctive nose design?
33. Does this aircraft have a forward-swept wing design?
34. Does this aircraft have an all-composite body structure?
35. Does this aircraft have rear-mounted engines?
36. Does this aircraft have a tricycle landing gear configuration?
37. Does this aircraft have variable-sweep wings?
38. Does this aircraft have a high-wing design?
39. Does this aircraft have a distinctive humpback design?
40. Does this aircraft have a high bypass ratio engine?
41. Is this aircraft primarily designed for long-haul flights?
42. Is this aircraft often used for regional transportation?
43. Is this aircraft a narrow-body model?
44. Does the aircraft have turbojet engines?
45. Does this aircraft have twin tail fins?
46. Does this aircraft have a straight-wing design?
47. Is the aircraft designed for short takeoff and landing (STOL)?
48. Does the aircraft feature fixed landing gear?
49. Is this aircraft designed to operate from aircraft carriers?
50. Is the aircraft known for its high maneuverability?

Table 13: **Multi-aspect questions generated by GPT-4o for the FGVC-Aircraft dataset.**

1. Does the object have a recognizable face?
2. Does the object have wings?
3. Is the object known for its ability to fly?
4. Does the object have a screen and interface for digital interaction?
5. Does the object have limbs and a recognizable head/body structure?
6. Is the object known for its speed or ability to move quickly?
7. Does the object have wheels and an enclosed space for passengers?
8. Does the object have multiple legs?
9. Does the object have a prominent trunk or elongated nose?
10. Does the object have fur or hair?
11. Does the object have claws or pincers?
12. Is the object typically found in water or aquatic environments?
13. Is the object known for its ability to cut or pierce?
14. Does the object have a blade or sharp edge for cutting?
15. Does the object have feathers?
16. Is the object a type of vehicle used for transportation?
17. Is the object a type of instrument used to produce sound?
18. Does the object have keys or buttons for producing musical notes?
19. Does the object have a flat surface for placing items on?
20. Is the object used for capturing images or videos?
21. Is the object commonly associated with human activities or use?
22. Is the object typically found outdoors in a natural environment?
23. Does the object have a distinct shape?
24. Is the object commonly found in a garden or botanical setting?
25. Is the object primarily composed of organic materials?
26. Does the object have a shell or hard outer covering?
27. Is the object likely to be found in a domestic setting (home, kitchen)?
28. Is the object typically seen in a kitchen setting?
29. Does the object have any moving parts or mechanisms?
30. Does the object have a repeating pattern or design on its surface?
31. Is the object a type of plant?
32. Does the object have scales or a rough texture?
33. Does the object have an elongated neck?
34. Does the object have a circular or rounded shape?
35. Does the object have a cylindrical shape?
36. Does the object have a distinctive pattern on its surface, like spots or stripes?
37. Does the object have a clear, defined purpose or function?
38. Does this object occupy a large part of the image?
39. Is the object likely part of a larger system or assembly (e.g., part of a car)?
40. Is the object's primary purpose for entertainment or recreation?
41. Is the object typically used for writing or drawing?
42. Is the object often associated with religious or spiritual practices?
43. Is the object often used in sporting activities?
44. Is the object an example of marine life?
45. Is the object a piece of furniture used for seating?
46. Is the dominant color of the object a warm color (red, orange, yellow)?
47. Is the dominant color of the object a cool color (blue, skyblue, gray)?
48. Does the object appear to be handheld or designed for human interaction?
49. Is the object's shape primarily geometric (circles, squares, etc.)?
50. Is the object known for producing light?

Table 14: **Multi-aspect questions generated by GPT-4o for the Caltech101 dataset.**

1. Is the animal in the image a type of bird?
2. Is a creature known for its colorful appearance depicted in the image?
3. Does the image show a vehicle with wheels?
4. Does the image feature a musical instrument?
5. Is there something used for communication in the image?
6. Does the image contain a dog breed with long ears?
7. Is a creature with a distinct mane shown in the image?
8. Is the object in the image made of glass?
9. Is the object in the image typically found in a bathroom?
10. Is a weapon depicted in the image?
11. Is something used in artistic creation featured in the image?
12. Is a type of mushroom shown in the image?
13. Is something used for navigation present in the image?
14. Is something used for measurement present in the image?
15. Is something used for entertainment featured in the image?
16. Is a primarily nocturnal creature depicted in the image?
17. Is a mammal known for swimming depicted in the image?
18. Is there a clothing accessory in the image?
19. Can an amphibian be seen in the image?
20. Is the object shown typically found in a kitchen?
21. Does the image contain an animal with stripes?
22. Is the object depicted primarily used for transportation?
23. Does the image show an object used for cooling?
24. Is there an item associated with food preparation in the image?
25. Is a venomous creature shown in the image?
26. Can a type of beetle be seen in the image?
27. Is the object in the image typically found in water?
28. Is the depicted animal a type of cat?
29. Is there a vehicle without wheels in the image?
30. Is something typically used in a garden present in the image?
31. Can a large marine vessel be seen in the image?
32. Does the image feature an object used in sports?
33. Does the image contain a type of marine life?
34. Is a wild cat depicted in the image?
35. Is something used for timekeeping present in the image?
36. Is the depicted animal a type of amphibian?
37. Is something used for personal grooming in the image?
38. Is a herding dog breed shown in the image?
39. Can an insect with wings be seen in the image?
40. Is the animal in the image known for its speed?
41. Does the image contain a type of fruit?
42. Is the depicted animal a type of reptile?
43. Is something used in photography shown in the image?
44. Is something in the image commonly used for storage?
45. Does the image feature an object used in construction?
46. Is a rodent depicted in the image?
47. Is something typically found in an office present in the image?
48. Does the image contain an animal with a shell?
49. Is there protective gear depicted in the image?
50. Is a creature known for its strength depicted in the image?

Table 15: **Multi-aspect questions generated by GPT-4o for the Mini-ImageNet dataset.**

1. Is the car color black?
2. Is the car a convertible?
3. Is the car a sedan?
4. Is the car from the year 2012?
5. Is the car from the make Acura?
6. Is the car from the make Audi?
7. Is the car from the make BMW?
8. Is the car from the make Chevrolet?
9. Is the car from the make Dodge?
10. Is the car from the make Ferrari?
11. Is the car from the make Ford?
12. Is the car from the make Honda?
13. Is the car from the make Hyundai?
14. Is the car from the make Jeep?
15. Is the car from the make Lamborghini?
16. Is the car from the make Mercedes-Benz?
17. Is the car from the make Nissan?
18. Is the car from the make Porsche?
19. Is the car from the make Rolls-Royce?
20. Is the car from the make Toyota?
21. Is the car from the make Volkswagen?
22. Is the car a coupe?
23. Is the car an SUV?
24. Is the car a hatchback?
25. Is the car a wagon?
26. Is the car a hybrid?
27. Is the car a van?
28. Is the car a minivan?
29. Is the car a crew cab?
30. Is the car a regular cab?
31. Is the car a quad cab?
32. Is the car a club cab?
33. Is the car from the luxury category?
34. Is the car from the sports category?
35. Is the car from the economy category?
36. Is the car from the midsize category?
37. Is the car from the full-size category?
38. Is the car a high-performance model?
39. Is the car a low-performance model?
40. Is the car a high-end model?
41. Is the car a budget-friendly model?
42. Is the car a classic model?
43. Is the car a modern model?
44. Is the car a luxury sports car?
45. Is the car a sedan with a sunroof?
46. Is the car a coupe with a spoiler?
47. Is the car a convertible with a soft top?
48. Is the car a hatchback with a rear spoiler?
49. Is the car a wagon with roof racks?
50. Is the car a van with tinted windows?

Table 16: **Multi-aspect questions generated by GPT-3.5-turbo for the StanfordCars dataset.**

1. Does the breed have a short coat?
2. Does the breed have a long coat?
3. Are the ears of the breed floppy?
4. Are the ears of the breed erect?
5. Does the breed have a solid-colored coat?
6. Does the breed have a spotted coat pattern?
7. Is the breed known for its distinctive facial markings?
8. Is the breed large in size?
9. Is the breed small in size?
10. Does the breed have a curly tail?
11. Does the breed have a bushy tail?
12. Is the breed known for its playful nature?
13. Is the breed known for being affectionate?
14. Does the breed have a brachycephalic (short-nosed) face?
15. Is the coat of the breed fluffy?
16. Does the breed have a stocky build?
17. Is the breed known for its intelligence?
18. Is the breed known for its hunting abilities?
19. Is the breed known for its vocal nature?
20. Does the breed have a specific color pattern unique to its breed?
21. Does the breed have a distinct pattern on its face?
22. Does the breed have a breed-specific tail shape?
23. Is the breed known for its expressive eyes?
24. Does the breed have a muscular build?
25. Does the breed have a sleek and shiny coat?
26. Is the breed known for its agility?
27. Does the breed have a fluffy mane or collar?
28. Is the breed known for its endurance or stamina?
29. Does the breed have a double coat?
30. Is the breed known for its calm temperament?
31. Does the breed have a distinctive vocalization?
32. Is the breed known for its protective instincts?
33. Does the breed have prominent whiskers?
34. Does the breed have a distinctive head shape?
35. Is the breed known for its high energy levels?
36. Does the breed have a short, stubby nose?
37. Is the breed known for its unique tail carriage?
38. Does the breed have a sleek and elegant posture?
39. Is the breed known for its friendly disposition?
40. Does the breed have a sleek and slender build?
41. Is the breed known for its independent nature?
42. Does the breed have a luxurious coat texture?
43. Is the breed known for its social nature?
44. Does the breed have a silky or velvety coat?
45. Is the breed known for its athletic abilities?
46. Does the breed have distinctive facial hair or markings?
47. Is the breed known for its guarding instincts?
48. Does the breed have a thick, protective coat?
49. Is the breed known for its clownish behavior?
50. Does the breed have a unique tail length relative to its body size?

Table 17: **Multi-aspect questions generated by GPT-3.5-turbo for the OxfordPets dataset.**

1. Does the object have wheels?
2. Is the object a type of musical instrument?
3. Does the object have wings?
4. Is the object commonly found in water?
5. Does the object have fur?
6. Is the object commonly used for transportation?
7. Does the object have a shell?
8. Is the object typically found in a household setting?
9. Is the object typically found in nature?
10. Does the object have a long neck?
11. Is the object typically found in an office environment?
12. Does the object have scales?
13. Is the object a type of bird?
14. Does the object have antennas?
15. Does the object have claws?
16. Is the object commonly used for entertainment?
17. Does the object have a tail?
18. Is the object a type of plant?
19. Does the object have a sharp beak?
20. Is the object typically used for sports?
21. Does the object have multiple legs?
22. Is the object typically found in a museum?
23. Does the object have a curved shape?
24. Is the object typically used for cooking?
25. Does the object have a distinctive color pattern?
26. Is the object a type of reptile?
27. Does the object have a smooth texture?
28. Is the object typically found in the sky?
29. Does the object have horns?
30. Is the object typically used for communication?
31. Does the object have a protective shell?
32. Is the object typically found in tropical regions?
33. Does the object have a distinctive smell?
34. Is the object commonly found in urban environments?
35. Does the object have a long tail?
36. Is the object typically used for relaxation?
37. Does the object have a shiny surface?
38. Is the object typically found in cold climates?
39. Does the object have a round shape?
40. Is the object commonly associated with music?
41. Does the object have stripes?
42. Is the object typically found near water?
43. Does the object have a unique pattern?
44. Is the object typically found in forests?
45. Does the object have large ears?
46. Is the object typically found in a desert environment?
47. Does the object have a broad head?
48. Is the object commonly found on farms?
49. Does the object have a pointy nose?
50. Is the object typically used for navigation?

Table 18: **Multi-aspect questions generated by GPT-3.5-turbo for the Caltech101 dataset.**

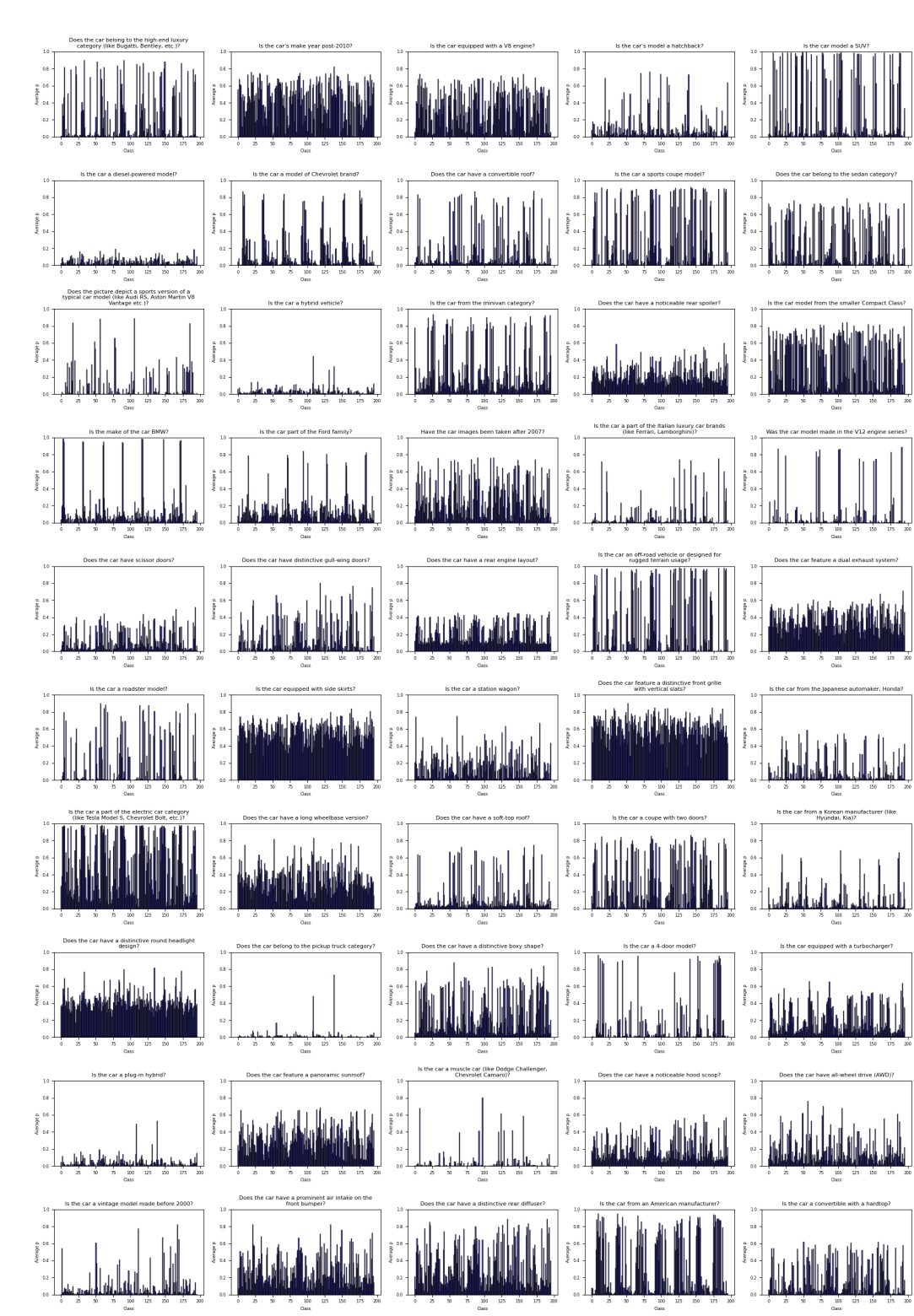

Figure 3: **Visualization of the average logit distribution for StanfordCars.**

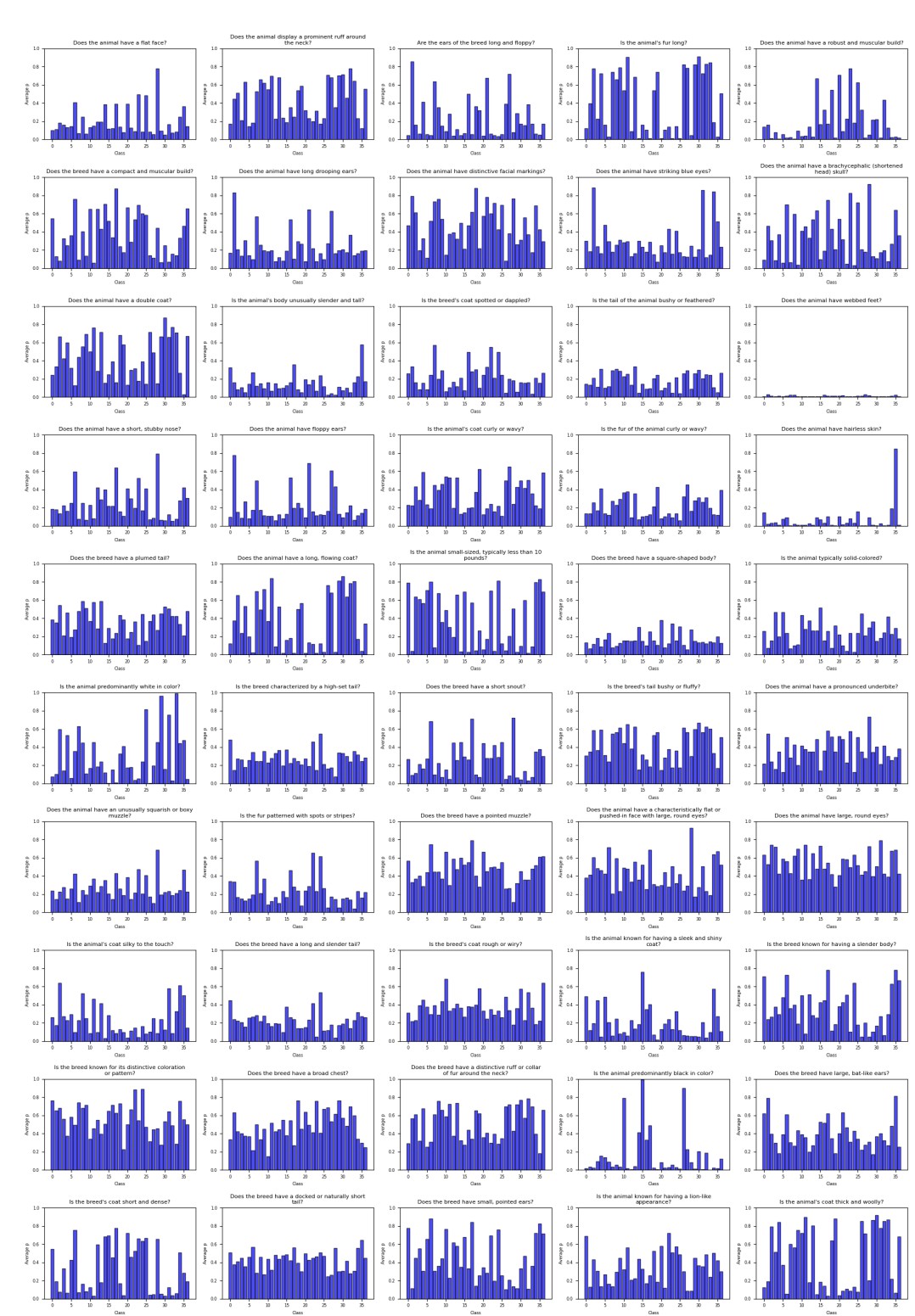

Figure 4: **Visualization of the average logit distribution for the multi-aspect of the OxfordPets.**

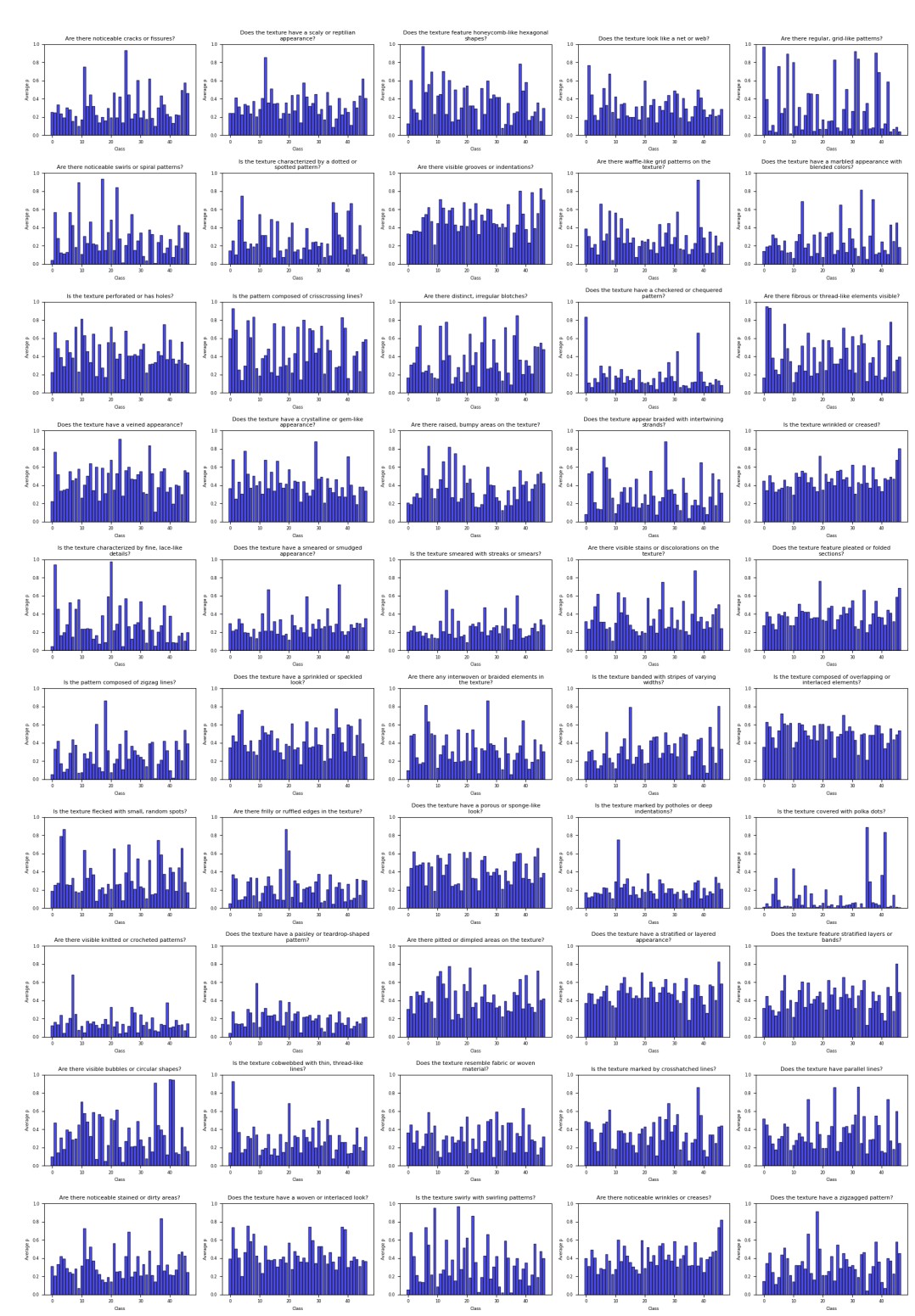

Figure 5: **Visualization of the average logit distribution for the multi-aspect of the DTD.**

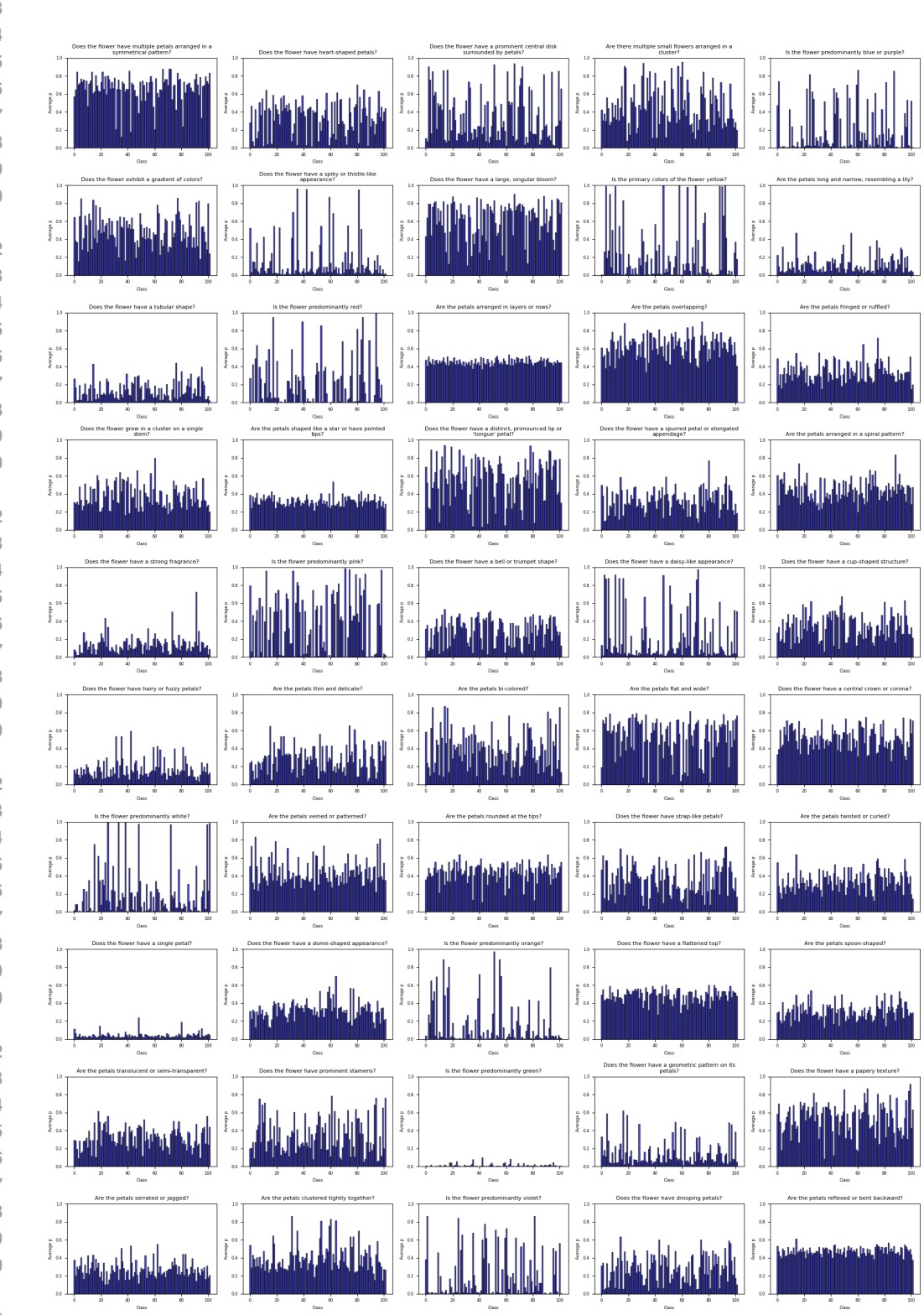

Figure 6: **Visualization of the average logit distribution for the multi-aspect of the 102Flowers.**

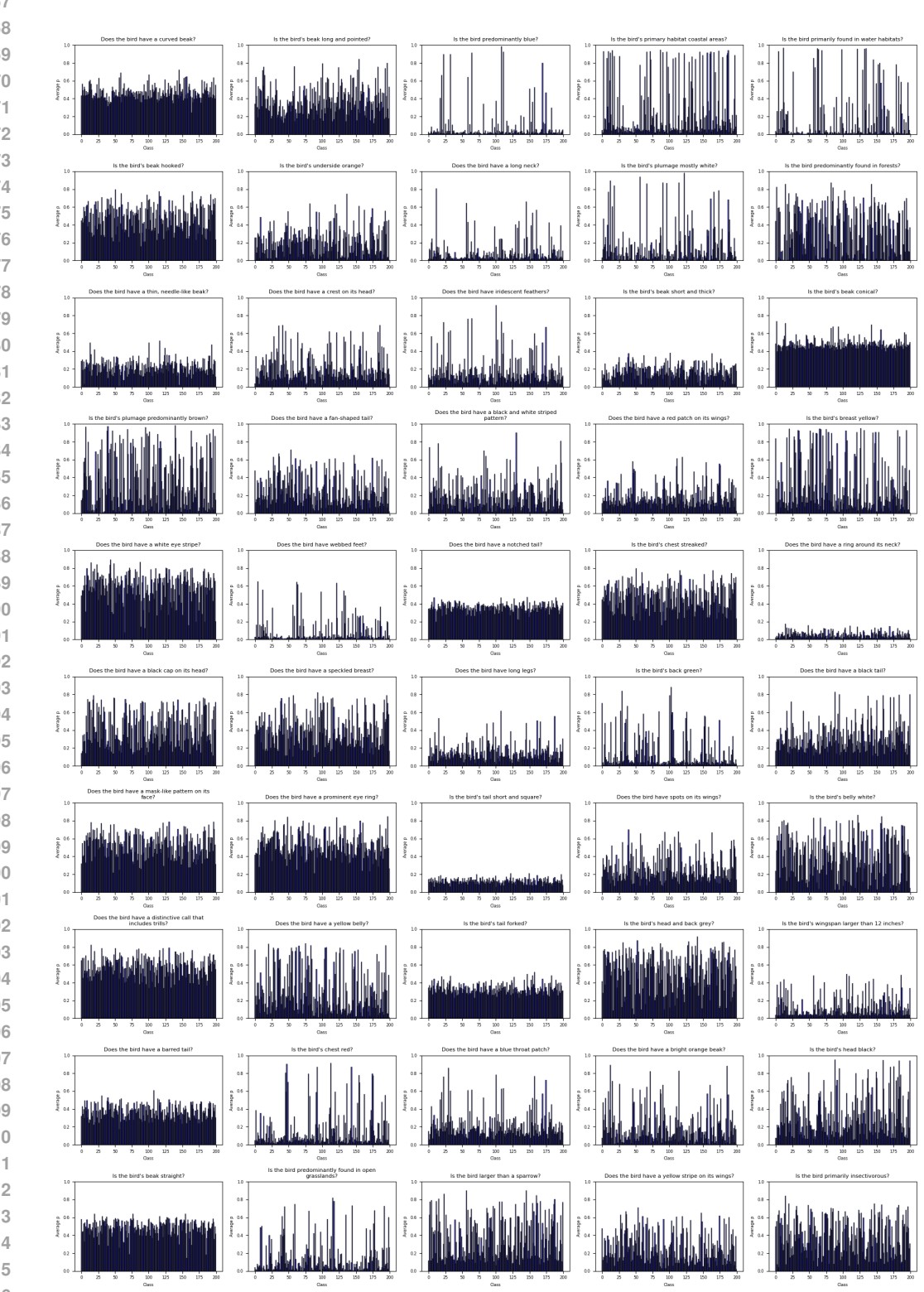

Figure 7: **Visualization of the average logit distribution for the multi-aspect of the CUB200.**

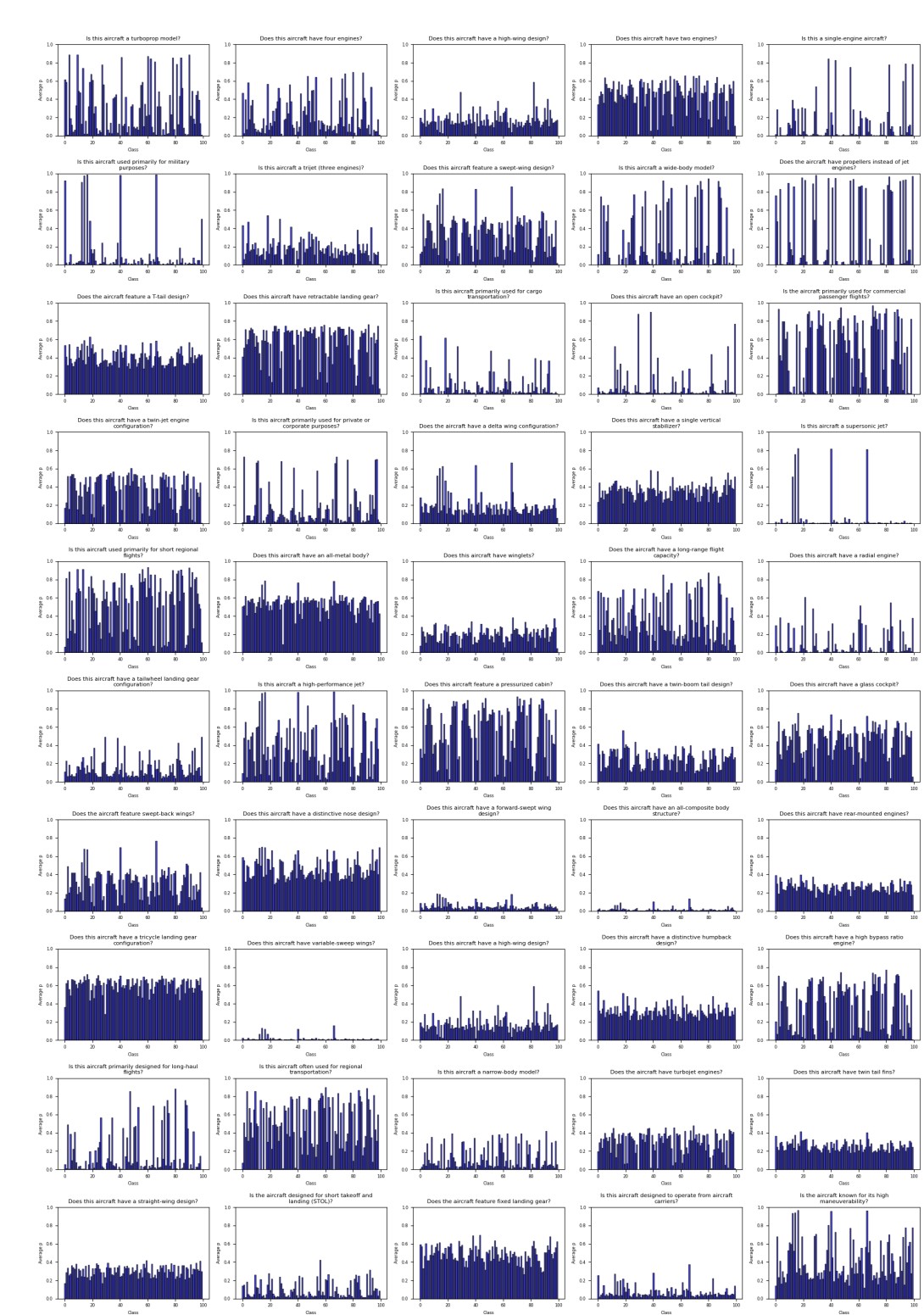

Figure 8: **Visualization of the average logit distribution for the multi-aspect of the FGVC-Aircraft.**

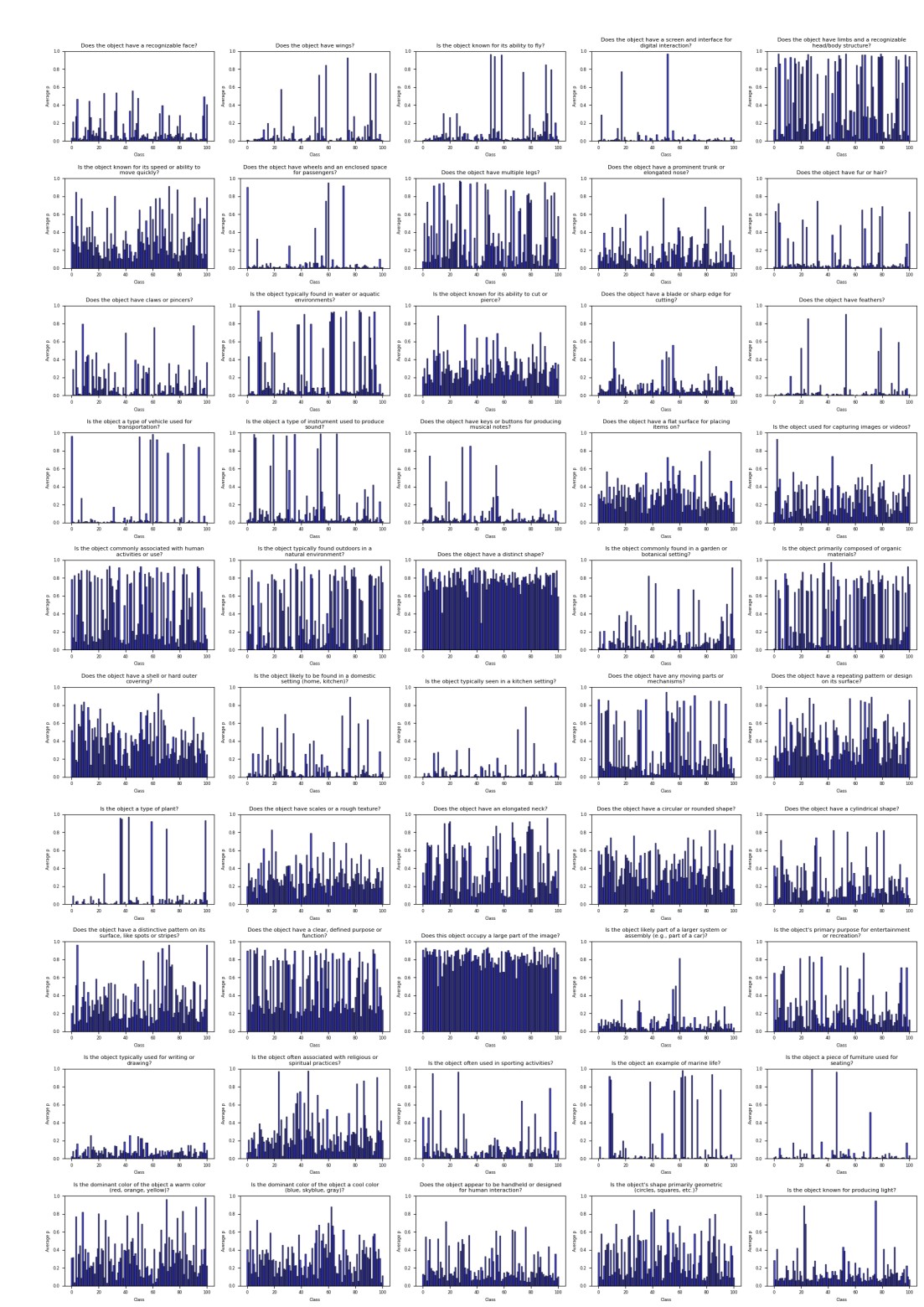

Figure 9: **Visualization of the average logit distribution for the multi-aspect of the Caltech101.**

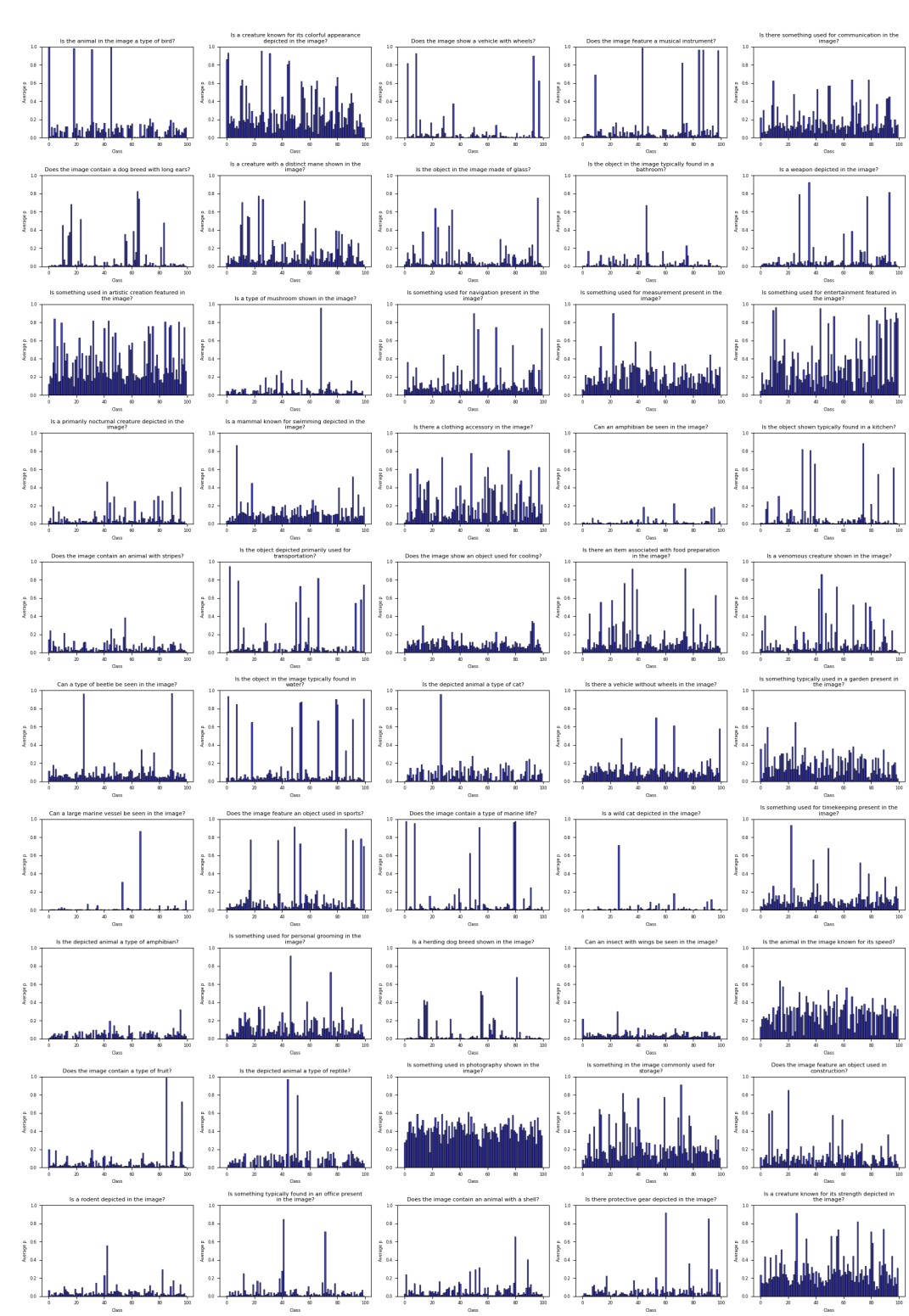

Figure 10: **Visualization of the average logit distribution for the multi-aspect of the Mini-ImageNet.**

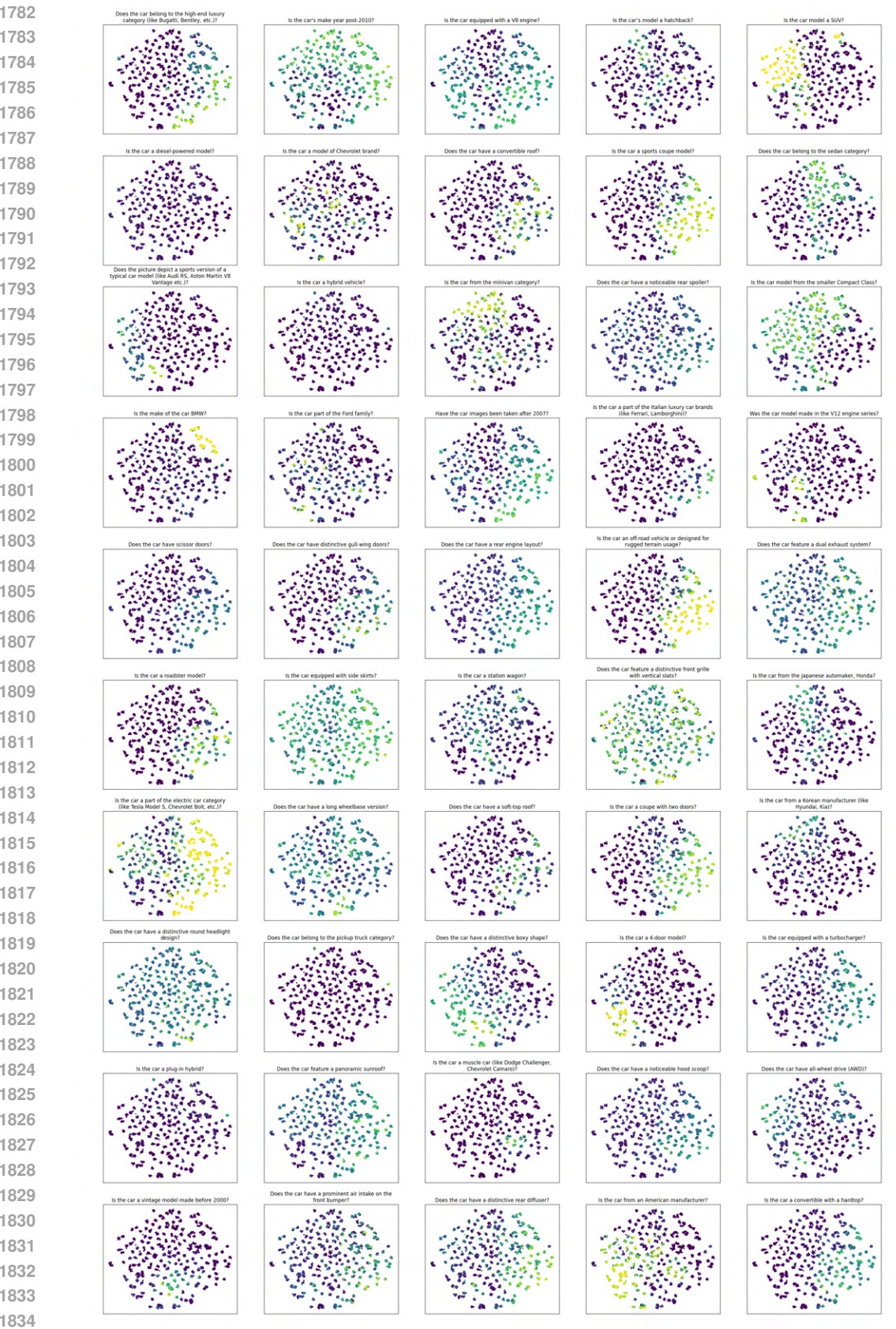

Figure 11: **Visualization of ground truth t-SNE embeddings for the multi-aspect of the StanfordCars.**

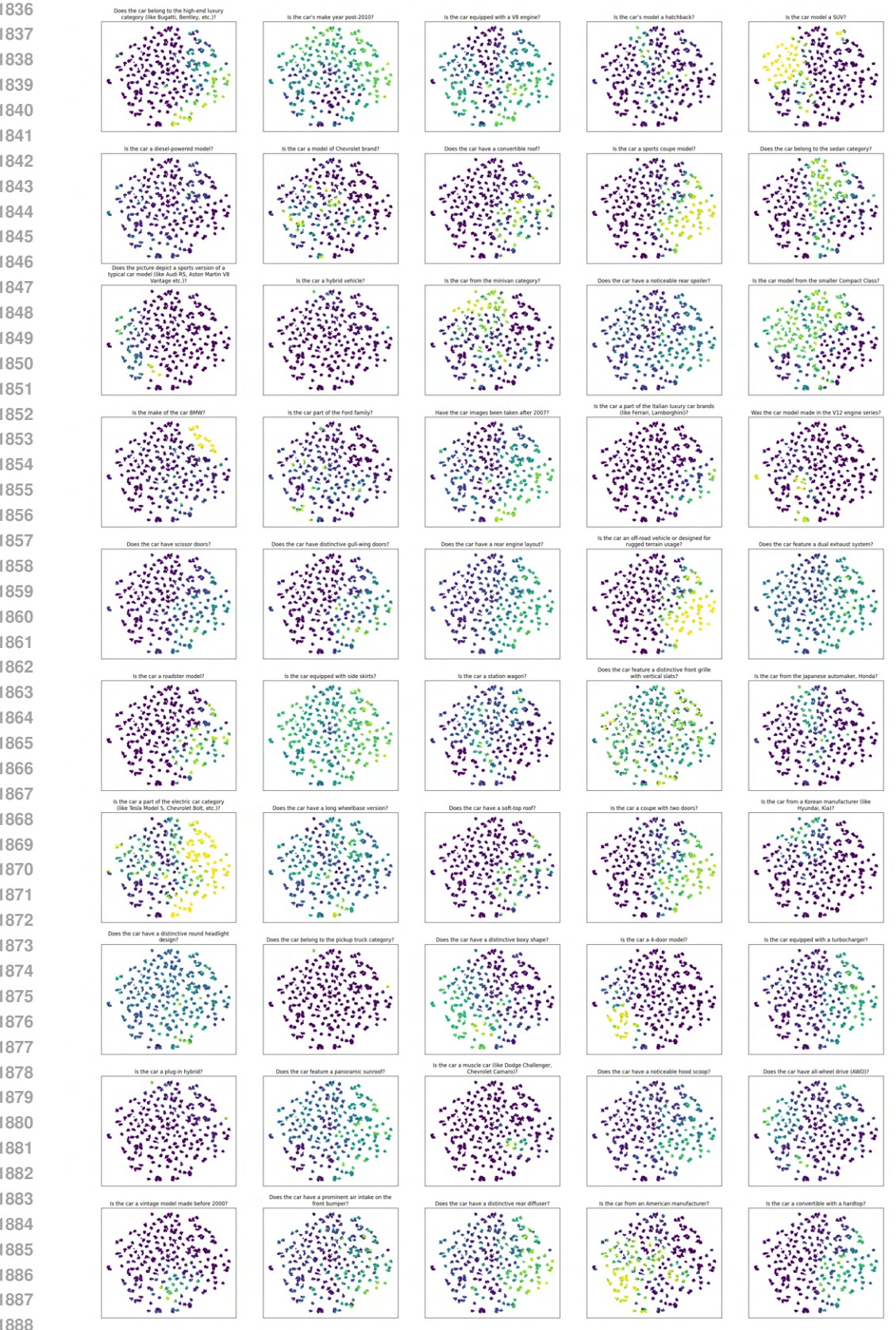

Figure 12: **Visualization of predicted result t-SNE embeddings for the multi-aspect of the StanfordCars.**

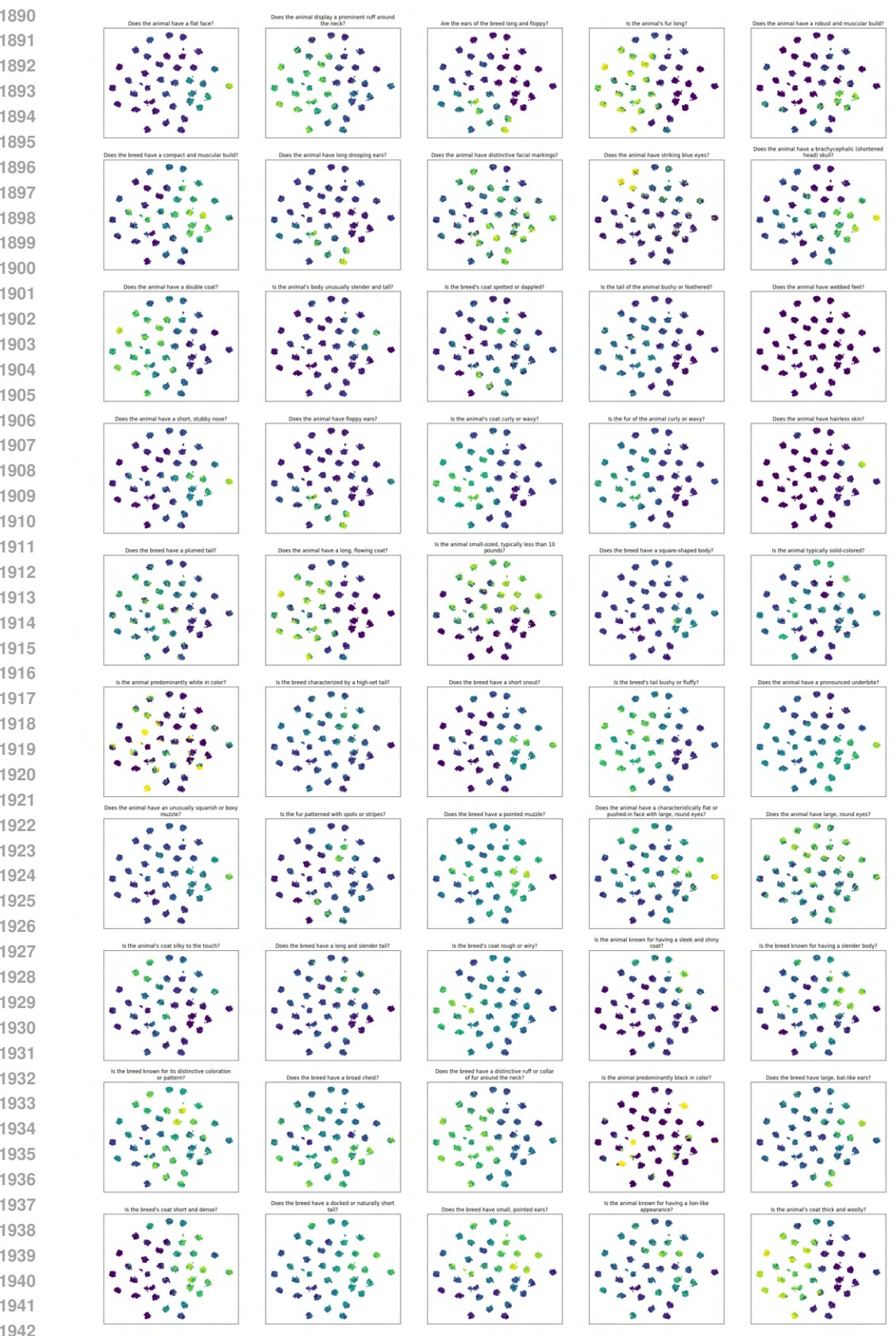

Figure 13: **Visualization of ground truth t-SNE embeddings for the multi-aspect of the Oxford-Pets.**

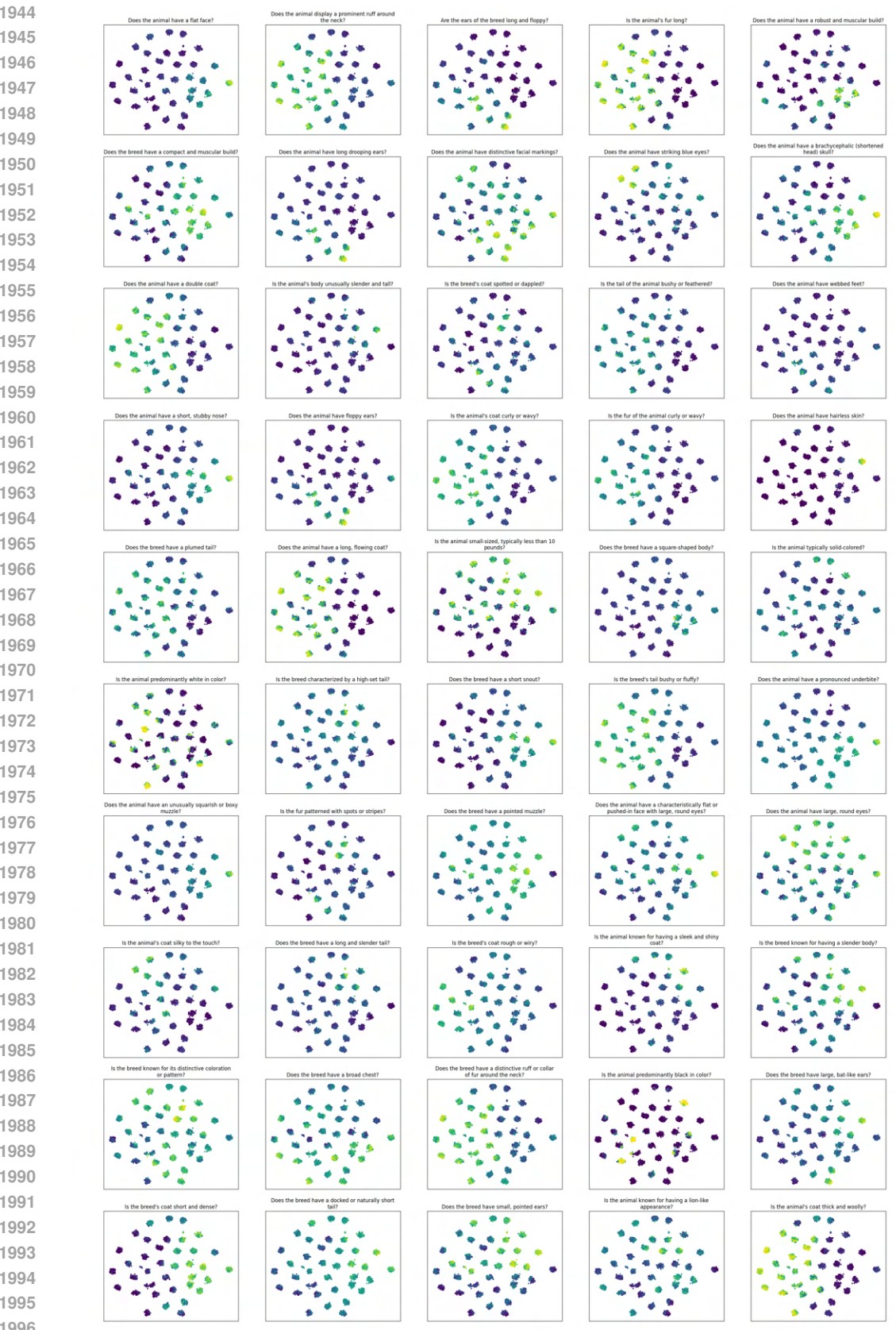

Figure 14: **Visualization of predicted result t-SNE embeddings for the multi-aspect of the Ox-fordPets.**

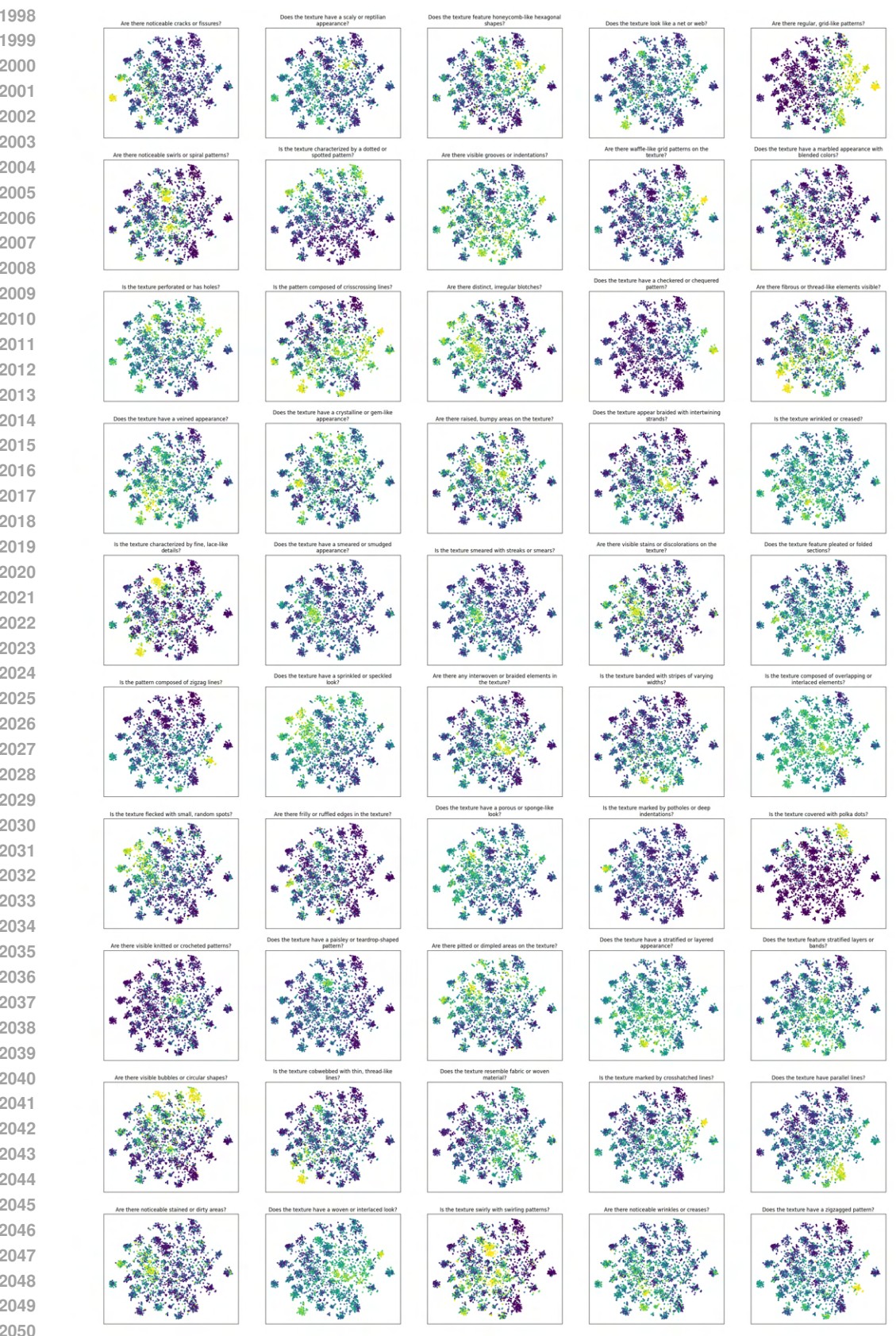

Figure 15: **Visualization of ground truth t-SNE embeddings for the multi-aspect of the StanfordCars.**

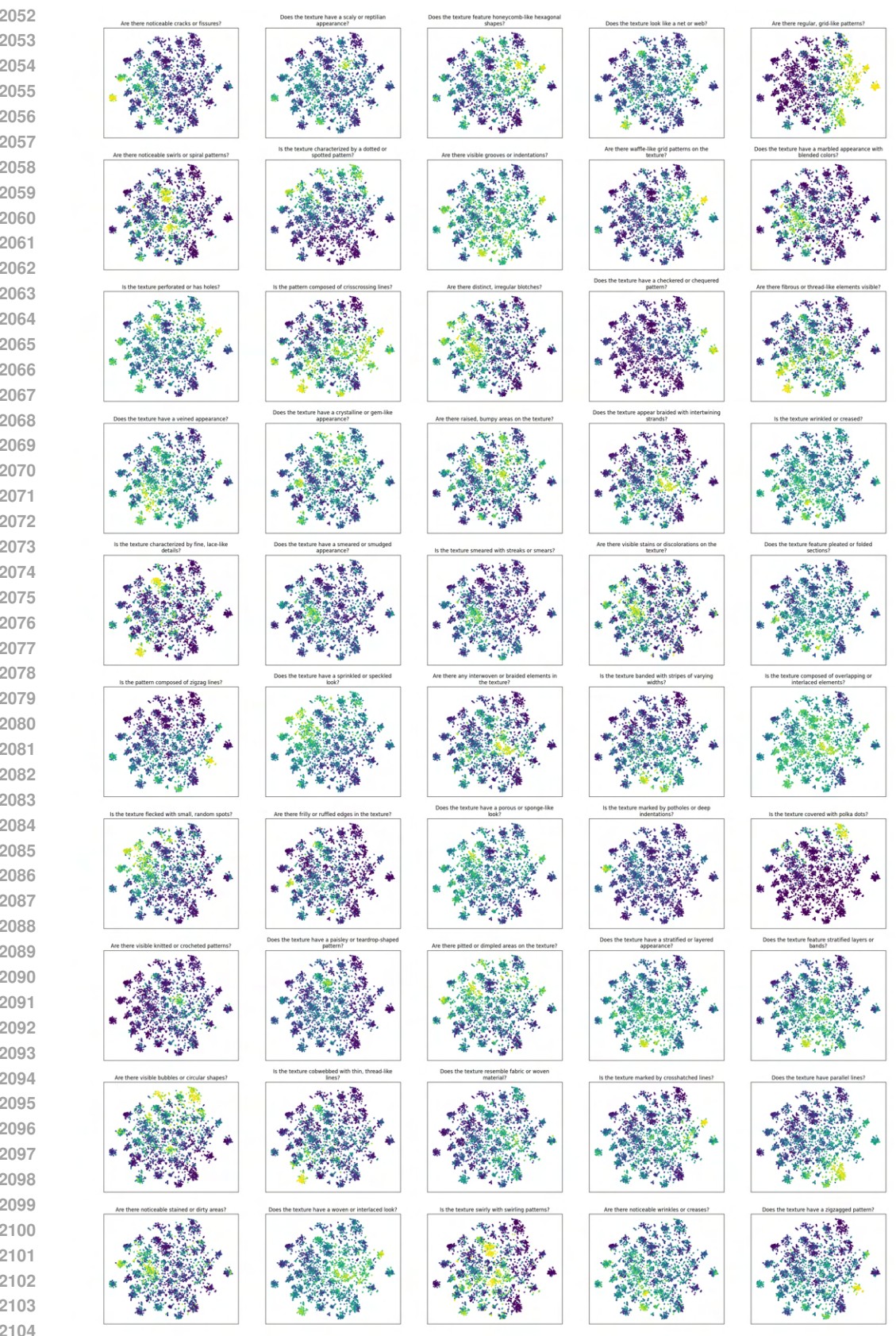

Figure 16: **Visualization of predicted result t-SNE embeddings for the multi-aspect of the Stanford Cars.**

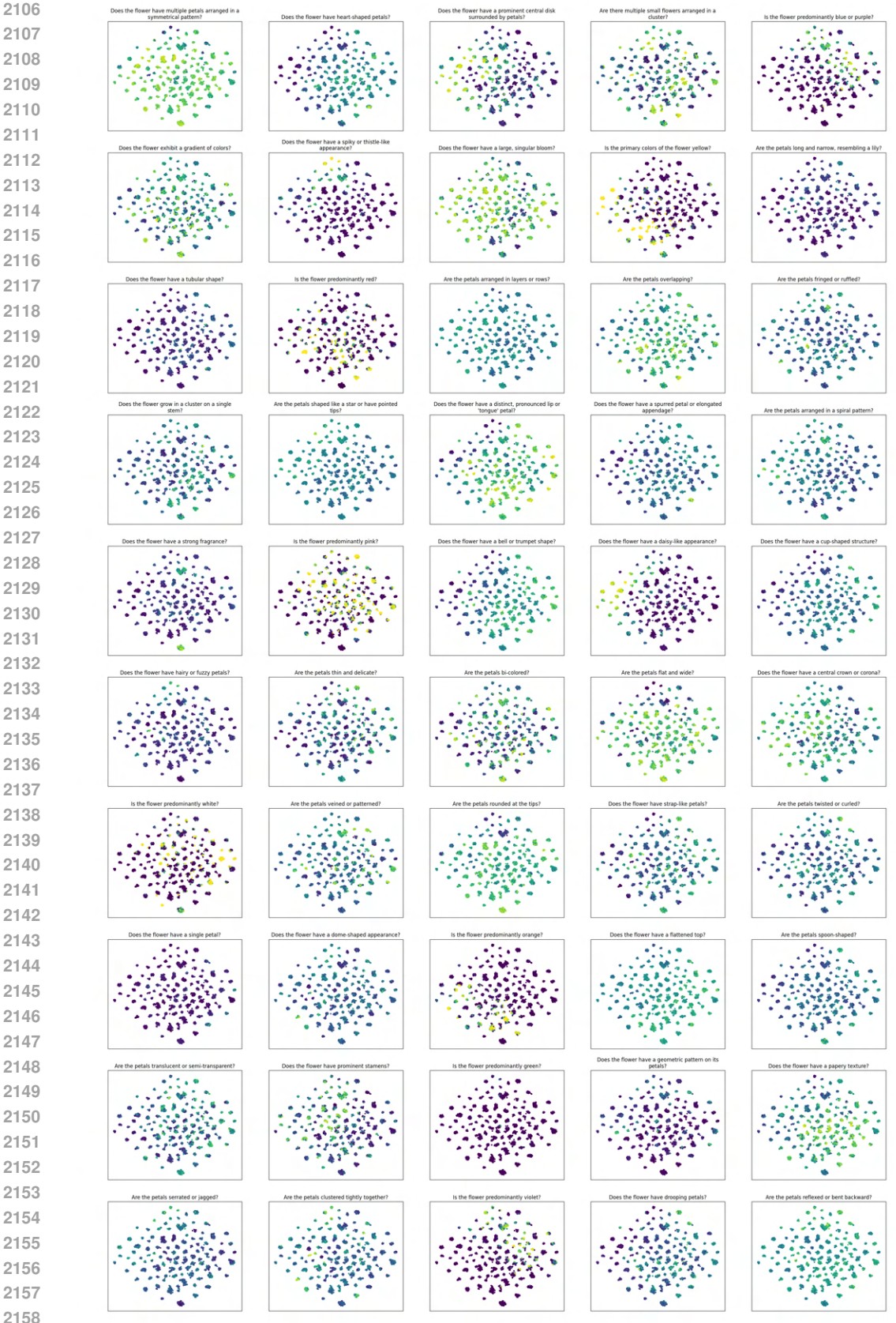

Figure 17: **Visualization of ground truth t-SNE embeddings for the multi-aspect of the DTD.**

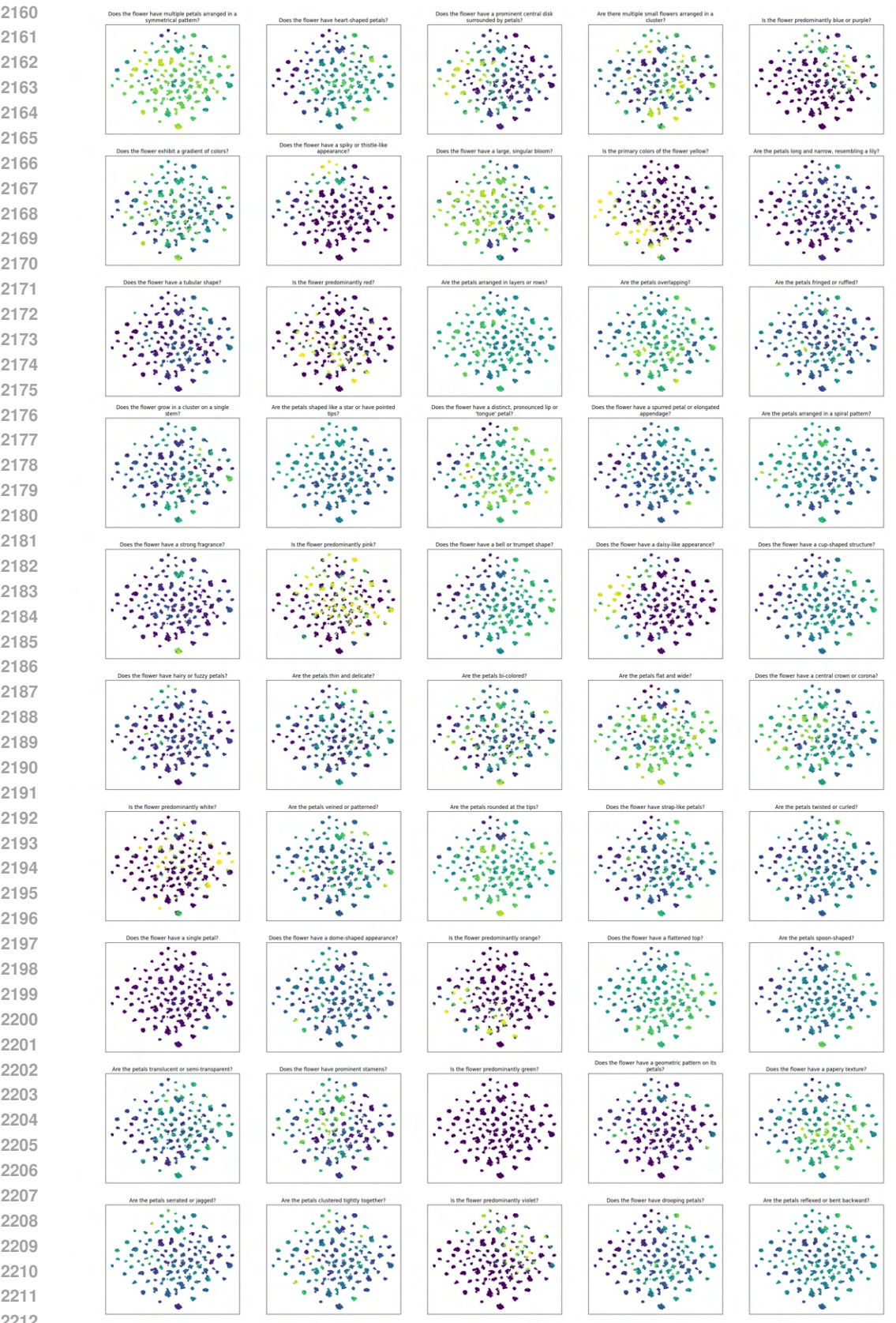

Figure 18: **Visualization of predicted result t-SNE embeddings for the multi-aspect of the DTD.**

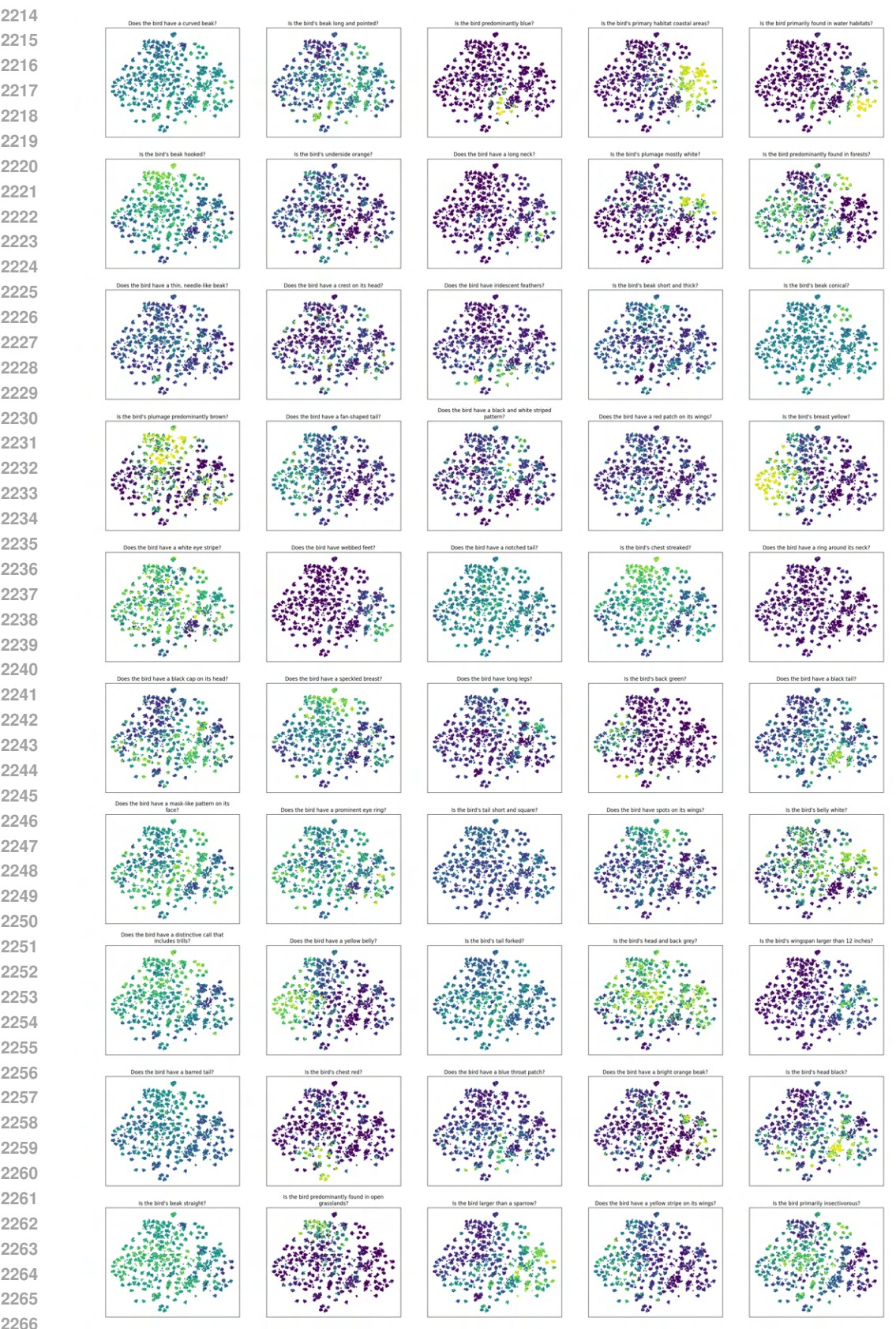

Figure 19: **Visualization of ground truth t-SNE embeddings for the multi-aspect of the CUB200.**

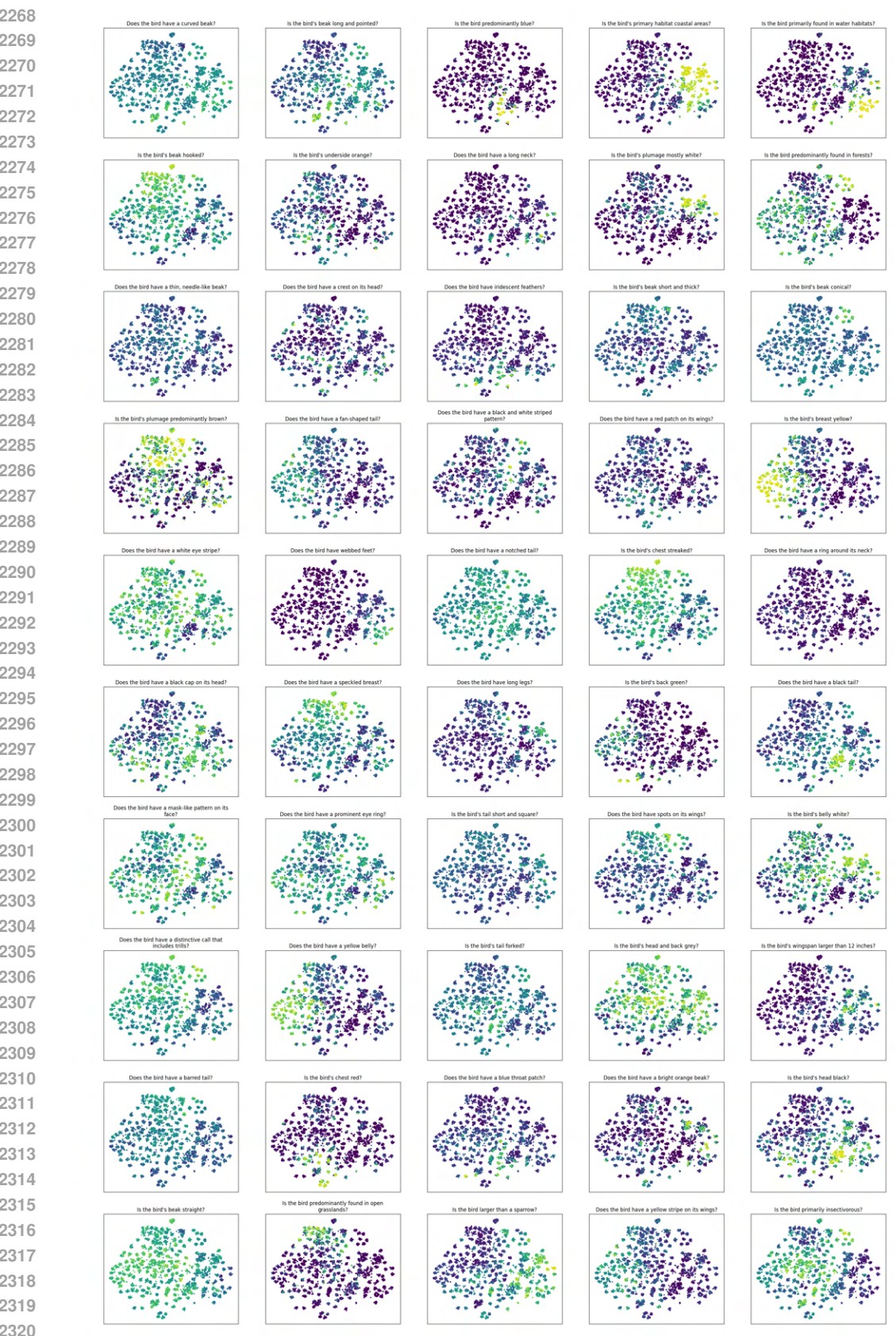

Figure 20: **Visualization of predicted result t-SNE embeddings for the multi-aspect of the CUB200.**

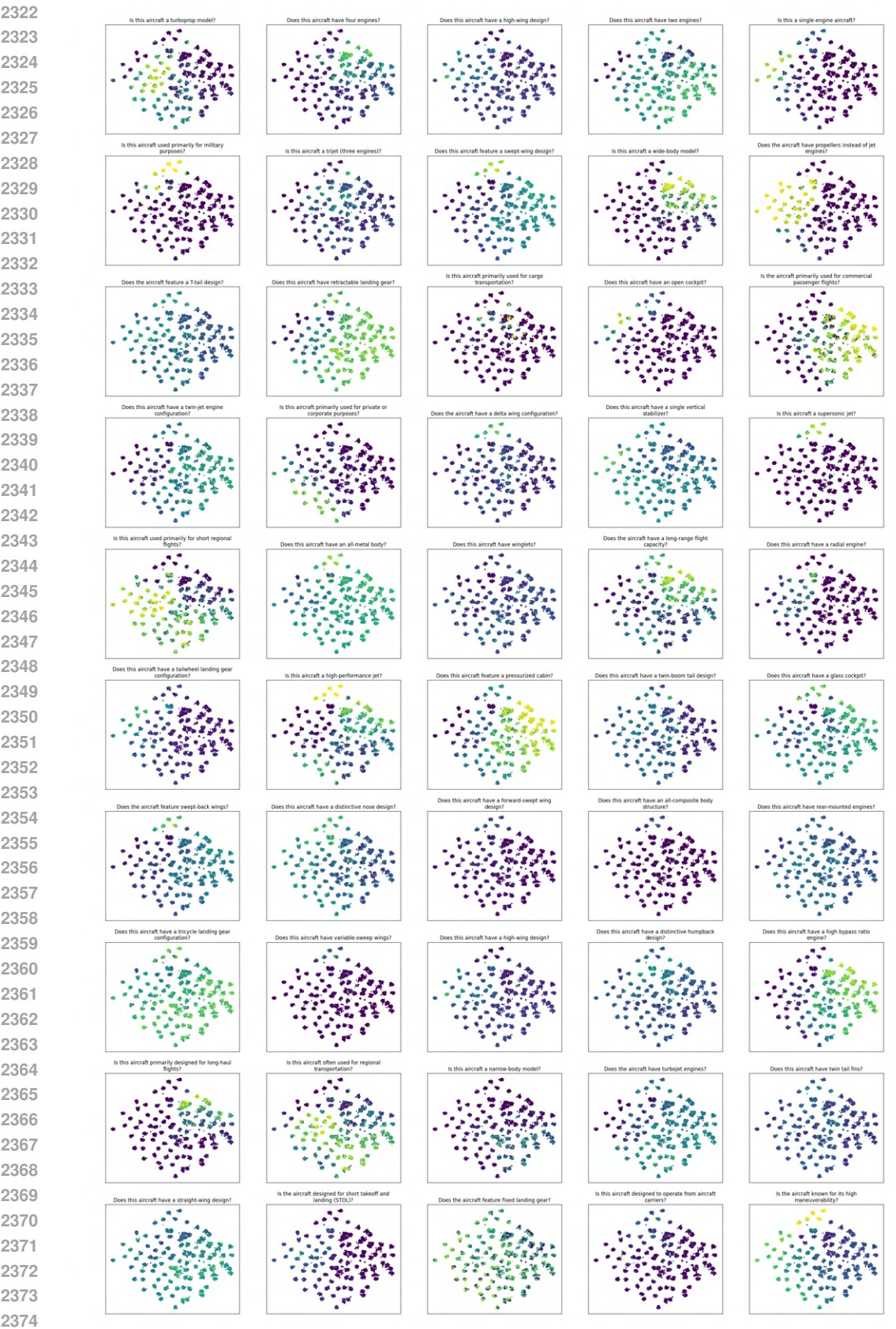

Figure 21: **Visualization of ground truth t-SNE embeddings for the multi-aspect of the FGVC-Aircraft.**

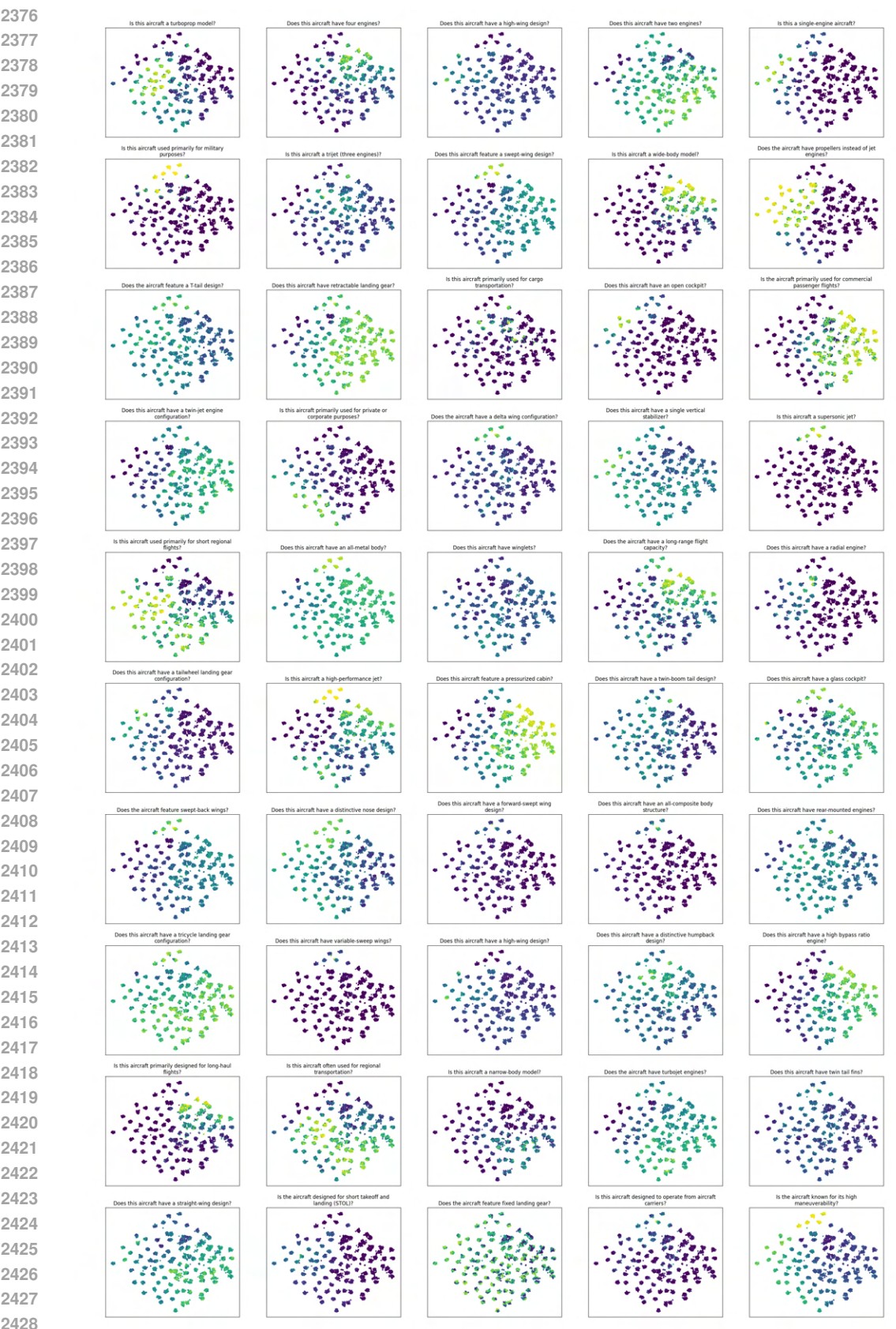

Figure 22: **Visualization of predicted result t-SNE embeddings for the multi-aspect of the FGVC-Aircraft.**

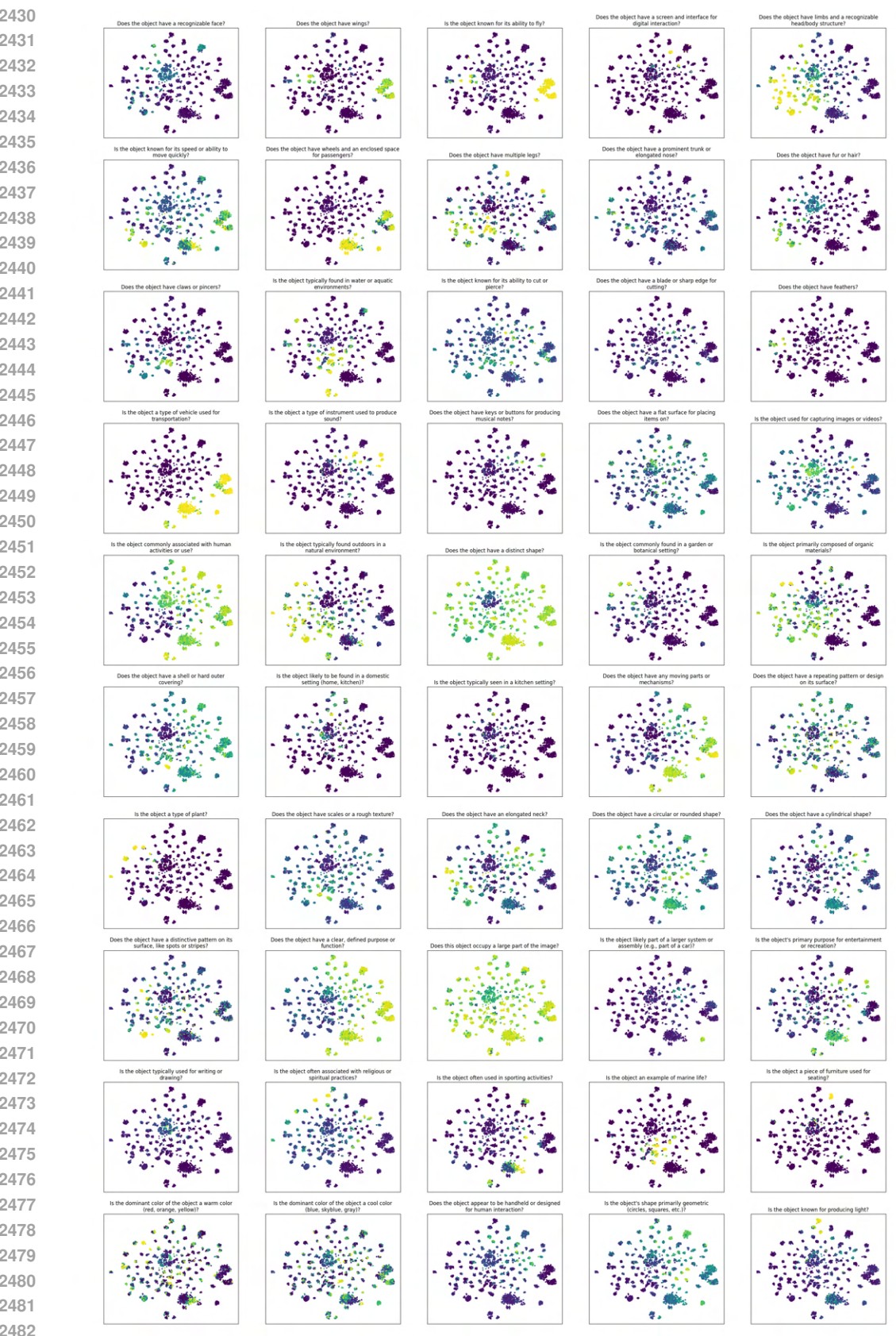

Figure 23: **Visualization of ground truth t-SNE embeddings for the multi-aspect of the Caltech101.**

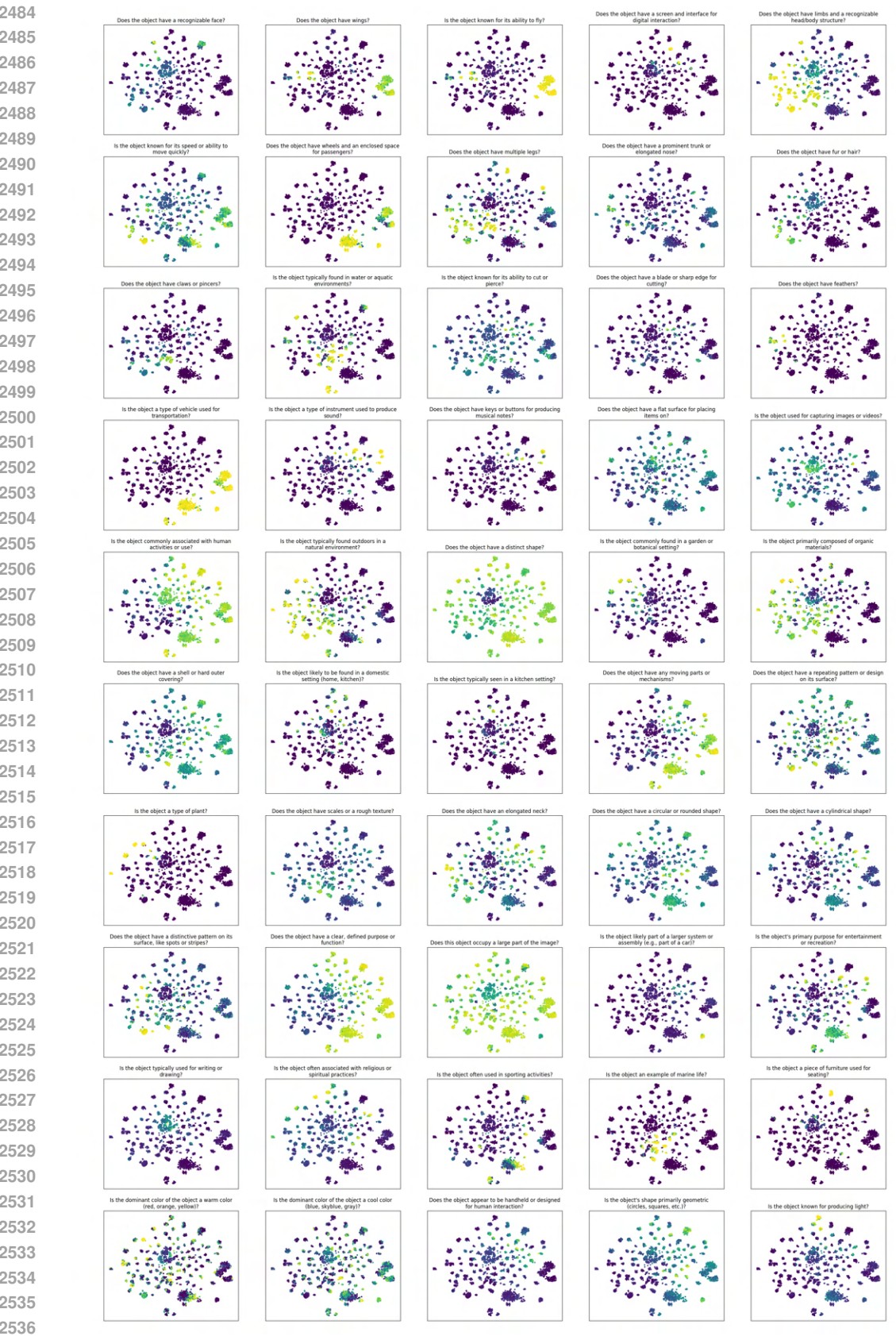

Figure 24: **Visualization of predicted result t-SNE embeddings for the multi-aspect of the Cal-tech101.**

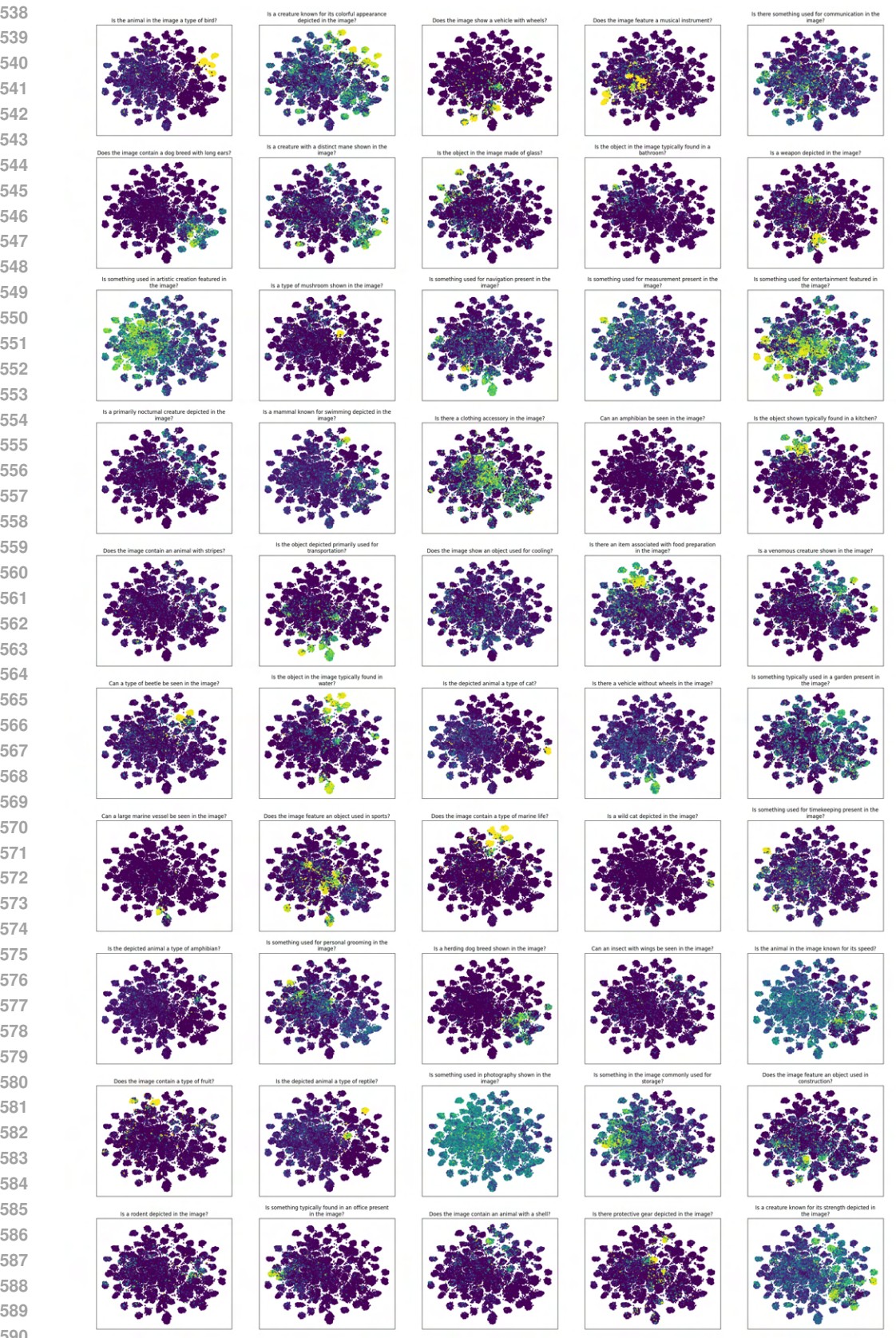

Figure 25: **Visualization of ground truth t-SNE embeddings for the multi-aspect of the Mini-ImageNet.**

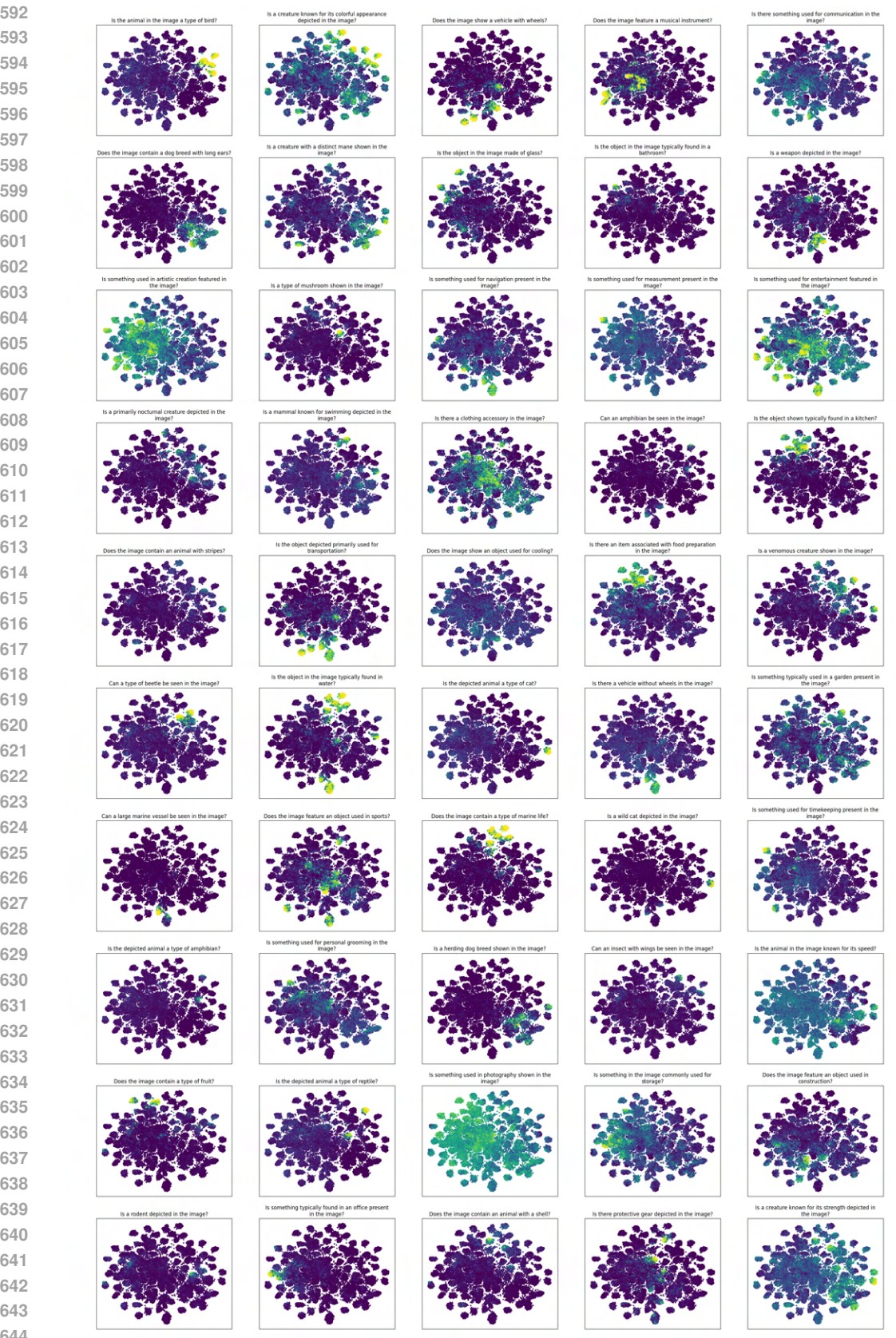

Figure 26: **Visualization of predicted result t-SNE embeddings for the multi-aspect of the Mini-ImageNet.**

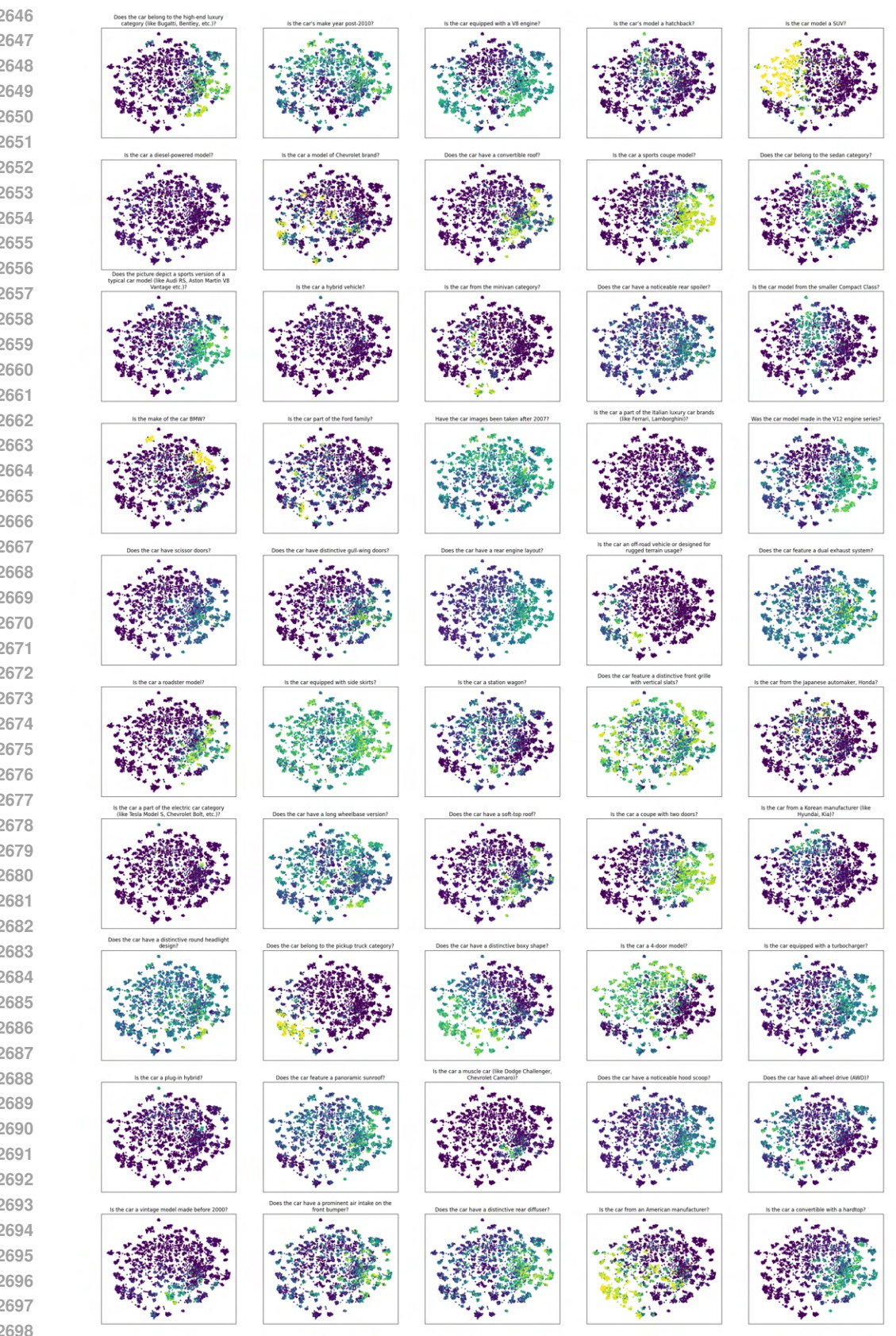

Figure 27: **Visualization of ground truth t-SNE embeddings for the multi-aspect of the StanfordCars testing dataset.**

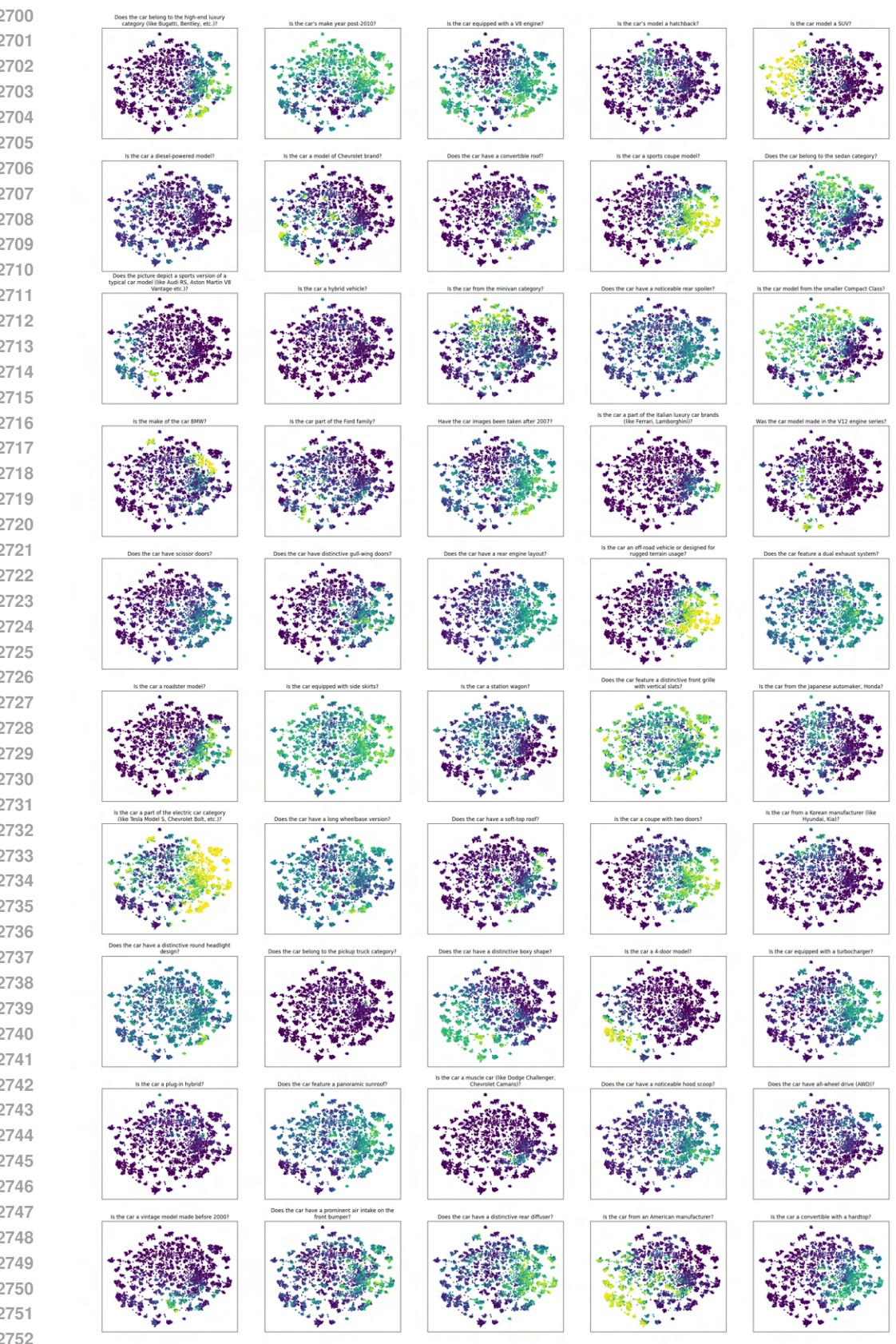

Figure 28: **Visualization of predicted result t-SNE embeddings for the multi-aspect of the StanfordCars testing dataset.**

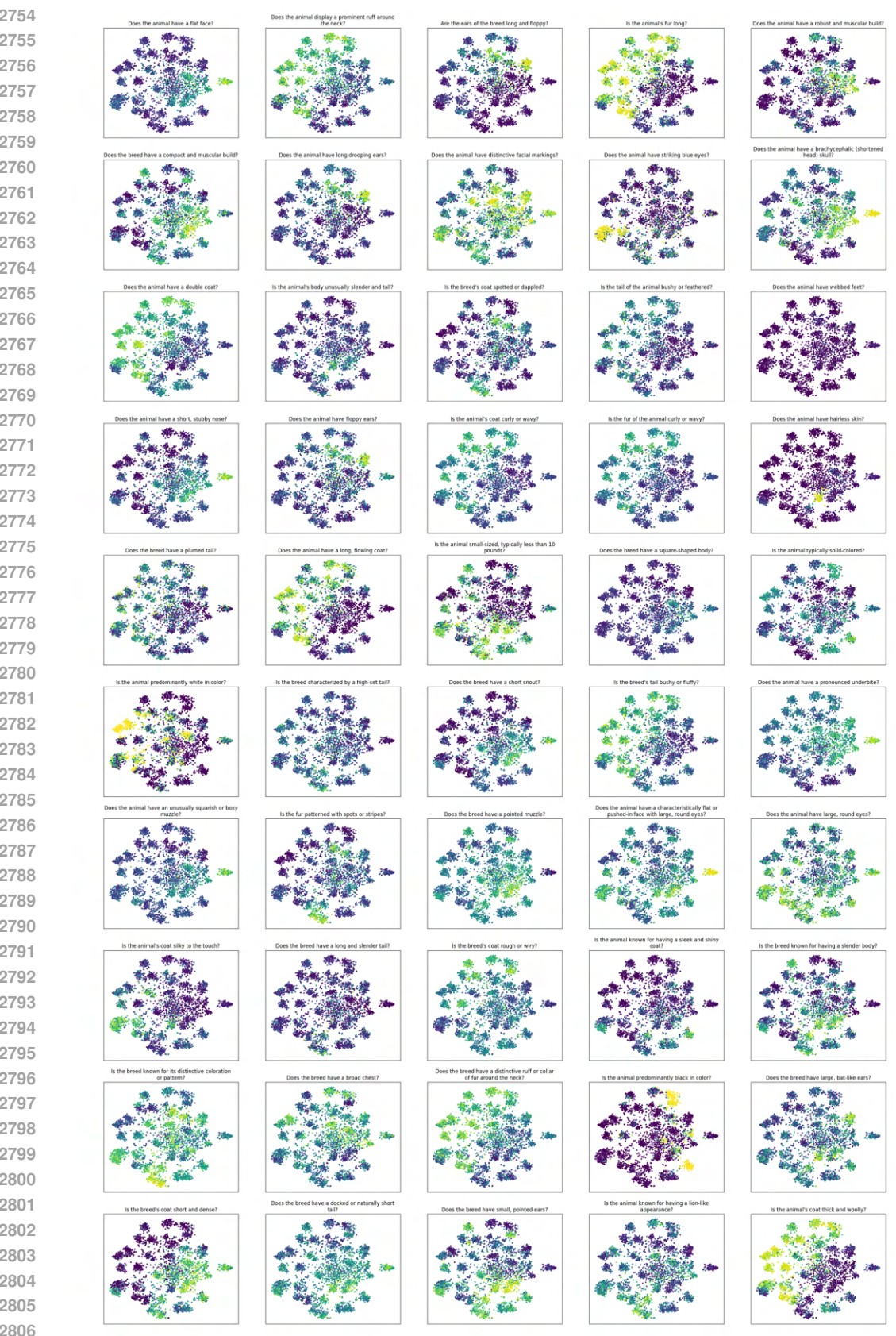

Figure 29: **Visualization of ground truth t-SNE embeddings for the multi-aspect of the Oxford-Pets testing dataset.**

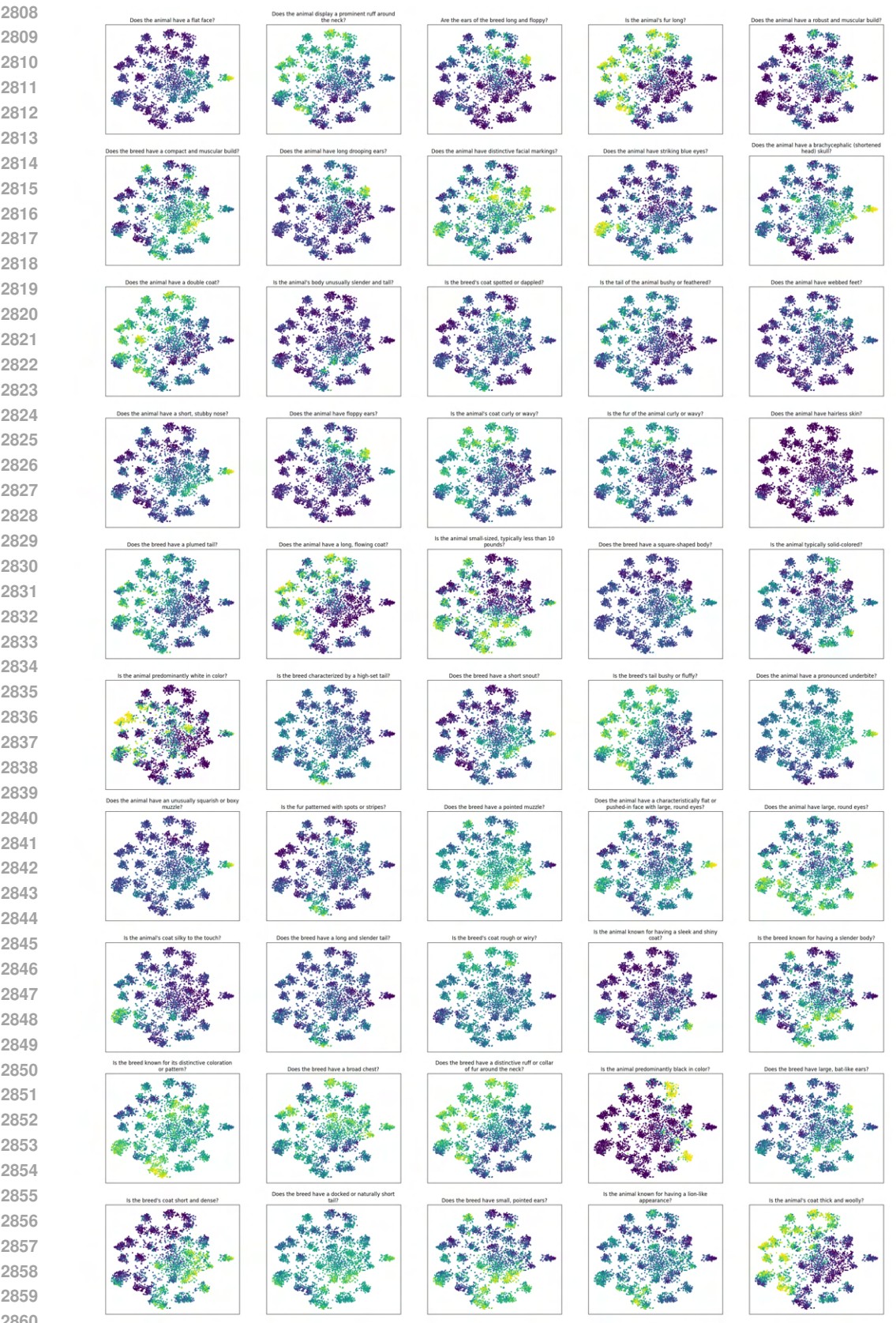

Figure 30: **Visualization of predicted result t-SNE embeddings for the multi-aspect of the Ox-fordPets testing dataset.**

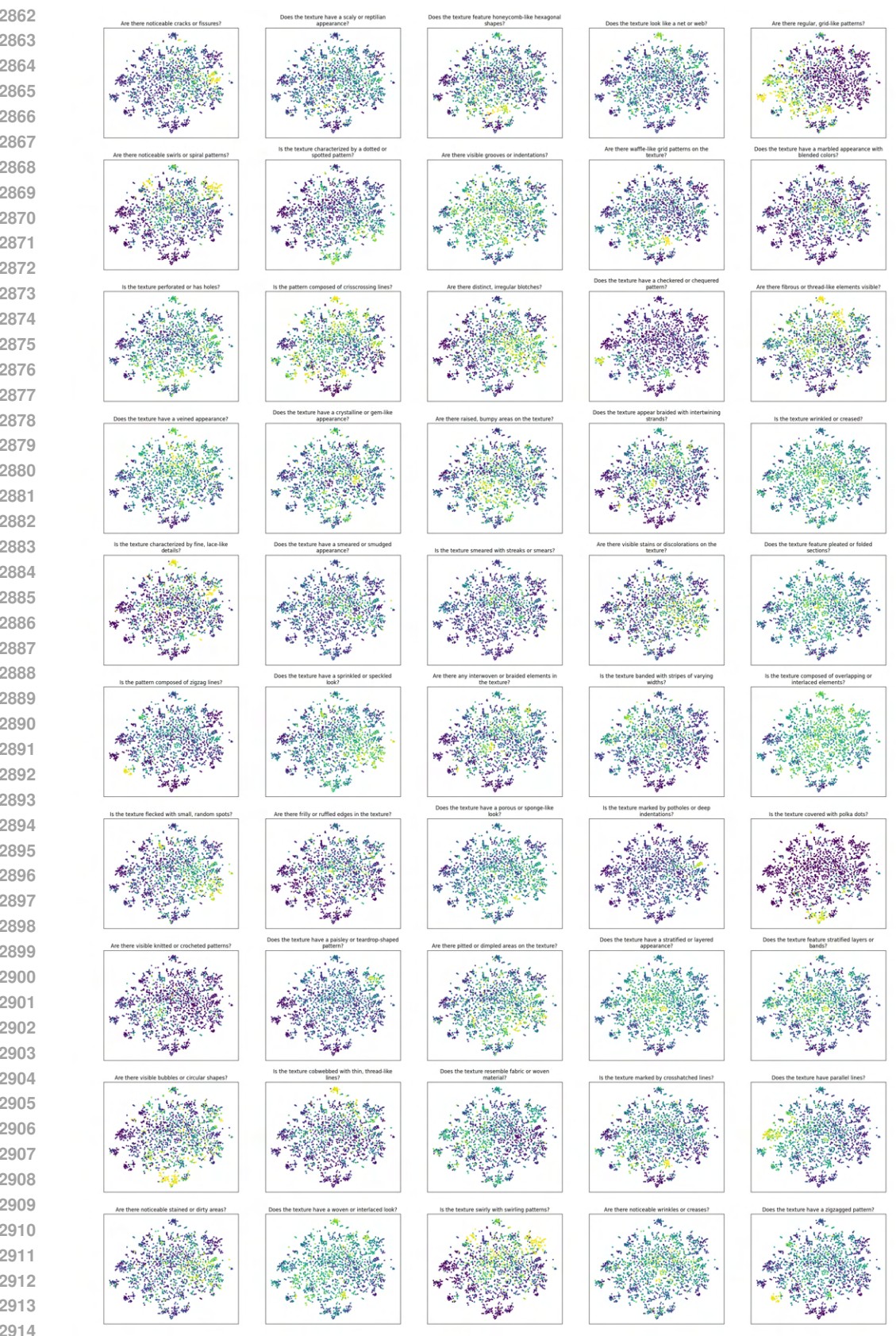

Figure 31: **Visualization of ground truth t-SNE embeddings for the multi-aspect of the DTD testing dataset.**

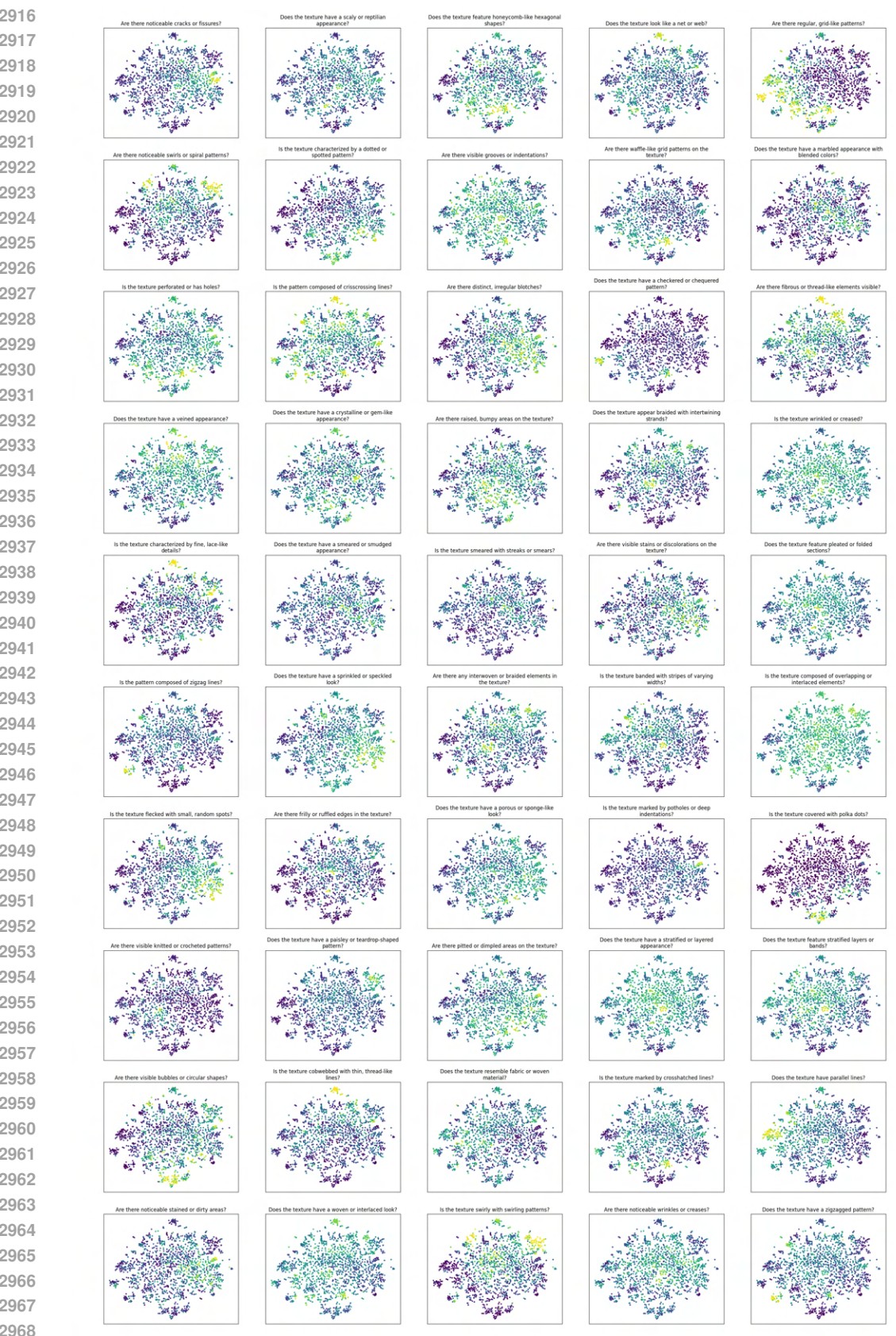

Figure 32: **Visualization of predicted result t-SNE embeddings for the multi-aspect of the DTD testing dataset.**

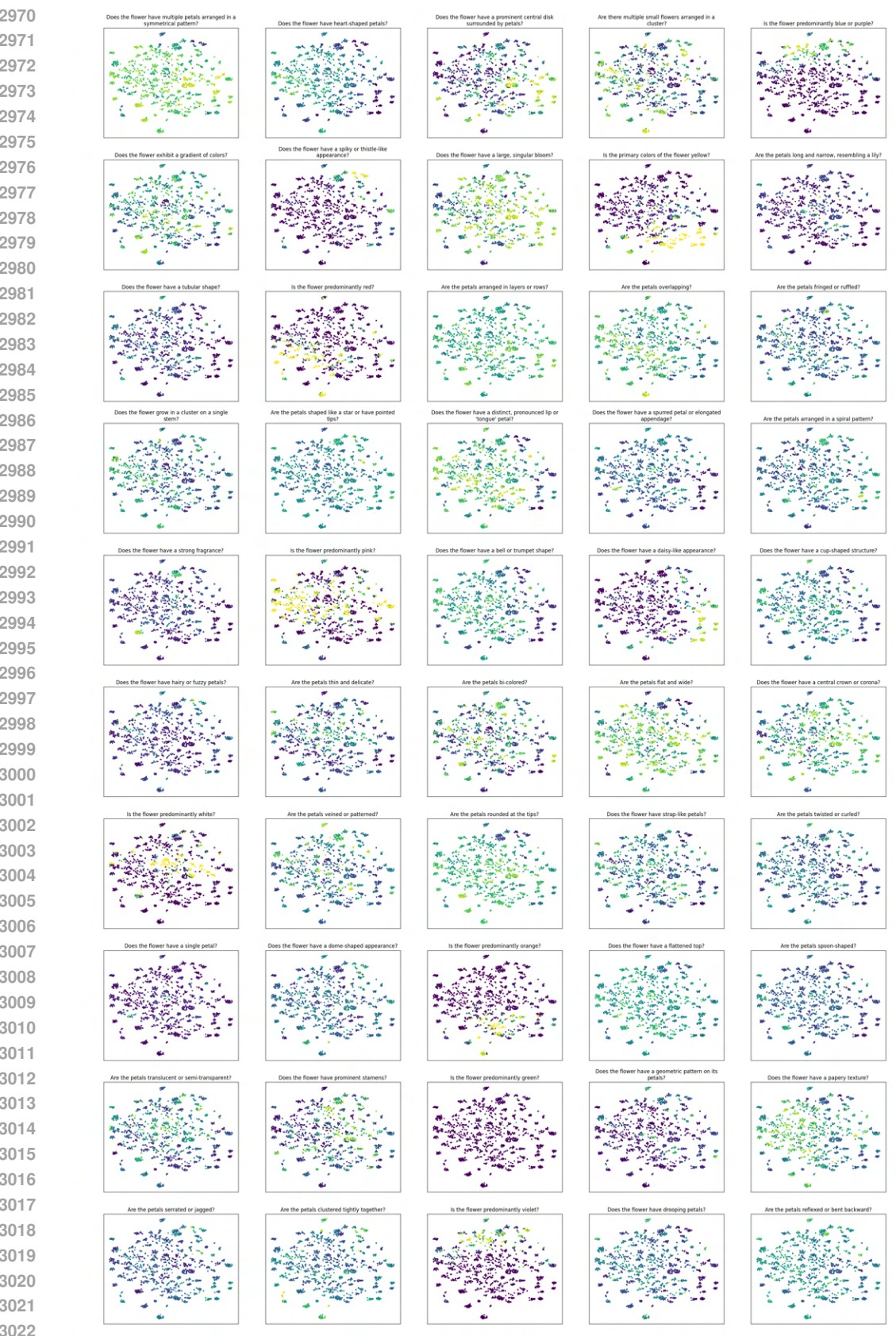

Figure 33: **Visualization of ground truth t-SNE embeddings for the multi-aspect of the 102Flowers testing dataset.**

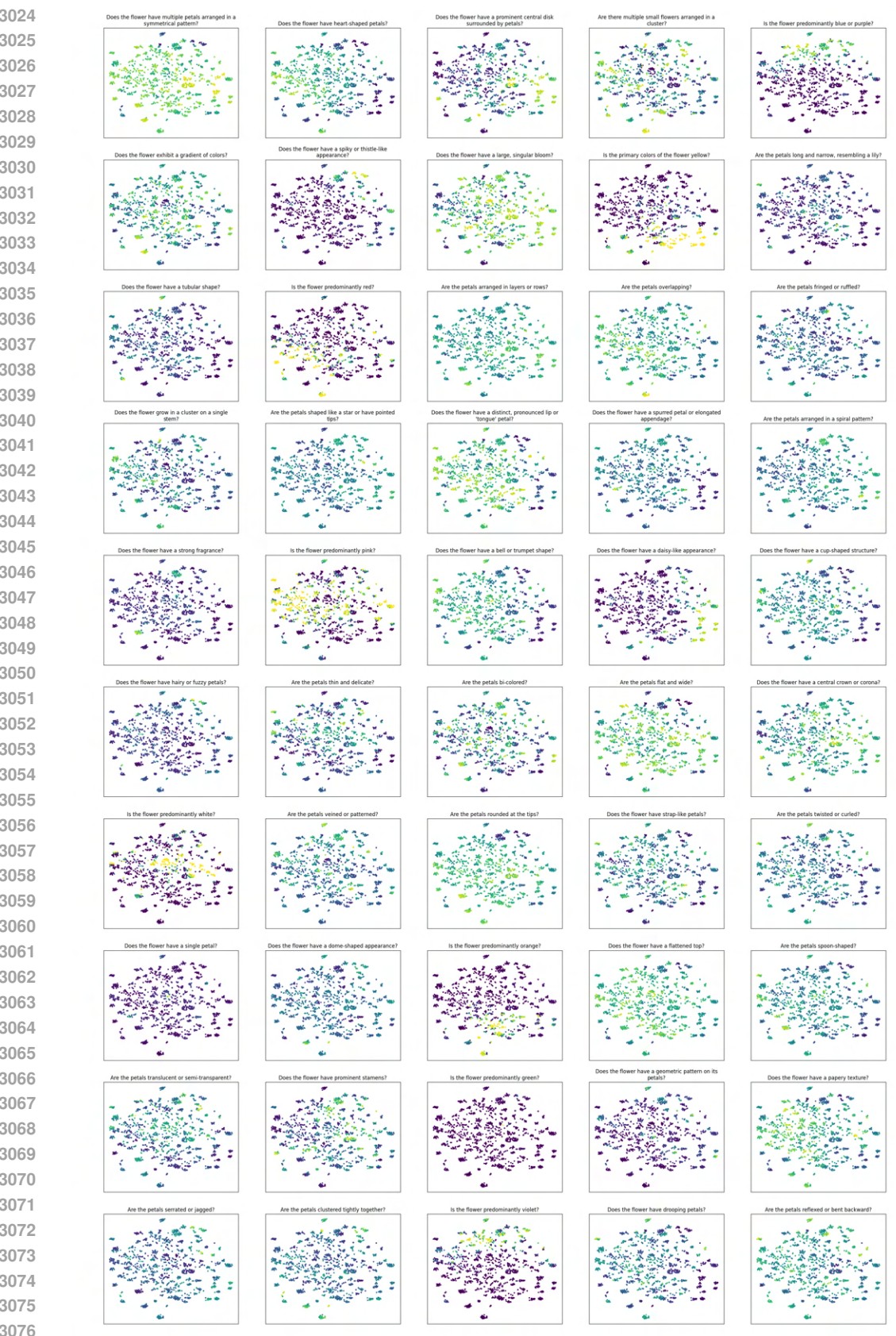

Figure 34: **Visualization of predicted result t-SNE embeddings for the multi-aspect of the 102Flowers testing dataset.**

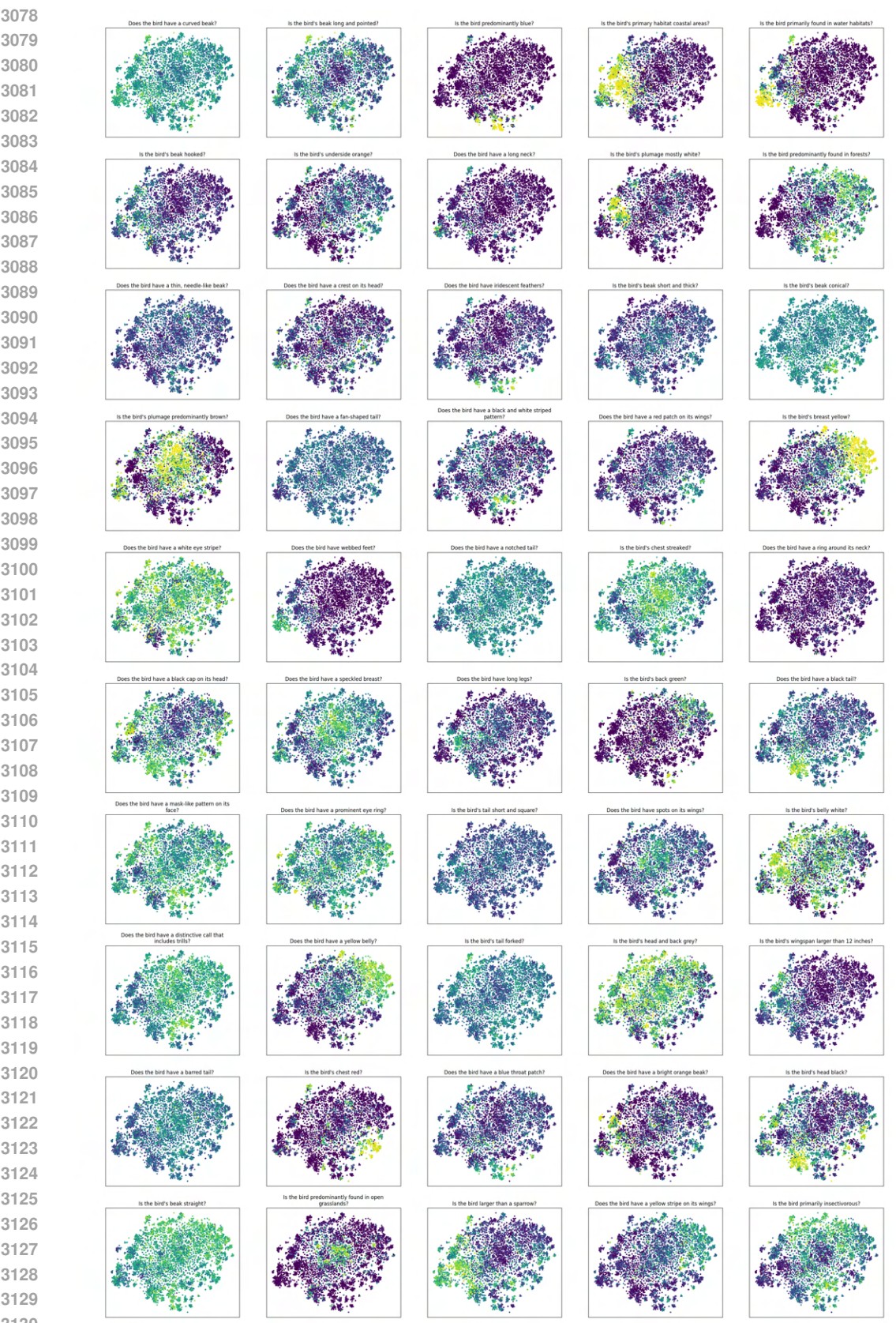

Figure 35: **Visualization of ground truth t-SNE embeddings for the multi-aspect of the CUB200 testing dataset.**

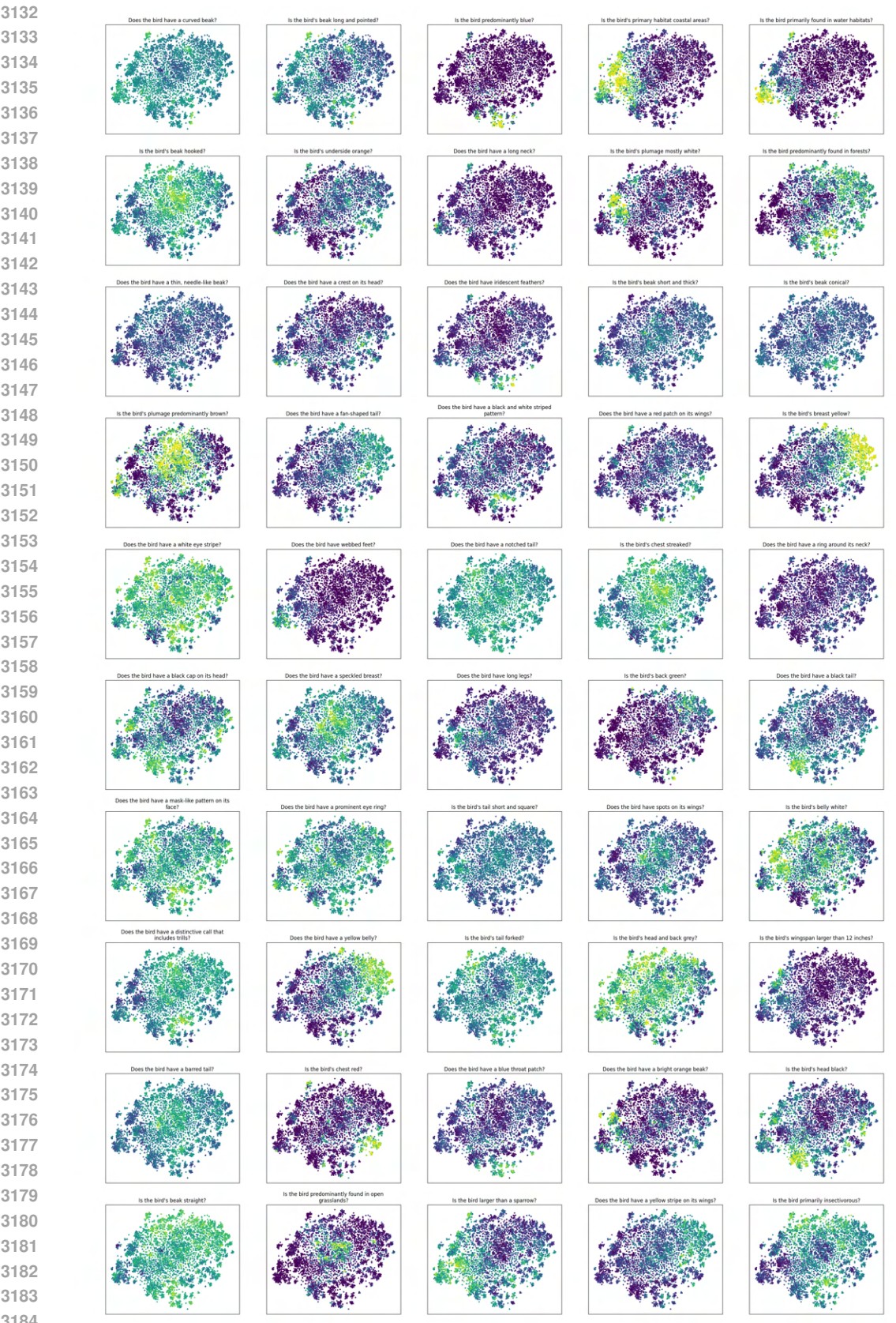

Figure 36: **Visualization of predicted result t-SNE embeddings for the multi-aspect of the CUB200 testing dataset.**

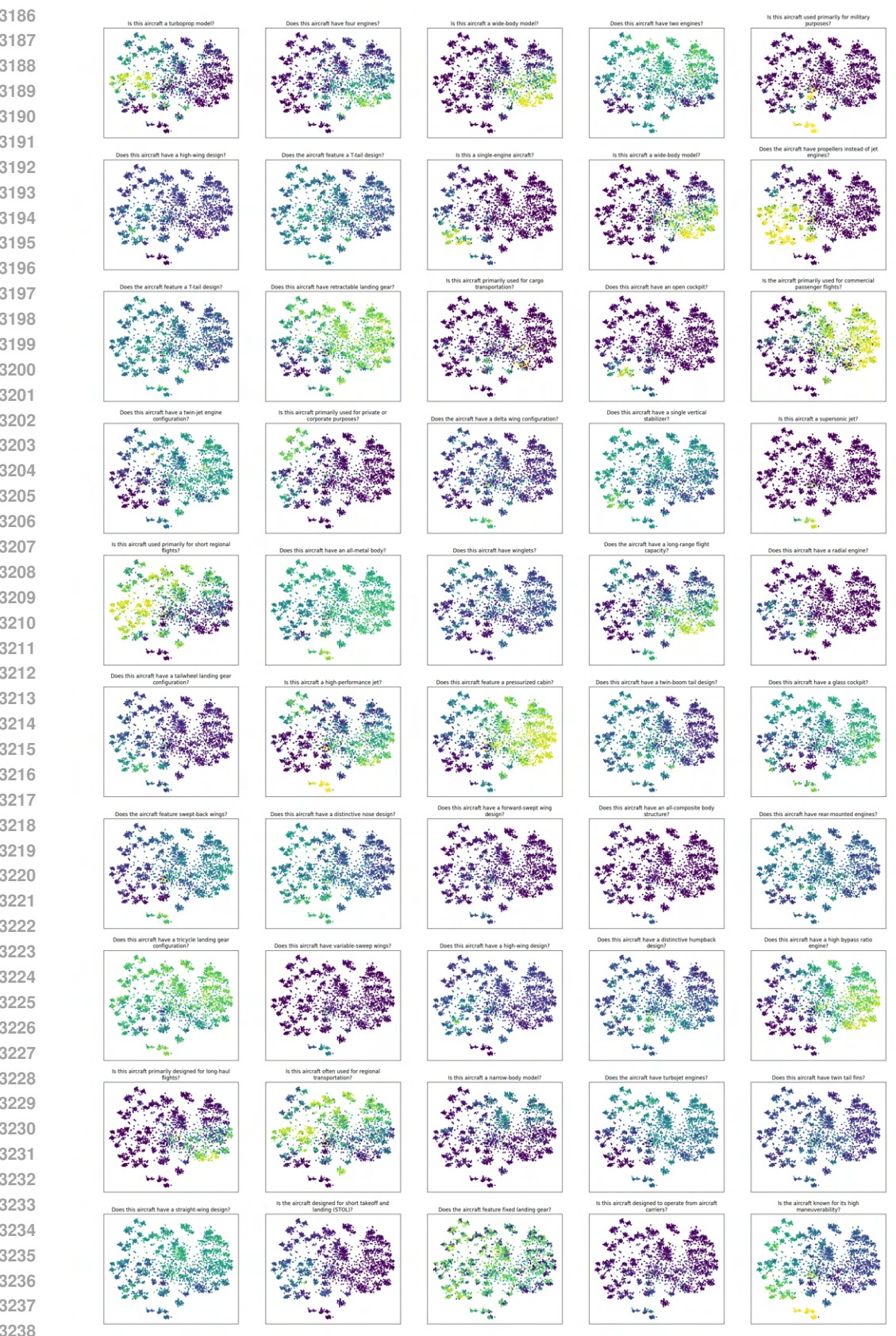

Figure 37: **Visualization of ground truth t-SNE embeddings for the multi-aspect of the FGVC-Aircraft testing dataset.**

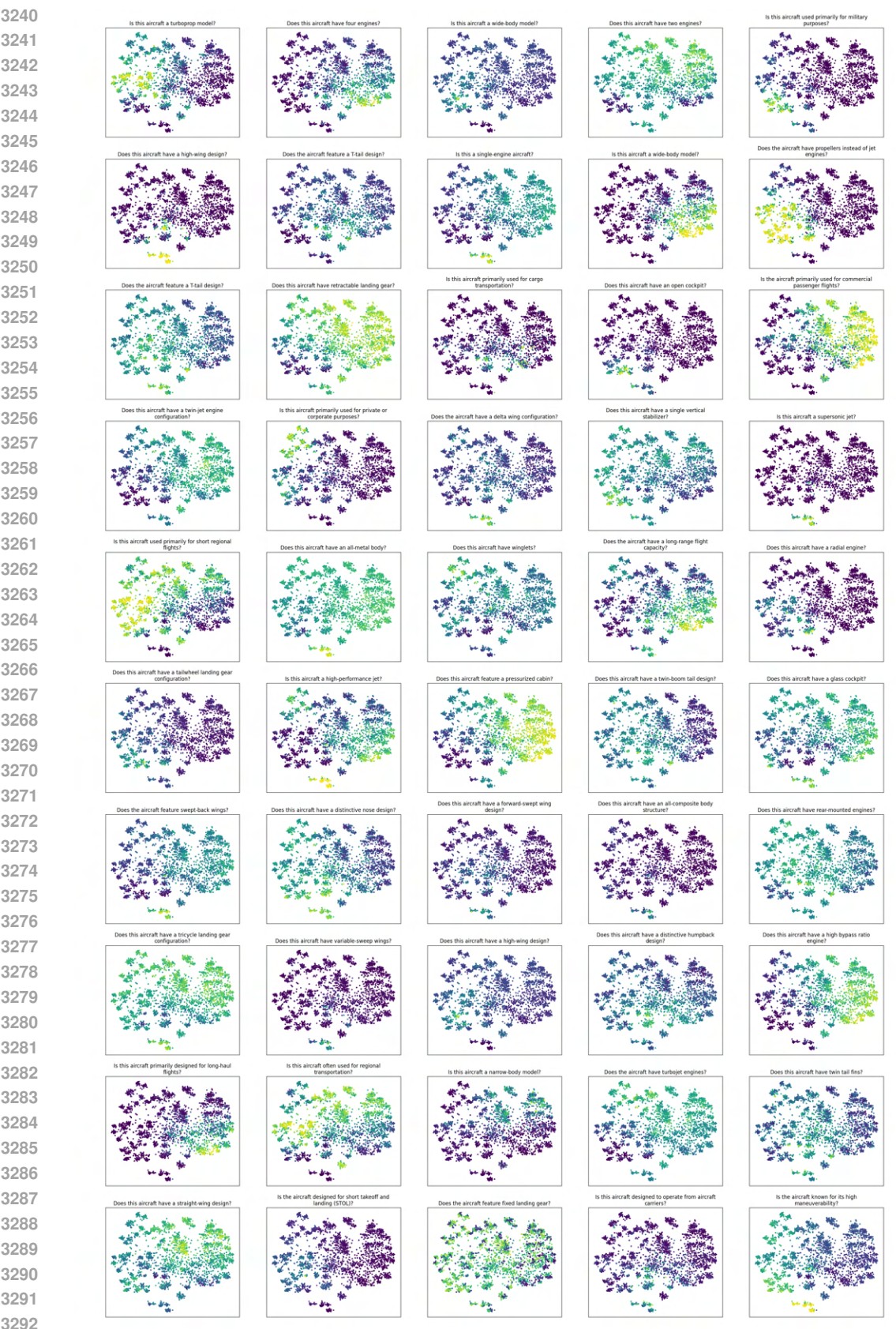

Figure 38: **Visualization of predicted result t-SNE embeddings for the multi-aspect of the FGVC-Aircraft testing dataset.**

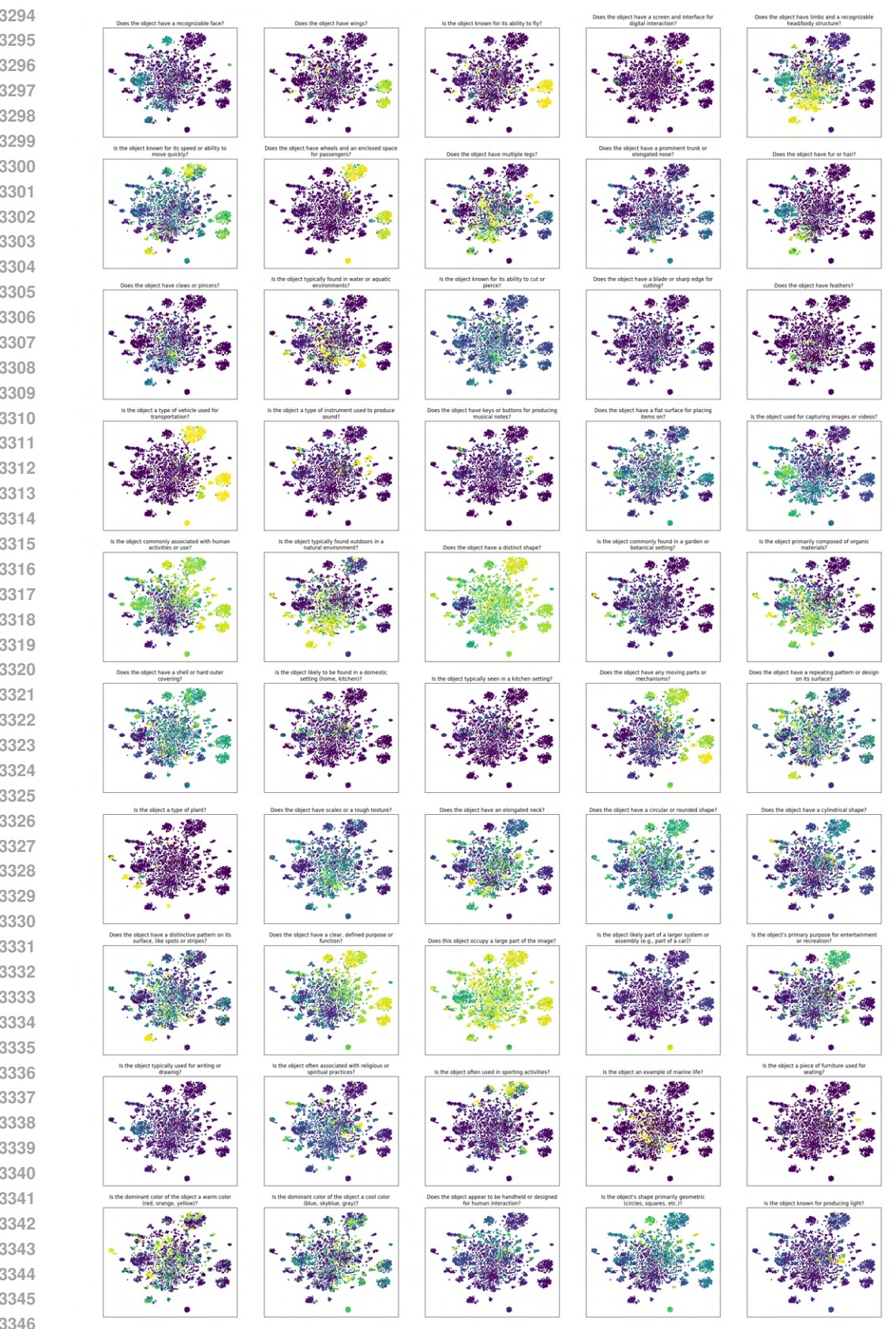

Figure 39: **Visualization of ground truth t-SNE embeddings for the multi-aspect of the Caltech101 testing dataset.**

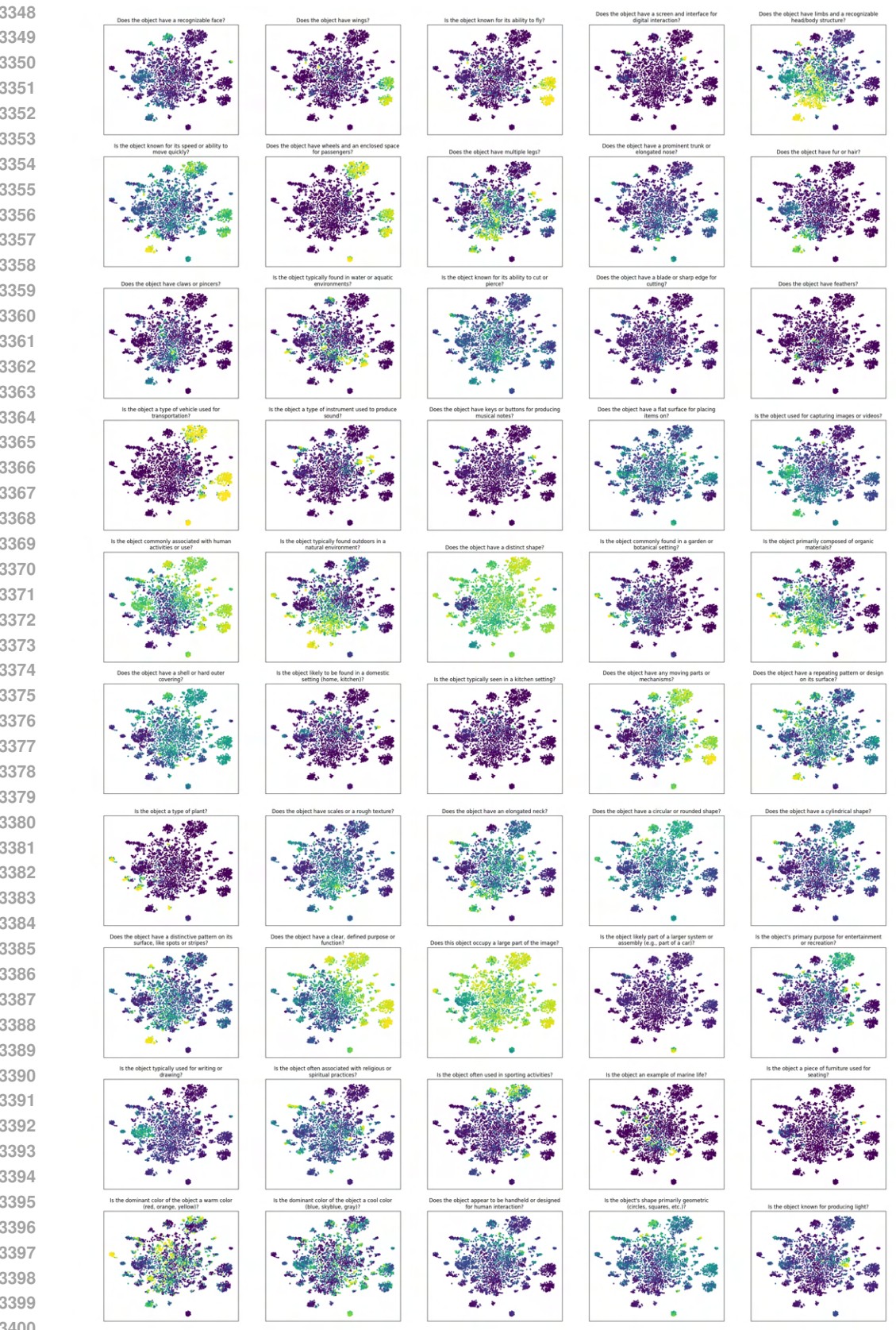

Figure 40: **Visualization of predicted result t-SNE embeddings for the multi-aspect of the Caltech101 testing dataset.**

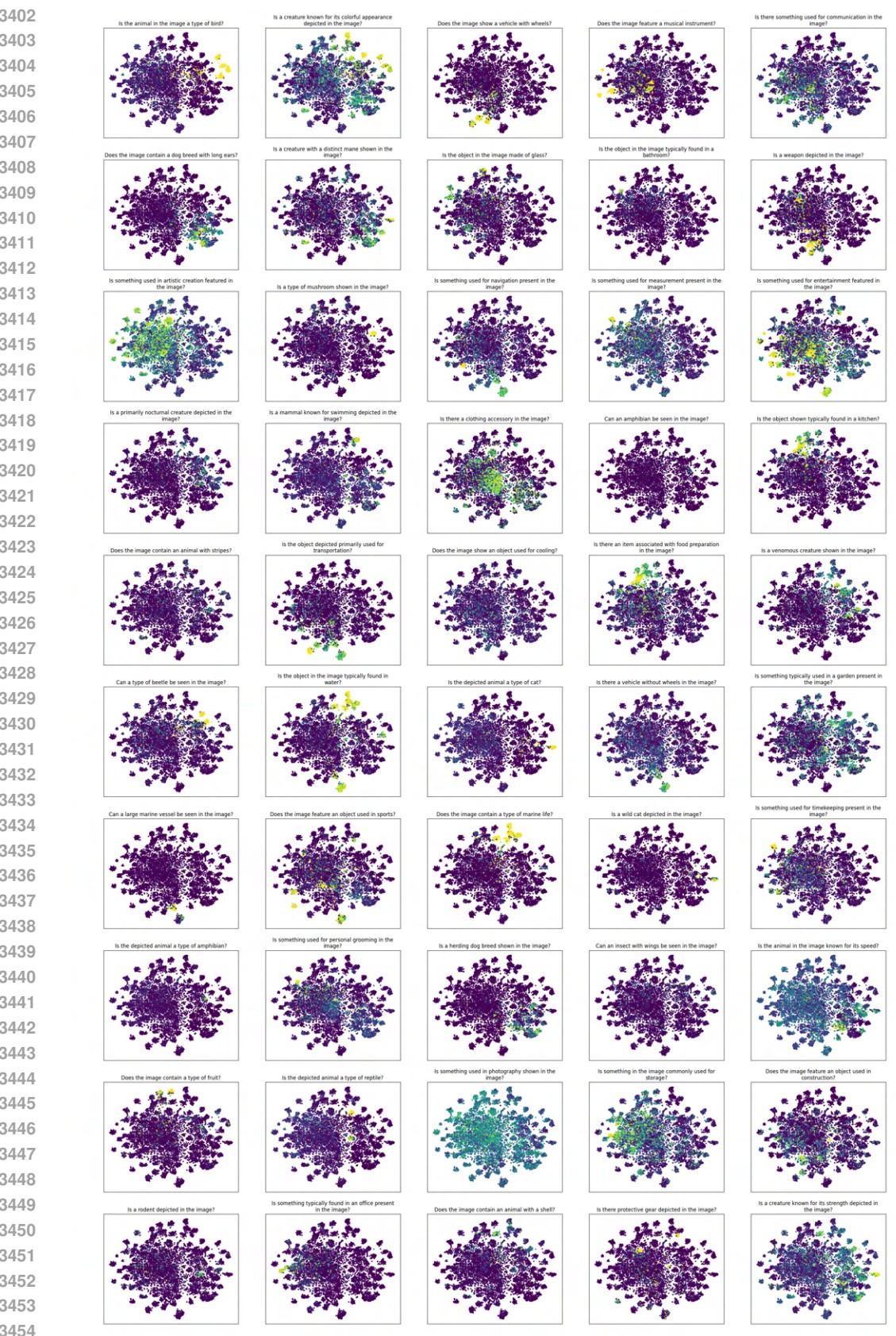

Figure 41: **Visualization of ground truth t-SNE embeddings for the multi-aspect of the Mini-ImageNet testing dataset.**

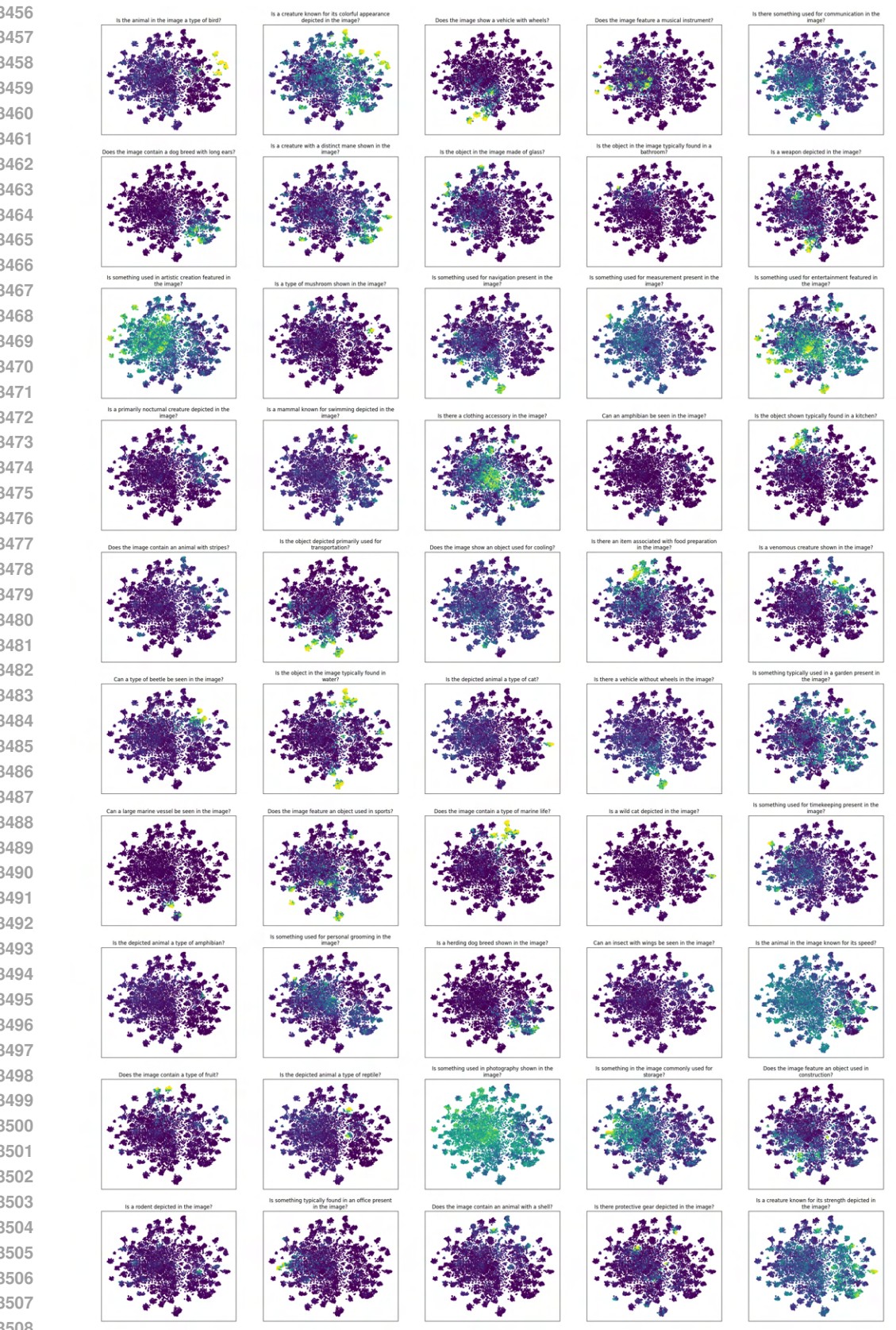

Figure 42: **Visualization of predicted result t-SNE embeddings for the multi-aspect of the Mini-ImageNet testing dataset.**