# OpenReview forum: "Multi-aspect Knowledge Distillation with Large Language Model"
_ICLR.cc/2025/Conference — Submitted to ICLR 2025_

### Official Review · Reviewer_gXGp · 2024-10-25

**Soundness:** 3
**Presentation:** 3
**Contribution:** 2
**Rating:** 5
**Confidence:** 4

**Summary:**

This paper introduces a novel approach to solving computer vision tasks, such as image classification and object detection, by enhancing conventional models' classification capabilities through knowledge distillation from Multimodal Large Language Models (MLLMs). The method involves expanding the dimensionality of the model's original logits, which improves classification accuracy. The paper provides numerous ablation experiments and conducts a thorough analysis of the results.

**Strengths:**

1. The paper combines traditional network models, such as ResNet, with MLLMs to enhance accuracy in classification and detection tasks.

2. It uses multi-aspect questions to extract knowledge from MLLMs, leveraging this knowledge to support classification.

3. The experiments are comprehensive.

**Weaknesses:**

The approach to utilizing knowledge distillation is a bit unclear—are you applying this strategy during training, or is it only used in inference? Additionally, there seems to be a lack of consideration for hallucination issues that may arise with GPT-4o during the generation of questions and responses.

**Questions:**

1. The approach to utilizing knowledge distillation is somewhat unclear. When are the multi-aspect logits extracted from the MLLM, and how are they incorporated into the model's training or inference objective?

2. Given that GPT-4o generates the multi-aspect questions and that the MLLM has not seen images from each category, especially considering these categories are often long-tailed and fine-grained. Do you have any validation or filtering steps in place for the generated questions and responses, or have you considered comparing the generated questions to human-curated ones? Which types of generated questions contribute most to performance improvements.

3. When generating responses to the multi-aspect questions for each image, has the potential hallucination issue within the MLLM been considered? How accurately can the MLLM (InternVL) answer these generated questions, and to what extent does the hallucination issue in InternVL affect the accuracy of its responses?

4. For the object detection task, have you attempted to use other datasets, such as the larger-scale LVIS?

I may reconsider my score based on your response to these issues.

---

> ### Author Response · Authors · 2024-11-22
>
> Dear reviewer gXGp,
> Thank you for your insightful comments. We address your concerns below.
>
> **Weaknesses + Question 1.** The approach to utilizing knowledge distillation is somewhat unclear. When are the multi-aspect logits extracted from the MLLM, and how are they incorporated into the model's training or inference objective?
>
> The multi-aspect logits are first extracted from the training dataset using MLLM before training the student model. These logits are then incorporated into the training process alongside the original training dataset. As a result, during inference, the image classification model is not only able to perform classification but also gains the capability to respond to aspects in a manner similar to MLLM.
>
> **Weaknesses + Question 2.** Given that GPT-4o generates the multi-aspect questions and that the MLLM has not seen images from each category, especially considering these categories are often long-tailed and fine-grained. Do you have any validation or filtering steps in place for the generated questions and responses, or have you considered comparing the generated questions to human-curated ones? Which types of generated questions contribute most to performance improvements.
>
> As mentioned in Sec 4.1, we manually reviewed the multi-aspects generated by GPT-4o and confirmed that all responses were valid for each dataset. Using the process detailed in the same section, GPT-4o initially generated 100 questions. These were then filtered to select the 50 most relevant ones, and we ranked them accordingly using GPT-4o.
> Figure 2 in the main paper shows that as the number of multi-aspects increases, the magnitude of performance improvement decreases. As shown in Figure 2, we empirically observed that higher-ranked aspects contribute more significantly to performance improvement than lower-ranked aspects.
>
> **Question 3.** When generating responses to the multi-aspect questions for each image, has the potential hallucination issue within the MLLM been considered? How accurately can the MLLM (InternVL) answer these generated questions, and to what extent does the hallucination issue in InternVL affect the accuracy of its responses?
>
> Thank you for the insightful feedback. InternVL demonstrates sufficiently good performance on the VLM benchmark (InfoVQA). As shown in Figure 5 of the main paper, we empirically found that it responds well to multi-aspect given a single image. InternVL effectively answers to the aspect “Does the animal have striking blue eyes?” for a Birman(cat) without relying on class information. Additionally, to ensure accurate answers to the generated questions, our method uses a prompt for MLLM that provides a single image and asks it to answer only with "Yes" or "No." Empirically, we observed a hallucination issue where MLLM generates irrelevant sentences instead of answering "Yes" or "No" when the image resolution is extremely low, such as below 100x100. To address this, we added a prompt stating, "Ignore the low resolution problem", for images with a resolution smaller than 128x128. As a result, InternVL accurately answers with "Yes" or "No" for all images in the datasets specified in the paper.
>
> **Question 4.** For the object detection task, have you attempted to use other datasets, such as the larger-scale LVIS?
>
> Unfortunately, we could not perform experiments on LVIS due to time constraints. Instead, we provide experiments on the scalability of tuning the MaKD Loss to further improve performance in Table 3.
> |     ResNet18     |   AP  |  AP50 |  AP75 |
> |:----------------:|:-----:|:-----:|:-----:|
> |     Baseline     | 33.18 | 53.54 | 35.31 |
> | Ours (alpha=1.0) | 33.35 | 53.90 | 35.58 |
> | Ours (alpha=2.0) | **33.48** | **53.98** | **35.76** |
>
> **Table 3. Additional object detection experiments with alpha values.** We run each experiment three times and report the average results.
>
> Also, our method's ability to extend to object detection, and incorporate multi-aspect questioning through visual grounding suggests its potential for improving performance in large-scale object detection datasets like LVIS. We believe this approach can lead to effective results in future studies.

---

> > ### Comment · Reviewer_gXGp · 2024-11-26
> >
> > Thank you for your comprehensive response and the supplementary experiments, which have helped address several of my initial concerns. However, I believe that the experimental setup still requires some further completion to ensure a more thorough evaluation. Additionally, the level of novelty could be further emphasized. I appreciate the efforts made to refine the work, but I believe further development is needed. Therefore, I have decided to keep my original rating.

---

> > > ### Author Response · Authors · 2024-12-04
> > >
> > > Thank you for your feedback. Our method leverages the generalized features learned by the MLLM, allowing the student model not only to predict classes but also to respond to multi-aspects. This makes our method both novel and simple, but effective. Additionally, Our experiments demonstrated notable performance improvements in fine-grained image classification, as well as scalability in ViT, KD, and object detection. We believe that our method has significant potential to achieve effective results in vision tasks and can be applied to real-world scenarios, such as in the medical domain.

---

### Official Review · Reviewer_xDGy · 2024-10-29

**Soundness:** 3
**Presentation:** 3
**Contribution:** 3
**Rating:** 5
**Confidence:** 3

**Summary:**

This paper presents a multi-aspect knowledge distillation framework that uses MLLMs to improve model performance in visual understanding and detection tasks. By expanding the model’s output dimensions, the method distills multi-aspect logits that encapsulate diverse visual and contextual features beyond standard class labels. Extensive experiments on various image classification datasets, complemented by thorough ablation studies, underscore the framework's effectiveness and robustness.

**Strengths:**

1. The core idea is simple but looks effective.
2. The paper writing is fluent and easy to follow.
3. The paper conducts experiments on six different fine-grained datasets and two different coarse-grained datasets. The results show that the proposed method achieves stable performance improvement, especially on the fine-grained datasets.
4. The ablation studies and related visualization are comprehensive and insightful.

**Weaknesses:**

1. The evaluation datasets in the paper are relatively small, and the model parameters appear insufficient in 2024 . Using ResNet18/34 as the primary model limits the assessment of the framework’s scalability. It would be valuable to test the framework on a larger dataset, such as ImageNet, and with a more complex model like ResNet101, to assess its effectiveness in a more challenging setting.

2. The paper lacks comparisons with other knowledge distillation (KD) baselines, which would provide a clearer benchmark for evaluating the proposed method’s relative performance.

3. The framework could explore additional ways to leverage the knowledge in MLLMs. For instance, distilling logits from the last token output by the MLLM after processing the input image may capture different aspects of visual representation.

4. While Section 5.5 discusses training time and computational cost, the analysis might be incomplete. The time required for MLLMs to annotate the training dataset should also be considered to provide a more comprehensive assessment of computational demands.

**Questions:**

1. I noticed that even using random logits leads to performance improvements (Table 3(b)). Could you clarify the underlying reason for this result?

---

> ### Author Response · Authors · 2024-11-22
>
> Dear reviewer xDGy,
> Thank you for your insightful comments. We address your concerns below.
>
> **Weakness 1.** The evaluation datasets in the paper are relatively small, and the model parameters appear insufficient in 2024 . Using ResNet18/34 as the primary model limits the assessment of the framework’s scalability. It would be valuable to test the framework on a larger dataset, such as ImageNet, and with a more complex model like ResNet101, to assess its effectiveness in a more challenging setting.
>
> Thank you for your valuable feedback. Unfortunately, due to time constraints, we were unable to conduct experiments on large datasets like the full ImageNet. To better assess the effectiveness of our approach, we included results for both ViT models and ResNet101 in Table 5.
> |           | StanfordCars |       |       | Mini-ImageNet |       |       |
> |:---------:|:------------:|:-----:|-------|---------------|-------|-------|
> | Model     | Base         | Ours  | Gap   | Base          | Ours  | Gap   |
> | ViT-B/16  | 10.88        | **11.62** | +0.74 | 43.41         | **44.18** | +0.77 |
> | ViT-B/16* | 85.72        | **86.13** | +0.41 | 96.64         | **96.87** | +0.23 |
> | ViT-B/32  | 9.32         | **9.73**  | +0.41 | 41.42         | **42.73** | +1.31 |
> | ViT-B/32* | 79.00        | **79.98** | +0.98 | 93.41         | **93.50** | +0.09 |
> | ResNet101 | 81.36        | **83.97** | +2.62 | 77.00         | **78.11** | +1.11 |
>
> **Table 5. Additional experiments with the ViT-based model and ResNet101 on StanfordCars and Mini-ImageNet.** * indicates that the model was trained using a pretrained model from ImageNet-1K. We run each experiment three times and report the average results.
>
> **These results demonstrate that our method is effective not only for ViT-based models but also for more complex CNN-based models like ResNet101.**
>
> **Weakness 2.** The paper lacks comparisons with other knowledge distillation (KD) baselines, which would provide a clearer benchmark for evaluating the proposed method’s relative performance.
>
> As pointed out by reviewers RKH6 and 2mpd, we conducted experiments with the latest KD methods [1, 2] and included the results in Table 2.
> |   StanfordCars  |                 |                     |    Caltech101   |                     |
> |:---------------:|:---------------:|:-------------------:|:---------------:|:-------------------:|
> |     Teacher     | ResNet34(80.93) | EfficientNet(86.41) | ResNet34(75.36) | EfficientNet(80.05) |
> |     Student     | ResNet18(77.53) |  MobileNetV1(82.84) | ResNet18(73.35) |  MobileNetV1(76.64) |
> |       Ours      |      83.38      |        85.43        |      75.76      |        79.14        |
> |        KD       |      79.62      |        85.11        |      74.53      |        78.71        |
> |     KD + LS     |      82.56      |        85.96        |      76.52      |        80.15        |
> |    KD + Ours    |      **83.44**      |        86.34        |      76.70      |        79.70        |
> |  KD + LS + Ours |      **83.44**      |        **86.69**        |      77.24      |        80.68        |
> |       DKD       |      82.55      |        85.93        |      76.37      |        79.95        |
> |     DKD + LS    |      82.82      |        86.13        |      76.57      |        80.39        |
> |    DKD + Ours   |      83.23      |        86.43        |      77.41      |        80.95        |
> | DKD + LS + Ours |      82.99      |        86.63        | **77.43**           |        **80.99**        |
>
> **Table 2. Comparisons with DKD and Logit Standardization in knowledge distillation methods on StanfordCars and Caltech101.** We run each experiment three times and report the average results.
>
> Our method demonstrates performance improvements when extended with [1, 2]. Specifically, on the StanfordCars dataset, where the teacher model is ResNet34 and the student model is ResNet18, our method outperforms other knowledge distillation methods [1, 2] (Ours: 83.38 > DKD [1] + Logit Standardization [2]: 82.82). **Also, other results show that extending our method further improves performance overall.**
>
> References
> [1] Zhao, Borui, et al. "Decoupled knowledge distillation." CVPR 2022.
> [2] Sun, Shangquan, et al. "Logit standardization in knowledge distillation." CVPR 2024.

---

> ### Author Response · Authors · 2024-11-22
>
> **Weakness 3.** The framework could explore additional ways to leverage the knowledge in MLLMs. For instance, distilling logits from the last token output by the MLLM after processing the input image may capture different aspects of visual representation.
>
> Thank you for the insightful suggestion. When MLLM generates sentences, the logits of the tokens are determined by the image or previously generated text tokens. This implies that the likelihood of generating tokens for the desired features may contain the visual representation. However, the logits for various text tokens related to the features we aim to extract may not always correspond precisely to the intended tokens. Our method addresses this by explicitly defining the features as aspects to obtain clear responses. By minimizing the computational cost associated with token generation, our approach ensures rapid and efficient responses. This makes it not only effective for traditional image classification models but also adaptable for tasks such as KD and object detection.
>
> **Weakness 4.** While Section 5.5 discusses training time and computational cost, the analysis might be incomplete. The time required for MLLMs to annotate the training dataset should also be considered to provide a more comprehensive assessment of computational demands.
>
> Thank you for your valuable feedback. We additionally provide the time required for MLLMs to extract aspect responses from the training dataset. For the StanfordCars dataset, it takes approximately 0.83 s ± 174 ms per image for a single aspect, while for Mini-ImageNet, it takes about 0.942 s ± 169 ms. The annotation time varies depending on the number of aspects and the size of the dataset.
>
> **Question 1.** I noticed that even using random logits leads to performance improvements (Table 3(b)). Could you clarify the underlying reason for this result?
>
> The clarification regarding Table 3(b) on distillation using random logits is as follows. According to studies on Teacher-free KD methods [4], teacher-free logits can also improve student performance when distilled. Similarly, using our extended logits as noise appears to impact performance, potentially as part of dark knowledge or through additional regularization effects.
>
> References
> [4] Yuan, Li, et al. "Revisiting knowledge distillation via label smoothing regularization." CVPR 2020.

---

> > ### Comment · Reviewer_xDGy · 2024-11-26
> >
> > Thank you for your detailed response and the additional experiments, which have addressed some of my concerns and strengthened the conclusions. However, after reviewing the comments from other reviewers, I believe it is necessary for the authors to complete the full experimental setup, such as including evaluations on the ImageNet dataset. This is particularly important given that related works cited in [1][2] also performed experiments on this dataset. Additionally, the novelty of the proposed approach remains somewhat limited. Furthermore, it appears that the performance improvement diminishes when evaluated on larger model settings. I encourage the authors to continue refining this project, and I have decided to maintain my current score.

---

> > > ### Author Response · Authors · 2024-12-04
> > >
> > > Thank you for your feedback. Our method differs from [1][2] by leveraging the generalized features learned by the MLLM, allowing the student model not only to predict classes but also to respond to multi-aspects. This makes the student model to learn more features even with a small dataset and significantly improves performance. As shown in Table 5 of the main paper, reducing the StanfordCars dataset by 40% still results in a 24.01% performance improvement compared to the baseline. We believe that our method has significant potential to achieve effective results in vision tasks, even in challenging low-resourced settings, and can be applied to real-world scenarios, such as in the medical domain.

---

### Official Review · Reviewer_2mpd · 2024-11-01

**Soundness:** 2
**Presentation:** 3
**Contribution:** 2
**Rating:** 5
**Confidence:** 4

**Summary:**

This paper proposes a new knowledge distillation method. It performs multi-aspect knowledge distillation with the LLM and the MLLM. LLM is utilized to generate multi-aspect questions by using the class and prompt. It further adopts the MLLM to extract the logit for multi-aspect questions and obtain the probabilities corresponding to yes token. The student is optimized by the original cross-entropy loss and the distilled binary cross-entropy loss. Extensive experiments are conducted to demonstrates the effectiveness of the proposed method. It also extends to object detection task to evaluate the great potential.

**Strengths:**

The paper is well-written, with a clear and logical flow from the introduction through to the conclusion. The authors present simple ideas in a straightforward manner, making the paper accessible to readers from diverse backgrounds. The experimental setup is meticulously organized, with each step of the process described in a way that facilitates reproducibility. The authors outline the methodologies, datasets, and evaluation metrics in clear subsections, allowing readers to follow the experimental design intuitively.

**Weaknesses:**

1、	Limited Experimental Setting：The experimental setting is narrow, which restricts the generalizability of the findings. The scale of datasets is small and may not be sufficient to demonstrate the robustness of the proposed method across different scenarios. Expanding the experimental scope to include more varied or challenging datasets such as the full ImageNet would significantly strengthen the paper.

2、	Lack of novelty: The proposed method directly adopts the MLLM’s output logit to perform distillation. The principle behind this design is not fully demonstrated. Why MLLM can help improve the performance of student and what features support this?

3、	Some details are missing, and some experimental comparisons are not fair. The parameter number of the MLLM is larger than the teacher model in the traditional KD. It is questionable whether the improvement is due to the large number of parameters or the inherent properties of the MLLM itself. What will happen to the performance of the student model if only adopting the large vision encoder in MLLM. Some comparison to the traditional methods is not fair. The basic KD adopted in experiment in classification is too old and many improved versions should be used.

4、	There is not the comparison to SOTA KD method in object detection and the baseline should also adopt the powerful setting.

**Questions:**

I hope the author can better explain the novelty of the paper and the principle of how the algorithm is really effective.

---

> ### Author Response · Authors · 2024-11-22
>
> Dear reviewer 2mpd,
> Thank you for your insightful comments. We address your concerns below.
>
> **Weakness 1.** Limited Experimental Setting：The experimental setting is narrow, which restricts the generalizability of the findings. The scale of datasets is small and may not be sufficient to demonstrate the robustness of the proposed method across different scenarios. Expanding the experimental scope to include more varied or challenging datasets such as the full ImageNet would significantly strengthen the paper.
>
> Thank you for your valuable feedback. The method proposed in our paper focuses on improving image classification performance, particularly when working with small datasets, including fine-grained datasets, which is challenging and practical. Our approach leverages MLLM to represent various visual features as aspects, distilling these aspect responses to enhance performance. Unfortunately, due to time constraints, we were unable to conduct experiments on large datasets such as the full ImageNet. To demonstrate greater robustness, we have included in Table 1 additional results showing performance improvements using larger parameter models like ViT.
> |           | StanfordCars |       |       | Mini-ImageNet |       |       |
> |:---------:|:------------:|:-----:|:-----:|:-------------:|:-----:|:-----:|
> |   Model   |     Base     |  Ours |  Gap  |      Base     |  Ours |  Gap  |
> | ViT-B/16 |     10.88    | **11.62** | +0.74 |     43.41     | **44.18** | +0.77 |
> |  ViT-B/16* |     85.72    | **86.13** | +0.41 |     96.64     | **96.87** | +0.23 |
> | ViT-B/32 |     9.32     |  **9.73** | +0.41 |     41.42     | **42.73** | +1.31 |
> |  ViT-B/32* |     79.00    | **79.98** | +0.98 |     93.41     | **93.50** | +0.09 |
>
> **Table 1. Additional experiments with the ViT-based model on StanfordCars and Mini-ImageNet.** * indicates that the model was trained using a pretrained model from ImageNet-1K. We run each experiment three times and report the average results.
>
> **Weakness 2.** Lack of novelty: The proposed method directly adopts the MLLM’s output logit to perform distillation. The principle behind this design is not fully demonstrated. Why MLLM can help improve the performance of student and what features support this?
>
> To the best of our knowledge, we are first on distillation of multi-aspect knowledge from MLLM, which is simple yet effective. Our method is different from traditional KD as it expands prediction logits instead of directly distilling class logits. Since MLLM has been trained on large datasets, it generates responses based on generalized knowledge. One of the key reasons MLLM improves the student's performance is that it trains the student model to predict the same responses for the given image features as MLLM. Through this process, we believe our method achieves significant performance gains in image classification tasks. Additionally, our approach is not limited to image classification. It can also be extended to traditional knowledge distillation and object detection tasks, where it demonstrates further performance improvements.

---

> ### Author Response · Authors · 2024-11-22
>
> **Weakness 3.** Some details are missing, and some experimental comparisons are not fair. The parameter number of the MLLM is larger than the teacher model in the traditional KD. It is questionable whether the improvement is due to the large number of parameters or the inherent properties of the MLLM itself. What will happen to the performance of the student model if only adopting the large vision encoder in MLLM. Some comparison to the traditional methods is not fair. The basic KD adopted in experiment in classification is too old and many improved versions should be used.
>
> Tab. 4 shows that our proposed method is more effective than performing KD based on class logits using MLLM as a teacher. Furthermore, Tab. 6 shows that our method can enhance performance even when applied in combination with existing teacher models. This highlights the distinction in approach, as our method builds upon the inherent properties of MLLM to achieve these improvements. As noted by Reviewer RKH6, we have also conducted experiments with two latest KD methods, [1, 2], and included the results in Table 2.
> |   StanfordCars  |                 |                     |    Caltech101   |                     |
> |:---------------:|:---------------:|:-------------------:|:---------------:|:-------------------:|
> |     Teacher     | ResNet34(80.93) | EfficientNet(86.41) | ResNet34(75.36) | EfficientNet(80.05) |
> |     Student     | ResNet18(77.53) |  MobileNetV1(82.84) | ResNet18(73.35) |  MobileNetV1(76.64) |
> |       Ours      |      83.38      |        85.43        |      75.76      |        79.14        |
> |        KD       |      79.62      |        85.11        |      74.53      |        78.71        |
> |     KD + LS     |      82.56      |        85.96        |      76.52      |        80.15        |
> |    KD + Ours    |      **83.44**      |        86.34        |      76.70      |        79.70        |
> |  KD + LS + Ours |      **83.44**      |        **86.69**        |      77.24      |        80.68        |
> |       DKD       |      82.55      |        85.93        |      76.37      |        79.95        |
> |     DKD + LS    |      82.82      |        86.13        |      76.57      |        80.39        |
> |    DKD + Ours   |      83.23      |        86.43        |      77.41      |        80.95        |
> | DKD + LS + Ours |      82.99      |        86.63        | **77.43**           |        **80.99**        |
>
> **Table 2. Comparisons with DKD and Logit Standardization in knowledge distillation methods on StanfordCars and Caltech101.** We run each experiment three times and report the average results.
>
> Our method demonstrates performance improvements when extended with [1, 2]. Specifically, on the StanfordCars dataset, where the teacher model is ResNet34 and the student model is ResNet18, our method outperforms other knowledge distillation methods [1, 2] (Ours: 83.38 > DKD [1] + Logit Standardization [2]: 82.82). Also, other results show that extending our method further improves performance overall. **These results demonstrate the potential of our method to be effectively extended to other KD methods and improve their performance**.
>
> **Weakness 4.** There is not the comparison to SOTA KD method in object detection and the baseline should also adopt the powerful setting.
>
> Thank you for your valuable feedback. Our method mainly focuses on image classification while also demonstrating its potential to be extended to tasks such as KD and object detection. Unfortunately, due to time constraints, we only include additional object detection results with tuned hyperparameter, alpha values (MaKD weight), in Table 3.
> |     ResNet18     |   AP  |  AP50 |  AP75 |
> |:----------------:|:-----:|:-----:|:-----:|
> |     Baseline     | 33.18 | 53.54 | 35.31 |
> | Ours (alpha=1.0) | 33.35 | 53.90 | 35.58 |
> | Ours (alpha=2.0) | **33.58** | **54.09** | **35.97** |
>
> **Table 3. Additional object detection experiments with alpha values.** We run each experiment three times and report the average results.
>
> The object detection results of tuning the alpha parameter in the MaKD loss, which show further performance improvements. These findings suggest that our method is also effective for object detection and can be applied to broader tasks in the future.
>
> References
> [1] Zhao, Borui, et al. "Decoupled knowledge distillation." CVPR 2022.
> [2] Sun, Shangquan, et al. "Logit standardization in knowledge distillation." CVPR 2024.

---

> > ### Comment · Reviewer_2mpd · 2024-11-30
> >
> > Thank you for your response and the additional experiments, which have filled in some missing details. I think the experiments with larger scale of datasets is neccessery and I encourage the authors to continue improving this project. I have decided to maintain my current score.

---

> > > ### Author Response · Authors · 2024-12-04
> > >
> > > Thank you for your feedback. Our method leverages the generalized features learned by the MLLM, allowing the student model to learn more features even with a small dataset. As shown in Table 5, reducing the StanfordCars dataset by 40% still results in a 24.01% performance improvement compared to the baseline. We believe that our method has significant potential to achieve effective results in vision tasks, even in challenging low-resourced settings, and can be applied to real-world scenarios, such as in the medical domain.

---

### Official Review · Reviewer_RKH6 · 2024-11-02

**Soundness:** 2
**Presentation:** 3
**Contribution:** 2
**Rating:** 5
**Confidence:** 4

**Summary:**

This paper starts from the perspective of how humans classify images, where humans typically consider multi-aspects such as context, shape, color, and other features. Motivated by this, the author proposes a multi-aspect knowledge distillation method that utilizes Multimodal Large Language Models (MLLMs) to improve image classification performance. By querying, extracting relevant logits, and expanding the model's output dimensions, the method achieves the knowledge learning of visual aspects and abstract knowledge. This method enhances the performance of baseline models across lots of experiments, demonstrating the potential of multi-aspect knowledge distillation in computer vision and other tasks.

**Strengths:**

1.This paper is written in a clear and straightforward manner, making it easy to quickly grasp the method's approach.
2.The paper conducted a lot of experiments, and the figures and tables are well-organized.
3.The authors claimed they are the first to offer a novel perspective on distilling multi-aspect knowledge regarding abstract and complex concepts. I have seen the author's efforts in the design of knowledge transfer.

**Weaknesses:**

1. The proposed method shows some improvement on some classic CNN-based models but lacks experiments on ViT-based models.
2. In the knowledge distillation task, the comparison is only done with KD, lacking comparisons with other knowledge distillation methods [1,2].
3. The improvement in object detection tasks is very limited in Tab7, and there is no comparison done on currently well-performing object detection methods. Object detection is inherently a more fine-grained visual task than classification. Still, the experiments in this paper do not demonstrate the method's effectiveness of multi-aspect knowledge distillation in detection.
4. The explanation for the poor zero-shot classification performance of MLLMs is missing in Tab 1. Incorrect knowledge could also be distilled to the student model.
5. Missing the training curve of MaKD Loss with the number of iterations. The visualization of t-SNE embeddings and the model's multi-aspect responses to a single image are presented in Fig 4 and 5. There is no overall evaluation of the model's responses to multi-aspect on the test dataset.

[1] Decoupled Knowledge Distillation
[2] Logit Standardization in Knowledge Distillation

**Questions:**

The paper develops a simple way to distill the multi-aspect knowledge of MLLM to perform image classification using a student model. The experiments show some improvements however I still believe that the contributions of this paper are quite limited. Additionally, the baseline models selected in this paper are quite outdated. In summary, the overall technical novelty of the direct injection of knowledge from large models seems incremental.

1. As shown in Tab 1, MLLMs perform badly in zero-shot classification on fine-grained image test datasets, how do we ensure that MLLMs provide correct answers across multiple aspects?
2. There are questions regarding the task details when extending to object detection: should the input to the MLLM be the object within the box or the entire image? The entire image may contain multiple objects, and the MLLM's response may not be accurate.

---

> ### Author Response · Authors · 2024-11-22
>
> Dear reviewer RKH6,
> Thank you for your insightful comments. We address your concerns below.
>
> **Weakness 1.** The proposed method shows some improvement on some classic CNN-based models but lacks experiments on ViT-based models.
>
> Following your suggestion, we conducted additional experiments by incorporating the ViT-B/16 and ViT-B/32 models, and have added new results to Table 1.
>
> || StanfordCars ||| Mini-ImageNet |||
> |:---------:|:------------:|:-----:|:-----:|:-------------:|:-----:|:-----:|
> |   Model   |     Base     |  Ours |  Gap  |      Base     |  Ours |  Gap  |
> | ViT-B/16 |     10.88    | **11.62** | +0.74 |     43.41     | **44.18** | +0.77 |
> |  ViT-B/16* |     85.72    | **86.13** | +0.41 |     96.64     | **96.87** | +0.23 |
> | ViT-B/32 |     9.32     |  **9.73** | +0.41 |     41.42     | **42.73** | +1.31 |
> |  ViT-B/32* |     79.00    | **79.98** | +0.98 |     93.41     | **93.50** | +0.09 |
>
> **Table 1. Additional experiments with the ViT-based model on StanfordCars and Mini-ImageNet**. * indicates that the model was trained using a pretrained model from ImageNet-1K. We run each experiment three times and report the average results.
>
> The hyperparameters were referenced from the ImageNet-1K training results in [3]. We applied our method to both learning from scratch models and pretrained models on ImageNet-1K and evaluated them on StanfordCars and Mini-ImageNet. The experimental results for the ViT-based models are as follows:
> - For learning from scratch models, applying our method led to performance improvements (StanfordCars: 9.32 → **9.73**, Mini-ImageNet: 41.42 → **42.73** on ViT-B/16 model). However, ViT models trained from scratch in small dataset, unless pretrained, does not achieve various regularization and data augmentation settings [3], their performance remains lower compared to CNN-based models reported in our paper.
> - For StanfordCars trained from ImageNet-1K pretrained models, applying our method to ViT-B/16 and ViT-B/32 resulted in performance gains of 0.41 and 0.98, respectively.
> - For Mini-ImageNet trained from ImageNet-1K pretrained models, applying our method to ViT-B/16 and ViT-B/32 resulted in performance gains of 0.24 and 0.09, respectively.
>
> **These experimental results demonstrate that our method enhances performance not only for CNN-based models but also for large models such as ViT.**
>
> **Weakness2.** In the knowledge distillation task, the comparison is only done with KD, lacking comparisons with other knowledge distillation methods [1,2].
> We conducted experiments on the other knowledge distillation methods you suggested [1, 2] following the settings described in our paper. The results have been included in Table 2.
> ||StanfordCars||Caltech101||
> |:---------------:|:---------------:|:-------------------:|:---------------:|:-------------------:|
> |     Teacher     | ResNet34(80.93) | EfficientNet(86.41) | ResNet34(75.36) | EfficientNet(80.05) |
> |     Student     | ResNet18(77.53) |  MobileNetV1(82.84) | ResNet18(73.35) |  MobileNetV1(76.64) |
> |       Ours      |      83.38      |        85.43        |      75.76      |        79.14        |
> |        KD       |      79.62      |        85.11        |      74.53      |        78.71        |
> |     KD + LS     |      82.56      |        85.96        |      76.52      |        80.15        |
> |    KD + Ours    |      **83.44**      |        86.34        |      76.70      |        79.70        |
> |  KD + LS + Ours |      **83.44**      |        **86.69**        |      77.24      |        80.68        |
> |       DKD       |      82.55      |        85.93        |      76.37      |        79.95        |
> |     DKD + LS    |      82.82      |        86.13        |      76.57      |        80.39        |
> |    DKD + Ours   |      83.23      |        86.43        |      77.41      |        80.95        |
> | DKD + LS + Ours |      82.99      |        86.63        | **77.43**           |        **80.99**        |
>
> **Table 2. Comparisons with DKD and Logit Standardization in knowledge distillation methods on StanfordCars and Caltech101**. We run each experiment three times and report the average results.
>
> Our method demonstrates performance improvements when extended with [1, 2]. Specifically, on the StanfordCars dataset, where the teacher model is ResNet34 and the student model is ResNet18, our method outperforms other knowledge distillation methods [1, 2] (Ours: 83.38 > DKD [1] + Logit Standardization [2]: 82.82). Also, other results show that extending our method further improves performance overall. **These results demonstrate the potential of our method to be effectively extended to other KD methods and improve their performance**.
>
> References
> [1] Zhao, Borui, et al. "Decoupled knowledge distillation." CVPR 2022.
> [2] Sun, Shangquan, et al. "Logit standardization in knowledge distillation." CVPR 2024.
> [3] Steiner, A., et al. "How to train your vit? data, augmentation, and regularization in vision transformers.” TMLR 2022.

---

> ### Author Response · Authors · 2024-11-22
>
> **Weakness 3.** The improvement in object detection tasks is very limited in Tab7, and there is no comparison done on currently well-performing object detection methods. Object detection is inherently a more fine-grained visual task than classification. Still, the experiments in this paper do not demonstrate the method's effectiveness of multi-aspect knowledge distillation in detection.
>
> Our paper mainly focuses on fine-grained image classification. Moreover, our method is not limited to image classification but demonstrates potential for broader applicability, including KD and object detection, as a promising direction for future work. Even though achieving state-of-the-art performance in object detection is not our focus, we would like to illustrate our potential and have included results in Table 3 showing performance improvements achieved by tuning a hyperparameter, alpha value (weight) of the MaKD Loss.
> |ResNet18|AP|AP50|AP75|
> |:----------------:|:-----:|:-----:|:-----:|
> |Baseline|33.18|53.54|35.31|
> |Ours (alpha=1.0)|33.35|53.90|35.58|
> |Ours (alpha=2.0)|**33.58**|**54.09**|**35.97**|
>
> **Table 3. Additional object detection experiments with alpha values.** We run each experiment three times and report the average results.
>
> Based on these results, we believe that the multi-aspect from MLLM can also be effective in object detection tasks and have the potential to be further developed through the application of various visual grounding techniques utilizing MLLM in the future work.
>
> **Weakness 4 + Question 1.** The explanation for the poor zero-shot classification performance of MLLMs is missing in Tab 1. Incorrect knowledge could also be distilled to the student model. As shown in Tab 1, MLLMs perform badly in zero-shot classification on fine-grained image test datasets, how do we ensure that MLLMs provide correct answers across multiple aspects?
>
> Thank you for kindly pointing out the areas where our paper could be improved. In Sec 5.1, we briefly mentioned the explanation for the poor zero-shot classification performance and the reasons for this poor performance were missing in Tab 1. We observed that MLLM demonstrates poor zero-shot classification performance on fine-grained datasets. This suggests that classifying subclasses within similar superclasses can be challenging if subclass-specific names are not adequately represented in the dataset used for MLLM training. However, when it comes to aspects, MLLM can effectively provide responses regarding the features of the objects visible in the image. For instance, if the object is a car such as “Rolls-Royce Phantom Drophead Coupe Convertible” or “Rolls-Royce Phantom Sedan”, and the aspect question is "Does the car have a convertible roof?" MLLM can answer whether the car has a roof or not, even without knowing the exact class of the car. As shown in Figure 5 of the main paper, the model effectively responds to the aspect “Does the animal have striking blue eyes?” for a Birman(cat) without relying on class information. Therefore, despite poor zero-shot classification performance, the multi-aspect questions allow MLLM to focus on and respond to specific features of the image or object. These aspects can be distilled to student model as seen in Figure 4 and 5 in the main paper. We will revise Tab 1 to include this information and also update the explanation in Section 5.1 of the main paper.

---

> ### Author Response · Authors · 2024-11-22
>
> **Weakness 5.** Missing the training curve of MaKD Loss with the number of iterations. The visualization of t-SNE embeddings and the model's multi-aspect responses to a single image are presented in Fig 4 and 5. There is no overall evaluation of the model's responses to multi-aspect on the test dataset.
>
>
> |    Dataset    |   L1   |  KL |
> |:-------------:|:------:|:------:|
> |  StanfordCars | 0.1174 | 0.1328 |
> | Mini-ImageNet | 0.0773 | 0.2562 |
> |   Caltech101  | 0.0881 | 0.2548 |
> | OxfordPets    | 0.0972 | 0.0969 |
> | DTD           | 0.1312 | 0.1463 |
> | FGVC-Aircraft | 0.0876 | 0.0423 |
>
> **Table 4. L1 distance and KL Divergence in aspect prediction between MLLM and ResNet18 on test dataset.**
>
> Thank you for your valuable comments. We included a figure illustrating the training curve of MaKD Loss with the number of iterations in Appendix A in our revised version. As shown in Figure 6 of Appendix A, MaKD Loss becomes close to 0 as training progreses. Additionally, we provide the overall evaluation of the model’s responses to multi-aspect on the test dataset using L1 distance and KL Divergence on Table 4. In the overall evaluation on the test dataset, the differences in L1 distance range between 7% and 13%. Also, regarding KL Divergence, we observed that the distributions of fine-grained datasets are closer compared to those of coarse-grained datasets. **These results demonstrate that our method effectively predicts multi-aspect features on fine-grained datasets than coarse-grained datasets.**
>
> **Question 2.** There are questions regarding the task details when extending to object detection: should the input to the MLLM be the object within the box or the entire image? The entire image may contain multiple objects, and the MLLM's response may not be accurate.
>
> Thank you for your insightful feedback. MLLM may struggle to provide accurate responses to aspects when multiple objects within an image have differing features or states. To address this, we believe incorporating visual grounding using bounding boxes for distinct objects in the image can enhance performance. By providing MLLM with a specific grounding target, it would be able to respond more accurately to the aspects related to the designated object.

---

> > ### Comment · Reviewer_RKH6 · 2024-11-28
> >
> > Thank you for your comprehensive response, addressing some of my concerns. The experiments show the potential of the proposed method, but we need to ensure MLLMs provide correct answers across multiple aspects. If some questions are not correctly answered by the MLLMs, the distilled model will receive incorrect supervision. Additionally, applying knowledge from multiple aspects to other vision tasks remains an area for further exploration. Therefore, I will raise my rating to 5.

---

> > > ### Author Response · Authors · 2024-12-04
> > >
> > > Thank you for your constructive feedback. As pointed out, MLLM may not answer some questions correctly. However, its responses are expressed in probabilities, indicating the confidence it has regarding the question. Additionally, our method trains the student model to predict multi-aspect response probabilities similar to MLLM, ensuring it to achieve a similar level of knowledge. Furthermore, as illustrated in Figure 3, the average probabilities of aspect questions represented reasonable correctness regarding each aspect in the corresponding question. Our experiments demonstrated notable performance improvements in fine-grained image classification, as well as scalability in ViT, KD, and object detection. We believe our method has significant potential to achieve effective results in vision tasks and can be applied to real-world scenarios, such as in the medical domain.

---

### Author Response · Authors · 2024-11-22

We thank the reviewers for their thoughtful and valuable feedback.

Our method provides a novel and simple yet effective approach that uses MLLM to infer multi-aspect knowledge, distill diverse information, and improve image classification on both fine-grained and coarse-grained datasets. It also shows potential for extending to tasks like KD and object detection.

We are encouraged that reviewers find our method is **novel** (gXGp), **straightforward and easy to follow** (RKH6, 2mpd), **insightful** (xDGy), **reproducible** (2mpd), and offers a **simple but effective way** (2mpd, xDGy),  to apply multi-aspect knowledge to different tasks. We also believe our approach can be extended to more challenging tasks, such as classification in the medical field.
Following the reviewers' feedback, we conducted the following experiments:
1. Based on the comments from reviewers RKH6, 2mpd, and xDGy regarding the lack of KD baselines, we added experiments using the latest knowledge distillation methods [1, 2].
2. As suggested by reviewers RKH6 and xDGy, we included experiments applying our method to models with larger models in the classification task.
3. To address reviewer RKH6's comment on object detection, we extended our experiments to demonstrate further applicability.
4. We incorporated missing figures and addressed specific comments raised by reviewer RKH6.
5. In response to reviewer xDGy’s feedback, we analyzed the additional annotation time required for MLLM to respond to aspects.
6. For the hallucination issues pointed out by reviewer gXGp, we explained how our method addresses and resolves these challenges.

We appreciate all reviewers again for their thoughtful and insightful comments, which give our paper going to be more clear and convincing. Please feel free to let us know for any additional questions or further clarifications.

References
[1] Zhao, Borui, et al. "Decoupled knowledge distillation." CVPR 2022.
[2] Sun, Shangquan, et al. "Logit standardization in knowledge distillation." CVPR 2024.

---

### Meta-Review · Area_Chair_Kw7u · 2024-12-18

**Metareview:**

The paper studies knowledge distillation for visual recognition models by querying the teacher models for multiple specific concepts. The authors demonstrate that this technique leads to better performance than vanilla knowledge distillation. The experiments in this paper are conducted on datasets - StanfordCars, OxfordPets, DTD, 102Flowers, FGVCAirCraft, CUB200, Caltech101 and miniImageNet using ResNet34 and ResNet18, EffiNet and Mb-N1 models.

Strengths
1. The paper is easy to follow and well written. The authors have provided a lot of details about the experiments and the method.
2. The authors have studied a lot of different datasets and model combinations in this work

Weaknesses
1. No comparisons to SOTA KD methods
2. Object detection results are limited
3. Datasets used in this work are relatively small scale, and thus the results may not hold on larger datasets + models
4. While the proposed method is different from existing KD work, there is limited explanation/insights into why this method works.

Justification
The AC read the paper and the reviews, and concluded that while interesting, the paper does have severe limitations in its experimental setup. The use of small scale datasets is concerning, and the lack of fine-grained visual tasks like object detection on LVIS style benchmarks limits the impact of this work. The reviewers also remain unconvinced about the paper.

**Additional Comments On Reviewer Discussion:**

The reviewers raised concerns about
1. Using Convolution architectures
2. Usage of small datasets like StanfordCars
3. Not comparing to SOTA KD methods
4. Limited explanation about why the method works


The authors have tried to address all three concerns by showing follow up experiments for (1,2,3) and offering explanations for (4)

The AC agrees with all the concerns raised by the reviewers. Points (2) and (4) are still not addressed and these are critical for the paper.

---

### Decision · Program_Chairs · 2025-01-22

Reject